# FROM PROMISE TO PRACTICE: REALIZING HIGH-PERFORMANCE DECENTRALIZED TRAINING

**Zesen Wang[†], Jiaojiao Zhang[†], Xuyang Wu[‡], Mikael Johansson[†]**
[†]KTH Royal Institute of Technology, [‡]Southern University of Science and Technology
{zesen,jiaoz}@kth.se, wuxy6@sustech.edu.cn, mikaelj@kth.se

## ABSTRACT

Decentralized training of deep neural networks has attracted significant attention for its theoretically superior scalability compared to synchronous data-parallel methods like All-Reduce. However, realizing this potential in multi-node training is challenging due to the complex design space that involves communication topologies, computation patterns, and optimization algorithms. This paper identifies three key factors that can lead to speedups over All-Reduce training and constructs a runtime model to determine when and how decentralization can shorten the per-iteration runtimes. To support the decentralized training of transformer-based models, we introduce a decentralized Adam algorithm that overlaps communications with computations, prove its convergence, and propose an accumulation technique to mitigate the high variance caused by small local batch sizes. We deploy our solution in clusters with up to 64 GPUs, demonstrating its practical advantages in both runtime and generalization performance under a fixed iteration budget[1].

## 1 INTRODUCTION

With the rapid advancement of deep neural networks (DNNs), distributed training has become the mainstream approach for efficiently scaling up models. Among the various parallelization strategies, data parallelism has emerged as the most straightforward and commonly used approach for distributing training across multiple devices. One of the most popular algorithms used in data-parallel training is All-Reduce (Li et al., 2020), known for its simplicity and its ability to maintain consistency with single-machine training.

However, All-Reduce training relies on high-speed network connections and homogeneous computational devices to ensure its efficiency (Zhang et al., 2020; Tandon et al., 2017). Large companies with abundant resources address these challenges by investing in specialized high-performance computing (HPC) clusters optimized for large-scale distributed training (Shoeybi et al., 2019). These clusters are equipped with cutting-edge uniform hardware and high-speed interconnects, making them ideal for All-Reduce-based training. Unfortunately, such resources are often inaccessible to the broader research community. Most researchers rely on self-built servers or shared clusters by universities or cloud service providers, which may lack the high-speed networking infrastructure necessary to fully leverage All-Reduce, thereby limiting the potential of the available computational resources.

Decentralized algorithms, which originally gained attention in the fields of consensus algorithms (Johansson et al., 2007; Shi et al., 2015) and privacy-preserving techniques (Yan et al., 2012), have recently been explored as alternatives to All-Reduce in distributed training, especially in environments with suboptimal network conditions. By replacing the costly All-Reduce operations with decentralized communication patterns, these algorithms can significantly reduce communication overhead. Over the past few years, decentralized algorithms have demonstrated promising practical speedup in setups with poor network performance (Lian et al., 2017; Assran et al., 2019; Yuan et al., 2022). Research in this area has focused on the convergence properties of gossiping on different communication topologies (Ying et al., 2021a; Takezawa et al., 2024), decentralized optimization algorithms (Lin et al., 2021; Yuan et al., 2021), and alternative communication patterns (Cattivelli &

---

[1]The experiment code is open-source at https://github.com/WangZesen/Decentralized-Training-Exp, and the extension code is open-source at https://github.com/WangZesen/Decent-DP.

Sayed, 2009). Despite this growing interest, decentralized approaches have yet to achieve the same level of popularity as All-Reduce-based methods in distributed training.

We believe there are several reasons for this: (1) Simply combining the best of each line of work does not necessarily lead to an effective overall system. The potential speedup of decentralized algorithms in real-world environments remains underexplored and poorly understood (Assran et al., 2019). (2) The generalization performance of models trained with decentralized methods often falls short compared to those trained with All-Reduce under the same iteration budget, raising concerns about the effectiveness of decentralized training (Lian et al., 2017; Assran et al., 2019). (3) Despite the growing interest in large language models (LLMs), there is still no well-suited decentralized version of the Adam optimizer, which is crucial for efficiently training transformer-based models in a decentralized setting.

This paper aims to develop a deeper understanding of decentralized training in practical setups and includes the following key contributions:

1. We propose a simple yet accurate runtime model that quantifies the impact of key environmental parameters and estimates potential speedups. This model enables the optimization of decentralized training configurations to achieve the best performance in different settings.

2. We design and analyze a decentralized variant of the Adam optimizer. Our algorithm leverages a collective gossip communication mechanism to enable parallelism between computation and communication. In addition, it introduces a novel accumulation technique that enhances performance and delivers promising experimental results.

3. We implement a PyTorch extension for decentralized training across multiple GPUs and nodes. Our implementation demonstrates strong performance on typical machine learning workloads, including image classification, machine translation, and GPT-2 pre-training. Extensive experiments validate the feasibility and practical benefits of decentralized training.

## 1.1 RELATED WORK

**Decentralized training** The combination of SGD updates with gossip communication (Lin et al., 2021; Yuan et al., 2021) has gained significant attention for its theoretical benefits: it retains the convergence rate as centralized/synchronous mini-batch SGD at a reduced communication cost. One line of work (Neglia et al., 2020; Koloskova et al., 2020; Ying et al., 2021a; Takezawa et al., 2024) focuses on the design of the communication topology to improve the guaranteed convergence rate in terms of the number of iterations. However, these papers demonstrate improvements in the number of iterations but do not show if these results translate to actual speedups in terms of total training time. A handful of more practical studies, *e.g.*, (Lian et al., 2017; Assran et al., 2019) report actual speedups of decentralized training with multiple workers, but they do not isolate system-level bottlenecks or bound the potential speedups. Moreover, Lian et al. (2017) focuses on low-bandwidth scenarios, and the implementation in Assran et al. (2019) groups GPUs in a node as a single unit for decentralization, which makes it sensitive to intra-node stragglers. Another line of work that is orthogonal to decentralized data parallelism is the slow momentum (Wang et al., 2019; Lin et al., 2021), which maintains an outer-loop momentum to improve the training loss. However, the method introduces additional communication and new hyperparameters.

**Variance in computation time** In the training of language models, one way to preprocess the input data is to group samples with similar lengths (Vaswani et al., 2017), but this may introduce the straggler effect because of non-uniform batches. Increasing the number of local mini-batches (Ott et al., 2019) or dropping samples on stragglers (Giladi et al., 2024) mitigate the problem but may worsen the generalization performance by altering the effective batch size (Keskar et al., 2016). In pre-training LLMs, it is common to pack the sequences from different documents (Raffel et al., 2020) to create uniform batches. However, the sequence pack technique has a negative influence on the training quality (Ding et al., 2024). Another possibility is to pad sequences to a fixed length (Smith et al., 2022), but this leads to lower resource utilization. Moreover, the methods are not compatible with tasks like video processing and seq-to-seq tasks. As for the performance of the decentralized training on imbalanced workloads, in Assran et al. (2019), the authors argued that decentralized training is more resilient to computation time variance by showing a smaller runtime variance. However, the speedup advantage brought by decentralized training remains unexplored.

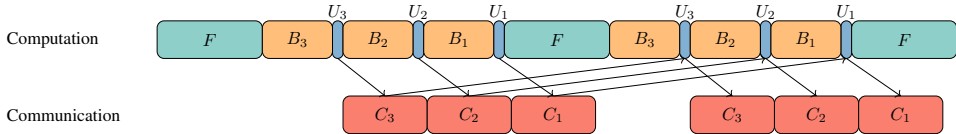

Figure 1: Timelines and dependency relations of decentralized training. $F$: forward pass, $B$: backward pass, $C$: decentralized communication/aggregation of model parameters with other workers, $U$: update model parameters. An arrow from task $X$ to $Y$ means that task $Y$ can start only after $X$ finishes.

**Orthogonal scalability techniques**   There are many other ways to improve scalability that are orthogonal to this work. This includes compression of gradients (Vogels et al., 2019) and models (Tang et al., 2018) to reduce the communication time, as well as model parallelism (Xu & You, 2023) and pipeline parallelism (Li & Hoefler, 2021) to scale up distributed training. These approaches could potentially be combined with our work to improve its scalability further.

## 2   IMPORTANT ASPECTS OF MULTI-NODE DECENTRALIZED TRAINING

The basic idea behind many decentralized optimization algorithms is to replace the global model parameter $x \in \mathbb{R}^D$ by local copies $x_i \in \mathbb{R}^D$ at each worker $i = 1, 2, \ldots, N$ and require that the local models reach consensus. In this spirit, a decentralized learning problem can be written as

$$\underset{\{x_1,\ldots,x_N\}}{\text{minimize}} \quad \frac{1}{N} \sum_{i=1}^{N} F_i(x_i), \quad \text{subject to } x_i = x_j, \ \forall i, j, \tag{1}$$

where $F_i(x_i)$ is the expected training loss of the local model $x_i$ on the data held by worker $i$. In our theoretical analysis, we let $F_i(x) = \mathbb{E}_{\xi_i \sim \mathcal{D}}[\ell(x; \xi_i)]$ where $\xi_i$ is a local data point of worker $i$ drawn from the distribution $\mathcal{D}$, and $\ell(\cdot)$ is a non-convex but smooth function. The assumption that each worker can draw samples from the same distribution $\mathcal{D}$ is reasonable if not for the privacy-preserving purpose (Ying et al., 2021a; Lian et al., 2017).

Decentralized algorithms for solving (1) allow the local model $x_i$ to disagree during the training process but drive them towards the same stationary point of the global training loss. We focus on adaptive momentum versions of decentralized gradient descent,

$$x_i^{(t)} \leftarrow -\alpha^{(t)} d_i^{(t)} + \sum_{j \in \mathcal{N}_i^{(t)}} w_{ij}^{(t)} x_j^{(t-1)}. \tag{2}$$

Here, $\alpha^{(t)}$ is the learning rate, $-d_i^{(t)}$ is an update direction that reduces the value of $F_i$ at $x_i^{(t-1)}$ and the final term accounts for the averaged model parameters of neighboring workers and drives the local parameters towards the same first-order stationary point. For decentralized SGD, $d_i^{(t)} = \nabla \ell(x_i^{(t-1)}; \xi_i^{(t)})$ while the expression is more complex for the adaptive momentum variations that we develop in § 4. The weights $w_{ij}^{(t)}$ satisfy $w_{ij}^{(t)} > 0$ if $i$ and $j$ communicate at iteration $t$ or $i = j$, and $0$ otherwise. We define $\mathcal{N}_i^{(t)}$ as the set of workers $j$ (including $i$ itself) such that $w_{ij}^{(t)} > 0$. The weights are selected so that if $d_i^{(t)} = 0$ for all $i$ and all $t$ in (2), the local models will converge to the same value; see § 4 for details.

Next, we will describe three factors that are essential to consider in order to implement (2) efficiently.

### 2.1   EXPLOITING OVERLAPPING COMMUNICATION AND COMPUTATION

Modern accelerators like Nvidia GPUs are capable of interleaving communications with computations. After deciding which training scheme to use, hiding communication by computation is one of the key optimizations in the design of an efficient distributed training system. For the decentralized implementation of (2), a general idea of achieving it is to restrict the search direction $d_i$ to only depend on the local model. While for DNN training, we can take this idea further.

Inspired by the gradient bucketing technique used in data-parallel training in PyTorch (Li et al., 2020), we divide both gradients and models into buckets corresponding to specific layers or groups of layers. Due to the sequential nature of backpropagation, we can begin communicating the gradients

| Setups \ Topology | Complete | One-peer Exp. | One-peer Ring | Alternating Exp Ring |
|---|---|---|---|---|
| 4×4×A100, 100Gbps | 55.31 s | 55.44 s | 32.97 s | **29.55** s |
| 4×4×A40, 25Gbps | 395.99 s | 429.47 s | 217.90 s | **185.93** s |

Table 1: Communication time of performing averaging operations for 8196 iterations over three 25MB FP32 tensors (selection of 25MB is based on the default bucket size used in PyTorch DDP (Li et al., 2020)). **Setup 1**: 16 A100 GPUs on 4 nodes inter-connected by 100Gbps Infiniband. **Setup 2**: 16 A40 GPUs on 4 nodes inter-connected by 25Gbps Ethernet. Here, Complete means global averaging. All other topologies are illustrated in Appendix A.4.

of earlier finished layers while the gradients of later layers are still being computed. As illustrated in Figure 1, we split the execution of the backward pass into buckets and interleave the corresponding local model updates. After updating a bucket, the corresponding communication can be initiated, and its results are not needed by the worker and its neighbors until the update of the same bucket in the next iteration. This significant overlap is possible because we have made the decentralized update independent of neighbor information within the same iteration. For All-Reduce training[2], which aggregates gradients instead of models, communication cannot overlap with the forward pass and the backward pass of the last bucket. This is because the forward pass needs the complete aggregation of gradients from the last iteration, and no gradient is available before the backward pass of the last bucket. This indicates that decentralized training, using the scheme described in equation 2, can utilize simultaneous computational resources more efficiently.

Making the update direction calculation independent of the local models at other workers in the same iteration is essential for overlapping computations and communications. Updates on the form $x_i^{(t)} \leftarrow \sum_{j \in \mathcal{N}_i^{(t)}} w_{ij}^{(t)}(x_j^{(t-1)} - \alpha^{(t)} d_j^{(t)})$, as used in (Lin et al., 2021; Gao & Huang, 2020), cannot easily achieve this; see Appendix B.2.

## 2.2 RESPECTING HETEROGENEOUS COMMUNICATION COSTS

A key advantage of well-designed decentralized training over All-Reduce training is the reduced communication cost due to the sparsity of gossip communication. Unlike All-Reduce, where all workers must participate in each iteration, decentralized training allows workers to communicate only with their immediate neighbors. A drawback with gossiping is that it only provides an approximate average that converges asymptotically to the true global average. Several studies have characterized how the convergence rate of gossiping, affecting both iteration complexity and generalization performance in decentralized training, is influenced by the communication topology (Xiao & Boyd, 2004; Shi et al., 2015; Nedić et al., 2018; Ying et al., 2021a; Takezawa et al., 2024). In practical cluster environments, latency and bandwidth can vary significantly between nodes. While communication between GPUs within a single compute node is very fast, communication between nodes is slower and subject to congestion at switches and other network interfaces. Thus, when designing a communication topology for decentralized training in such environments, it is crucial to consider both the degree of connectivity, which affects the speed of information propagation, and the heterogeneity of networking connections, which impacts the execution time of each communication round.

To demonstrate the practical impact of topology and heterogeneous connections on communication time, we set up an experiment with two scenarios: four nodes, each equipped with four Nvidia GPUs, are interconnected by either 25Gbps Ethernet or 100Gbps Infiniband, Table 1 shows the communication time of performing averaging operations with different communication topologies. Even though the One-peer Exponential topology enjoys rapid convergence of consensus errors in terms of iterations, it requires more time than the Complete topology in this setup due to frequent inter-node communication. The One-peer Ring topology nearly halves the per-iteration communication time compared to the Complete topology. However, it exhibits a slower convergence rate of consensus errors than both the Complete and the One-peer Exponential topology in terms of communication rounds, which could potentially influence the theoretical generalization guarantees. However, experiments in (Kong et al., 2021) have shown that as long as the consensus error is below

---

[2]We provided the pseudocode of the distributed algorithm using All-Reduce in Appendix A.1 and its timeline in Appendix A.3. All-Reduce is also considered in our experiments for comparison.

a certain value, it does not negatively affect performance. We have observed the same phenomenon in our practical deployments.

Unlike other implementations of decentralized training, such as those described in Assran et al. (2019); Lin et al. (2021); Ying et al. (2021a) which perform decentralized communication on the node level, we do not synchronize gradients within nodes. This implementation choice can provide significant resilience to variations in computation time, which will be discussed in § 2.3.

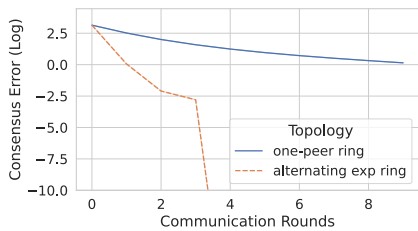

To adapt to the communication imbalance in the two clusters used in our experiments, we designed an Alternating Exponential Ring (AER) topology, see Figure 12 in Appendix A.4. The last column in Table 1 and the comparison in Figure 2 demonstrate that the proposed topology offers advantages in both communication time and convergence rate. Additionally, our numerical experiments in Section 5 demonstrate that it is able to strike a favourable balance

Figure 2: Comparison of consensus errors by communication rounds with four 4-GPU nodes.

between the connectivity of communication topologies and per-iteration runtime while maintaining comparable test accuracy.

It is important to note that the effectiveness of a specific topology may vary depending on the operational environment. By quantifying the communication costs associated with different connections and the congestion caused by concurrent communications, one can formulate an optimization problem that targets both the convergence rate and average runtime as objectives. We leave this as future work.

## 2.3 REDUCING THE SENSITIVITY TO VARYING COMPUTATION TIMES

The timeline in Figure 1 shows a single worker and assumes that all other workers run their corresponding tasks in perfect synchrony. This is certainly not realistic. Even in homogeneous HPC environments, variable computation times appear due to system noise (Hoefler et al., 2010) and workload imbalance (Ott et al., 2019). Figure 3 shows the distribution of actual gradient computation times for two different tasks. For image classification with ResNet-50 (He et al., 2016), even though all mini-batches are of identical size, the variance still exceeds $0.001$. For neural machine translation, the variance is much higher since mini-batches vary in size even if we batch together sequences with similar lengths (Vaswani et al., 2017; Ott et al., 2019).

In All-Reduce training, workers have to wait for the slowest one, which makes the approach sensitive to workload variability. If the computation times of workers would be i.i.d. and follow a normal distribution, the expected value of the maximum of the $N$ computation times is asymptotically $\Theta(\sqrt{\log N})$ (Giladi et al., 2024). Hence, we can expect it to be less efficient as $N$ increases. For decentralized training, as shown in Appendix A.5, it is still possible to hide some (and in some cases all) workload variability. Simulations of the runtimes of All-Reduce and decentralized training, shown in Figure 4, confirm these predictions. The runtime for All-Reduce increases with $N$, while the runtime for decentralized training remains essentially constant. This suggests that decentralized training could achieve better strong scaling efficiency (fixed global batch size).

In other decentralized frameworks like SGP (Assran et al., 2019)[3], intra-node gradient synchronization is typically performed before the local update, and the parameter communication occurs at node level. However, this design becomes suboptimal in terms of runtime when the variance in computation time is non-negligible, since gradient synchronization is susceptible to intra-node stragglers. To better leverage the resilience of decentralized training to variations in computation time, we decouple the dependency of the local update on the gradient information of its neighbors within the same iteration. To demonstrate the benefits of the new design, we perform transformer experiments under heavy computation workloads ($2\times4\times$A100 and 100K tokens as global batch size) and fast network connections (NVLink within the node and 100Gbps Infiniband inter-connecting nodes) to make sure the network is not the bottleneck. The results show that removing the dependency decreases the per-iteration runtime by $10.95\%$ (from 225.09 ms to 200.43 ms). For more comparisons, see § 5.2.

---

[3]https://github.com/facebookresearch/stochastic_gradient_push

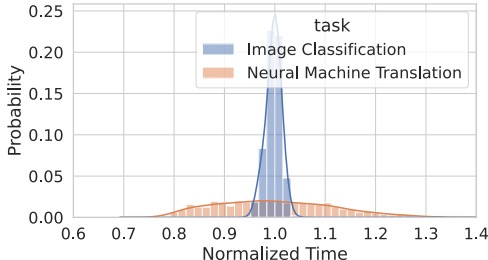
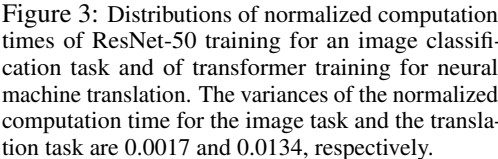
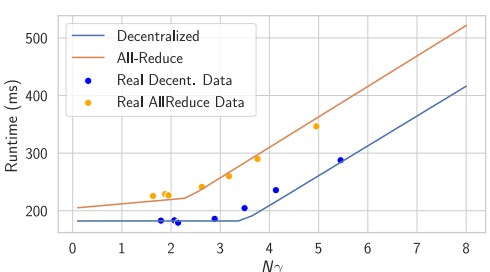

Figure 3: Distributions of normalized computation times of ResNet-50 training for an image classification task and of transformer training for neural machine translation. The variances of the normalized computation time for the image task and the translation task are 0.0017 and 0.0134, respectively.

Figure 4: Comparison of the predicted runtime by runtime model and the actual runtime for translation task based on transformer model. The lines are predicted runtime based on the runtime model with scaling to the real runtime in milliseconds. The scatters are the measured runtime.

## 3 RUNTIME MODEL AND DESIGN INSIGHT

In this section, we propose a runtime model that characterizes how the per-iteration runtime depends on critical system parameters and workload characteristics. The model is simple yet surprisingly accurate when validated against experiments on real systems (see Appendix A.5), and helps to identify when and with what means we can obtain significant reductions in the per-iteration runtime.

The model considers neural network training in an environment with $N$ workers and assumes that the model parameters are organized into $b$ buckets. It uses the time that a single worker needs to process the forward pass with the global batch size as unit time. In this way, the forward pass of one worker takes $1/N$ time units while the backward pass is assumed to take $2/N$ time units. The time to update a bucket is $\theta$ time units. To characterize the system and the training workload, we introduce three additional parameters: (1) $\gamma \in (0, \infty)$ is the time taken for a global All-Reduce on one bucket; this parameter is large in systems with slow communication or many workers, (2) $\omega \in (0, 1]$ is the ratio of the time taken by a decentralized communication round and a global All-Reduce. A small $\omega$ means a sparse decentralized communication topology, and (3) $\sigma^2$ is the variance of a truncated normal distribution that models the varying computation time. A large $\sigma^2$ means the computation time taken by the forward pass and backward pass may vary significantly. Based on the workload measurements in Figure 3, we restrict the computation time between 0.5x and 1.5x of its average value.

The model is detailed in Appendix A.5. In brief, it simulates randomly varying forward and backward pass calculation times and respects the dependency relationships between communication and update tasks described in Figure 1 and Figure 13.

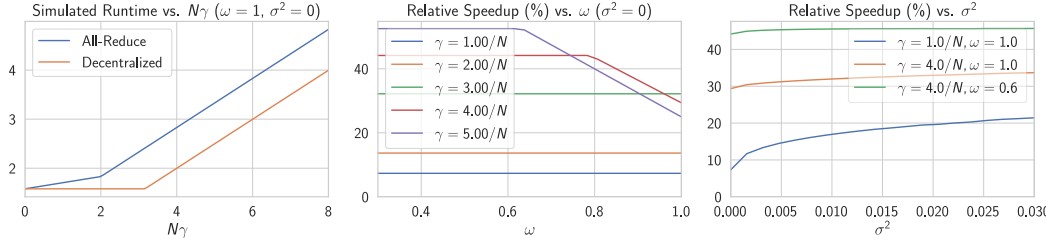

Figure 5: Simulation by the runtime model. In the simulation, $N = 8$ (8 workers), $b = 4$ (4 buckets, from ResNet-50 experiments), $\theta = 0.2$ (time taken by updating one bucket is 0.2 unit time), and the communication topology of decentralized training is complete topology[4].

Figure 5 shows some representative results generated by the runtime model. The leftmost figure illustrates how the per-iteration runtime of All-Reduce and decentralized training depends on $\gamma$ when $\omega = 1$ and $\sigma^2 = 0$. For decentralized training, the communication time can be completely hidden by the corresponding forward, backward, and bucket update times, as long as the network is fast enough ($\gamma \leq (3 + \theta)/N$). After this point, the per-iteration time increases as $\Theta(b\gamma)$. In All-Reduce training,

on the other hand, the last bucket can never be hidden and the per-iteration time increases also for small values of $\gamma$. More importantly, since the All-Reduce time of buckets can only be hidden by the corresponding backward pass, the more rapid increase in runtime starts already at $\gamma = 2/N$.

Figure 5 (middle) shows the relative speedup of decentralized training over All-Reduce as a function of $\omega$ for different values of $\gamma N$. For $\omega = 1$ and a fixed value of $\gamma N$, the speedup is just the ratio of the two curves shown in the leftmost figure. Decreasing $\omega$ reduces the time for a bucket communication in the decentralized setting, which effectively moves us to the left along the decentralized runtime graph. This leads to increasing speed-ups until $\omega\gamma = (3 + \theta)/N$, after which no runtime improvements are obtained by decreasing $\omega$ further. This explains why no effect of $\omega$ is seen when $\gamma N \leq (3 + \theta)$, and the speed-ups saturate below $\omega = (3 + \theta)/\gamma N$. The model can also be used to estimate the best possible speedup. Specifically, when $\sigma^2 = 0$, we find

$$\text{best possible speedup}(\gamma) = \begin{cases} 1 + \frac{1}{b} \cdot \frac{N\gamma}{3 + \theta N} & \gamma \in (0, 2/N], \\ 1 + \frac{N\gamma - 2 + 2/b}{3 + \theta N} & \gamma \in (2/N, \infty). \end{cases} \tag{3}$$

As $N$ grows large, the second case suggests that the speedup tends towards $1 + \gamma/\theta$. Since $\gamma$ increases with $N$, the model suggests that the speedups will increase in larger networks.

Figure 5 (right) shows the resilience of the computation time variance of decentralized training. The three combinations of $\gamma$ and $\omega$ represent the scenarios that (a) computation-bound ($\gamma = 1/N$, $\omega = 1$) (b) communication-bound ($\gamma = 4/N$, $\omega = 1$) (c) communication-bound and using a decentralized communication topology ($\gamma = 4/N$, $\omega = 0.6$). In the three scenarios, increasing $\sigma^2$ brings moderate improvement in the speedup of decentralized training.

As for the influence of communication topologies on runtime, they do influence runtime because of the changes in the dependencies of the operations, but the influences caused by adapting decentralized training and reduced communication cost by sparser topologies (see § A.5.2 in appendix).

In Figure 4, we validate the accuracy of the runtime model by comparing the predictions with the actual runtime as $\gamma$ increases. The parameters $b = 7$, $N = 32$, $\theta = 0.364$, $\sigma^2 = 6.95 \times 10^{-3}$ measured from the profiling results are used for the predictions, and we manipulate the global batch size to mimic different values of $\gamma$. Given its simplicity, the model is surprisingly accurate.

## 4 DECENTRALIZED ADAM AND CONVERGENCE

The per-iteration runtime that we have just studied is only part of the story. Equally important is the ability of the decentralized optimization algorithm to converge quickly and reliably toward a solution with good generalization properties. This section introduces a decentralized Adam algorithm (DAdam, described in Algorithm 1) that enables overlapping communications and computations and proves its convergence. The notations are defined in Appendix B.1.1.

---

**Algorithm 1** Decentralized Adam on worker $i$

---

1: $m_i^{(0)}, v_i^{(0)} \leftarrow 0$; $x_i^{(0)} \leftarrow x^{(0)}$
2: **for** $t = 1, 2, \ldots, T$ **do**
3: $\quad g_i^{(t)} \leftarrow \nabla\ell\big(x_i^{(t-1)}; \xi_i^{(t)}\big)$
4: $\quad m_i^{(t)} \leftarrow \beta_1 m_i^{(t-1)} + (1 - \beta_1)g_i^{(t)}$
5: $\quad v_i^{(t)} \leftarrow \beta_2 v_i^{(t-1)} + (1 - \beta_2)\big[g_i^{(t)}\big]^2$
6: $\quad x_i^{(t)} \leftarrow -\alpha \frac{m_i^{(t)}/(1-\beta_1^t)}{\sqrt{v_i^{(t)}/(1-\beta_2^t)} + \epsilon} + \sum_{j \in \mathcal{N}_i} w_{ij} x_j^{(t-1)}$
7: **end for**

---

Algorithm 1 can be cast into the form of equation 2 with a particular search direction. For simplicity, we use a static topology with mixing matrix $W$, i.e., $W^{(t)} = W$ for all $t$. In every iteration, each worker $i$ first draws a mini-batch from its local data and computes a local stochastic gradient $g_i$. Then, it proceeds similarly to the standard Adam scheme to update the local first- and second-order momentum $m_i$ and $v_i$, respectively. Subsequently, worker $i$ implements the mixing of neighbor

models and uses the result to update the local model where $\alpha$ is a step-size and $\epsilon, \beta_1, \beta_2$ are parameters. Note that $m_i$, $v_i$ and the resulting search direction do not depend on models $x_j$ of neighboring workers. Therefore, the algorithm supports overlapping communication and computations as discussed in Section 2.1. The comparison of Algorithm 1 with other decentralized Adam-based methods is in Appendix B.2.

Next, we establish convergence of Algorithm 1, based on the following Assumptions.

**Assumption 1.** *For each worker $i$, the gradient $\nabla F_i$ is L-Lipschitz continuous:*

$$\|\nabla F_i(x) - \nabla F_i(y)\|_2 \leq L\|x - y\|_2, \ \forall x, y \in \mathbb{R}^D.$$

**Assumption 2.** *There exists a constant $R \geq \sqrt{\epsilon}$ such that $\|\nabla \ell(x; \xi_i)\|_\infty \leq R - \sqrt{\epsilon}$ almost surely.*

**Assumption 3.** *The mixing matrix $W = [w_{ij}]$ is symmetric, non-negative, and doubly stochastic. The eigenvalues of $W$ in order satisfies $1 = \lambda_1 > \lambda_2 \geq \cdots \geq \lambda_N > -1$.*

Assumptions 1 and 2 are used in Défossez et al. (2020) to provide a simple analysis for centralized Adam. Assumption 3 is standard in decentralized learning and holds when the underlying communication topology is connected. Under Assumption 3, we denote $\lambda = \max\{|\lambda_2|, |\lambda_N|\} \in [0, 1)$. Fix the number of iterations as $T$. We define $\tau_T$ as a random index from $\{0, \ldots, T-1\}$ with distribution

$$\forall \widetilde{t} < T, \quad \mathbb{P}[\tau = \widetilde{t}] \propto 1 - \beta_1^{T-\widetilde{t}}, \tag{4}$$

and denote $F(\bar{x}) := \frac{1}{N} \sum_{i=1}^N F_i(\bar{x})$ where $\bar{x} = \frac{1}{N} \sum_{i=1}^N x_i$. Then, we have the following theorem.

**Theorem 4.1.** *Under Assumptions 1–3, if $0 < \beta_1 < \beta_2 < 1$, for Algorithm 1, we have*

$$\mathbb{E}\left[\left\|\nabla F(\bar{x}^{(\tau)})\right\|_2^2\right] \leq \frac{4R}{\alpha \tilde{T}}\left(F(\bar{x}^{(0)}) - F_*\right) + E\left[\frac{1}{\tilde{T}}\ln\left(1 + \frac{R}{\epsilon(1-\beta_2)}\right) - \frac{T}{\tilde{T}}\ln(\beta_2)\right], \tag{5}$$

*where $\bar{x}^{(t)} = \frac{1}{N} \sum_{i=1}^N x_i^{(t)}$, $\tau$ is defined in equation 4, $\tilde{T} = T - \frac{\beta_1}{1-\beta_1}$, $F_*$ is the optimal value of equation 1, and $E = \frac{24DR^2\sqrt{1-\beta_1}}{\sqrt{1-\beta_2}(1-\beta_1/\beta_2)^{3/2}} + \frac{2\alpha DLR(1-\beta_1)}{(1-\beta_2)(1-\beta_1/\beta_2)} + \frac{4\alpha^2 L^2 D\beta_1}{(1-\beta_1/\beta_2)(1-\beta_2)^{3/2}} + \frac{8\alpha^2(1+\lambda^2)RL^2D\sqrt{1-\beta_1}}{(1-\lambda^2)^2(1-\beta_2)(1-\beta_1/\beta_2)\sqrt{\epsilon}}$.*

We refer the readers to Appendix B.4 for the proof. Compared to Défossez et al. (2020, Theorem 4) for the single-machine Adam, the error bound in Theorem 4.1 only has one additional term – the last term in $E$ which is proportional to $\alpha^2$ and can be ignored for sufficiently small values of $\alpha$.

## 5 NUMERICAL EXPERIMENTS

In this section, we compare decentralized training with All-Reduce training and assess the generalization performance and practical training-time speedups in cluster environments. We consider three large-scale tasks: neural machine translation, image classification, and GPT-2 pretraining. For neural machine translation, following Vaswani et al. (2017), we trained the transformer ($\sim$65M parameters for base variant and $\sim$213M parameters for big variant) on the English-German and English-French WMT14 dataset (Bojar et al., 2014). In each run, we evaluate the trained model by BLEU (Papineni et al., 2002) and METEOR (Banerjee & Lavie, 2005) on the test set. In the image classification task, we trained ResNet-50 (He et al., 2016) on ImageNet-1K (Deng et al., 2009). In each run, we evaluate the trained model by Top-1 and Top-5 accuracies on the validation set. For the GPT-2 pretraining task, we trained GPT-2 (small) (Radford et al., 2019) on OpenWebText Gokaslan et al. (2019), and we report the training and validation losses of the trained models. The baseline, All-Reduce training, uses Adam-based optimizers for all experiments. Detailed descriptions of all tasks and experimental setups can be found in Appendix A.6.

### 5.1 GENERALIZATION PERFORMANCE

To make decentralized training useful, it should not compromise the generalization performance of the trained model. As shown in Tables 2 and 3, decentralized training can obtain a generalization performance that is comparable with All-Reduce training. In all experiments, AccumAdam outperforms the others, while DAdam and All-Reduce perform similarly.

| | | BLEU ↑ / METEOR ↑ | |
|---|---|---|---|
| Method | Topology | En-De / Transformer (base) | En-Fr / Transformer (big) |
| AllReduce | Complete | 27.79 ± 0.16 / 0.5465 ± 0.0018 | 45.70 ± 0.30 / 0.6676 ± 0.0017 |
| DAdam | AER | 27.58 ± 0.06 / 0.5440 ± 0.0013 | 45.60 ± 0.12 / 0.6664 ± 0.0014 |
| | Complete | 27.60 ± 0.18 / 0.5445 ± 0.0021 | 45.65 ± 0.22 / 0.6670 ± 0.0017 |
| AccumAdam | AER | 27.71 ± 0.09 / **0.5481 ± 0.0027** | 46.05 ± 0.03 / **0.6715 ± 0.0029** |
| | Complete | **27.80 ± 0.34** / 0.5474 ± 0.0027 | **46.18 ± 0.20** / 0.6704 ± 0.0017 |

Table 2: **Neural Machine Translation on WMT14 (Bojar et al., 2014)**: The results show the generalization performance of training transformer (base) on English-to-German and transformer (big) on English-to-French. The last checkpoint of each method is evaluated by BLEU score (Papineni et al., 2002) and METEOR (Banerjee & Lavie, 2005) the corresponding newtest2014 testset. The number of workers is 16 and the number of workers per node is 4 for all experiments. The error bands with $\pm 2\sigma$ based on 3 runs are reported.

| | Top-1 Acc. ↑ / Top-5 Acc. ↑ / Train Loss ↓ | |
|---|---|---|
| Methods | Alternating Exp Ring | Complete |
| AllReduce | 76.01 / 93.05 / 1.864 | |
| DAdam | 75.31 / 92.54 / 1.998 | 75.27 / 92.60 / 1.993 |
| AccumAdam | **76.54 / 93.18 / 1.836** | **76.34 / 93.15 / 1.835** |

Table 3: **Image Classification on ImageNet-1K (Deng et al., 2009)**: The results are show the generalization of training ResNet-50 on ImageNet-1K for image classification. Top-1 and Top-5 accuracies and averaged training loss in the last epoch are reported, and the accuracies are evaluated on the ImageNet validation set. The number of workers is 32 and the number of workers per node is 8 for all experiments.

## 5.2 PER-ITERATION RUNTIME AND SCALABILITY

We will now demonstrate the practicality of decentralized training by showing its superior runtime compared to All-Reduce training in various scenarios, including communication-bound, straggler-bound, and computation-bound cases. For a fair comparison, we report strong scaling performance in Figure 6. Other works, such as Assran et al. (2019); Wang et al. (2019), present weak scaling performance which keeps the worker batch size fixed. However, this results in fewer iterations and a larger effective global batch size for the same number of epochs, which alters the training dynamics.

As shown in Figure 6, the results align with the proposed runtime model: (1) a sparse communication topology enables more significant speedups in communication-limited scenarios compared to computation-limited scenarios; (2) decentralized training has better resilience to computation time variance; (3) decentralized training is more efficient than All-Reduce training in all tested scenarios. Additionally, we re-implement 1-OSGP (Assran et al., 2019) (the best asynchronous variant of SGP) with the one-peer exponential graph for comparison. Due to the extra intra-node broadcast required to synchronize the parameters, 1-OSGP may experience longer per-iteration runtimes when the intra-node connection is not significantly faster than the inter-node connection (nodes with 8×T4 GPU with 100Gbps InfiniBand connection, for example). Consistent with the analysis in § 2.3, our implementations outperform 1-OSGP in all setups, especially in computation-bound scenarios.

To demonstrate the relationship between generalization performance and runtime, we present training curves for an En-De translation task in Figure 7, showing a $\sim 60\%$ speedup at the same validation loss.

We also conducted experiments on the pre-training of GPT-2 (small) (Radford et al., 2019). Compared to All-Reduce training, decentralized training with AccumAdam and the Alternating Exponential Ring graph achieves better training loss (2.8419 vs. 2.8468), validation loss (2.8561 vs. 2.8626), and runtime (30.41 hours vs. 32.461 hours) after 600k training steps on OpenWebText (Gokaslan et al., 2019) using 4×4×A40 GPUs. Even though the workloads are balanced among the workers, it

still achieves a relative speedup of 6.31%. More speedup can be expected by replacing the sequence packing with other batching strategies that may introduce imbalance in workloads.

Due to space limitations, we defer the presentation of other speedup results to § A.8 in the appendix.

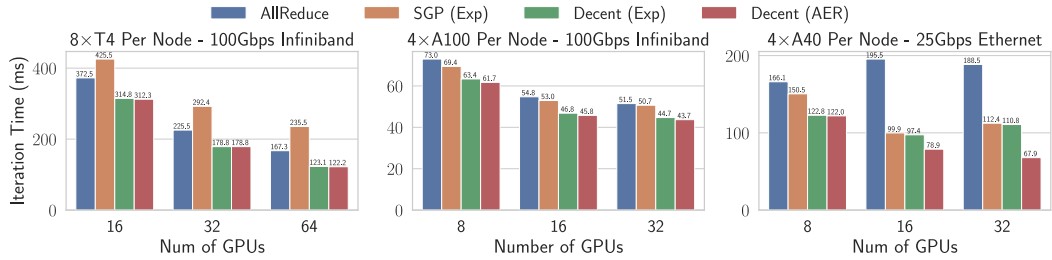

Figure 6: The results show the strong scaling performance (fixed global batch size, 25K tokens) of All-Reduce, SGP (Assran et al., 2019) and our implementations by reporting the per-iteration runtime with various numbers of workers and network settings. Here, `Exp` and `AER` stand for one-peer exponential graph and alternating exponential ring, respectively.

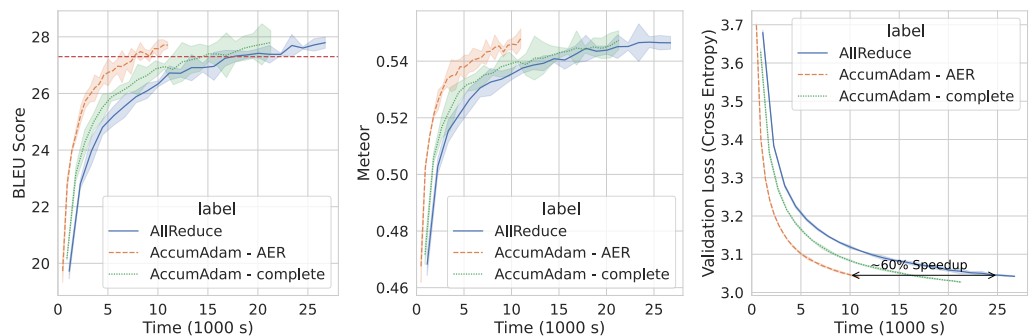

Figure 7: Training a transformer (base) model on the English-to-German translation task on four 4xA40 nodes with 25Gbps Ethernet connections. The error bands of $\pm 2\sigma$ are based on three runs. The red dashed line on the left is the baseline (27.3 BLEU score) from Vaswani et al. (2017).

## 6 CONCLUSIONS

This paper has aimed to address practical and sometimes overlooked aspects of decentralized training, to make it superior to All-Reduce in terms of total training time and generalization performance in practical settings. We identified key sources of speed-up in decentralized training of deep neural networks and developed a simple model to optimize per-iteration runtimes. We introduced a decentralized Adam algorithm that supports overlapping communication and computation and proposed an accumulation mechanism to mitigate the high variance that occurs in small local batch sizes. Numerous deployments of our methods in practical environments validated their effectiveness, achieving a faster training speed than standard All-Reduce training without compromising generalization.

## 7 ACKNOWLEDGMENT

The computation and storage resources were enabled by resources provided by the National Academic Infrastructure for Supercomputing in Sweden (NAISS), partially funded by the Swedish Research Council through grant agreement no. 2022-06725. This work was partially supported by the Wallenberg AI, Autonomous Systems and Software Program (WASP) funded by the Knut and Alice Wallenberg Foundation.

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

# A APPENDIX / SUPPLEMENTARY MATERIAL

## A.1 ALL-REDUCE ADAM IN THE COMPARISON OF EXPERIMENTS

The All-Reduce Adam is shown as Algorithm 2. In contrast to the decentralized algorithms in Section 4, it averages gradients instead of models and uses All-Reduce instead of gossip communication.

---

**Algorithm 2** All-Reduce Adam on worker $i$

---

1: $m_i^{(0)}, v_i^{(0)} \leftarrow 0 \; ; \; x_i^{(0)} \leftarrow x^{(0)}$
2: **for** $t = 1, 2, \ldots, T$ **do**
3:      $g_i^{(t)} \leftarrow \nabla \ell \left( x_i^{(t-1)}; \xi_i^{(t)} \right)$
4:      $\bar{g}^{(t)} \leftarrow \frac{1}{N} \sum_{i=1}^N g_i^{(t)}$    (All-Reduce)
5:      $m_i^{(t)} \leftarrow \beta_1 m_i^{(t-1)} + (1 - \beta_1) \bar{g}^{(t)}$
6:      $v_i^{(t)} \leftarrow \beta_2 v_i^{(t-1)} + (1 - \beta_2) \left[ \bar{g}^{(t)} \right]^2$
7:      $x_i^{(t+1)} \leftarrow -\alpha \frac{m_i^{(t)}/(1-\beta_1^t)}{\sqrt{v_i^{(t)}/(1-\beta_2^t)+\epsilon}} + x_i^{(t)}$
8: **end for**

---

## A.2 HETEROGENEOUS COMMUNICATIONS IN HPC CLUSTERS

Although there is no single HPC architecture, a typical HPC cluster with many GPU nodes includes high-performance CPUs and GPUs, high-speed interconnects, scalable storage solutions, and management and software stack to handle various computational tasks. Figure 8 shows the typical topology that we have encountered in our experiments on different clusters. Here, the GPUs within a node communicate with high bandwidth and low latency, while communication through the switch is much more limited. While Figure 8 shows one example of cluster topology, there are many other possible topologies such as (1) fast NVLinks inter-connecting intra-node GPUs (2) star, ring, tree, mesh, or hybrid connection topologies that link nodes together as a cluster. Different topologies impose different limitations on the communication between GPUs in the same and in different nodes.

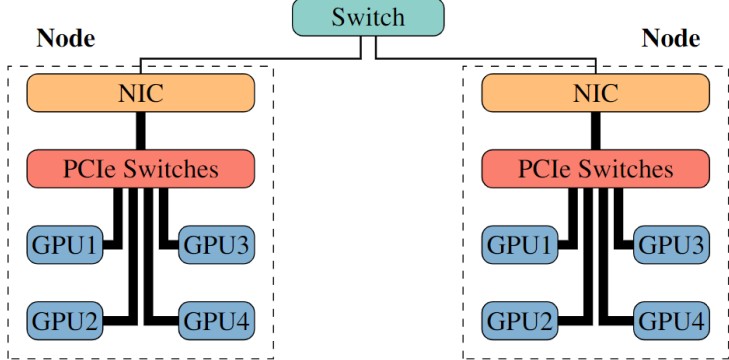

Figure 8: In this HPC environment, each of the two **nodes** comprises four **workers** interconnected via PCIe (Peripheral Component Interconnect Express) switches and Network Interface Cards (NICs). The nodes are linked to each other through a switch. We use line thickness to indicate the bandwidth of these connections: wider lines represent higher bandwidth, facilitating faster communication within nodes, while thinner lines depict the narrower bandwidth of the switch, resulting in more time-consuming inter-node communication.

## A.3 TIMELINES AND DEPENDENCY GRAPHS OF DIFFERENT DATA PARALLEL SCHEMES

To better illustrate the capability of our scheme in overlapping communication and computation, we consider two comparison schemes, including the vanilla scheme and a well-optimized scheme that is also used in modern distributed training frameworks like PyTorch's DataDistributedParallel (Paszke

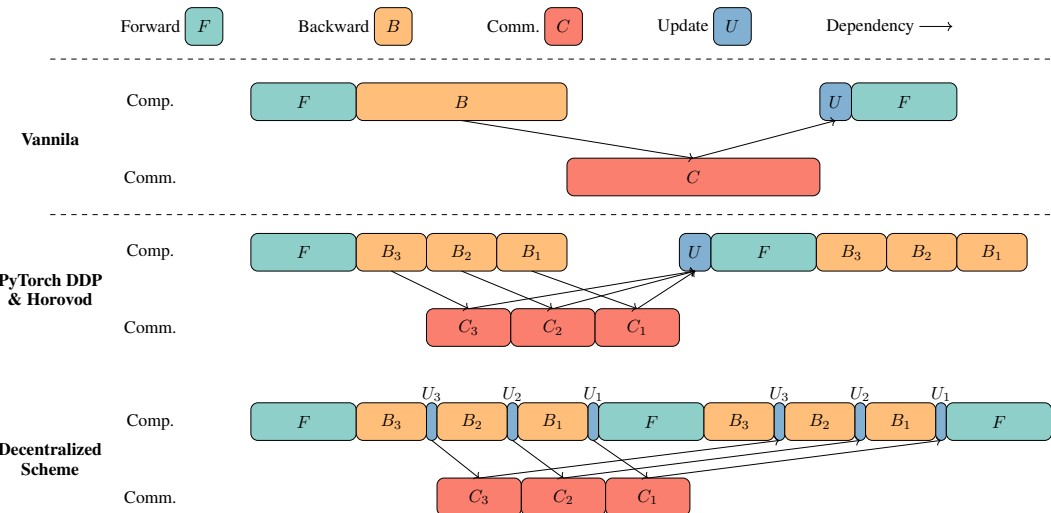

Figure 9: Timelines and dependency relations of decentralized training. $F$: forward pass, $B$: backward pass, $C$: communication/aggregation of gradients or model parameters, $U$: update model parameters. Arrow from $X$ to $Y$ means that $Y$ can start only after $X$ finishes.

et al., 2017) and Horovod (Sergeev & Balso, 2018). We visualize the communication&computation of all three schemes in Figure 9. In the vanilla scheme, a communication operation only begins to aggregate the gradients among all the workers after completing both a forward pass and a backward pass. The update of the model only happens after the aggregation ends, and the computation does not overlap with the communication. Similar to our scheme, in the PyTorch DDP & Horovod scheme, the model parameters are divided into buckets (the parameters are divided into 3 in the example in Figure 9). The communication of the bucket can start immediately after its corresponding backward pass finishes. In this scheme, some time of the communication interleaves with the computation, which greatly improves the utilization. However, the communication still can not be interleaved with the computation of the forward pass and the last bucket of the backward pass. In contrast to the two comparison schemes, our decentralized scheme hides all communications by computations.

## A.4 DECENTRALIZED COMMUNICATION TOPOLOGIES

We display the One-peer Ring (Figure 10), the One-peer Exponential (Figure 11), and the Alternating Exponential Ring (Figure 12) topologies in this subsection.

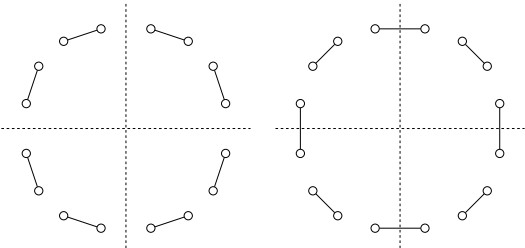

Figure 10: One-peer Ring for a 16-worker example where workers are distributed in four nodes. Dashed lines are the boundaries of the compute nodes.

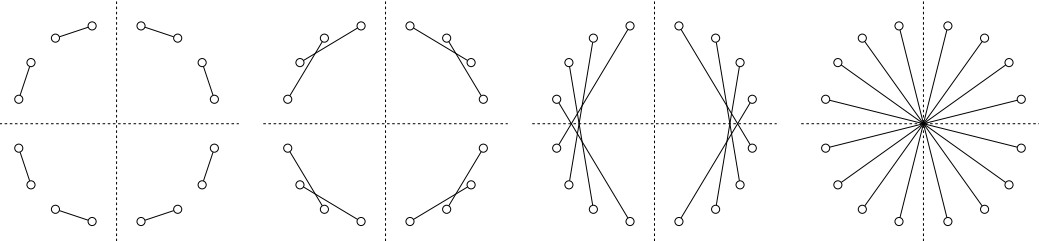

Figure 11: One-peer Exponential graph for a 16-worker example where workers are distributed in four nodes. Dashed lines are the boundaries of the compute nodes.

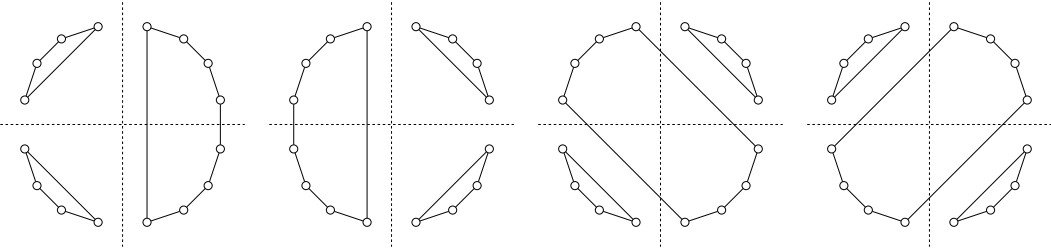

Figure 12: Alternating Exponential Ring topology (ours) for a 16-worker example where workers are distributed in four nodes. Dashed lines are the boundaries of nodes. The loops mean All-Reduce across workers in the loop.

### A.5 RUNTIME MODEL

Our runtime model is constructed under some reasonable settings and experimental results. It assumes that there are $N$ workers and that the neural network model parameters are divided into $b$ buckets so that the average computation and communication time of each bucket are all the same. The global batch size is fixed and the time needed to process the forward pass on a single worker takes 1 unit time. According to Kaplan et al. (2020) and our profiling results, the time taken by the backward pass is roughly twice as the forward pass. With the workload evenly distributed on $N$ workers, the time taken by the forward pass is then $1/N$ unit time and the time taken by the backward pass is $2/N$ unit time. We also assume the time that it takes to update one bucket is $\theta$ time units where $\theta$ is the same for All-Reduce training and decentralized training and independent of the number of workers. Three additional factors are introduced to characterize the system setup and workload:

1. $\gamma \in (0, \infty)$ is the time of an All-Reduce communication of one bucket. Note that $\gamma$ is the actual communication time, while the actual time of the All-Reduce operation could be longer due to synchronization across workers. The $\gamma$ parameter is influenced by several factors:

   (a) Network technology. A poor network connection leads to a slower All-Reduce and a larger $\gamma$.
   (b) The number of workers. More workers lead to a slower All-Reduce and a higher value of $\gamma$.

2. $\omega \in (0, 1]$ is the ratio of the time of a decentralized communication round and an All-Reduce operation. Note that $\omega$ could exceed one, but in that case it would always be better to use All-Reduce (as discussed in Section 2.2). Using a more sparse communication topology could potentially decrease $\omega$.

3. $\sigma^2 \in [0, \infty)$ is the variance in the truncated normal distributions used to model the variance in the computation time. For worker $i$ at iteration $t$, a random number $p_i^{(t)}$ is sampled from the truncated normal distribution with mean 1, variance $\sigma^2$, and support $[0.5, 1.5]$ (chosen to match our experimental results in Figure 3). The random number is used as a multiplier for the computation time of worker $i$ at iteration $t$, *i.e.* the time for both forward and backward passes are multiplied by the same $p^{(i,t)}$.

With these quantities, we further take the dependency relationship into consideration. We denote the completion time of task $X$ at worker $i$ in iteration $t$ by $T_X^{(i,t)}$. Then, based on the dependency relation

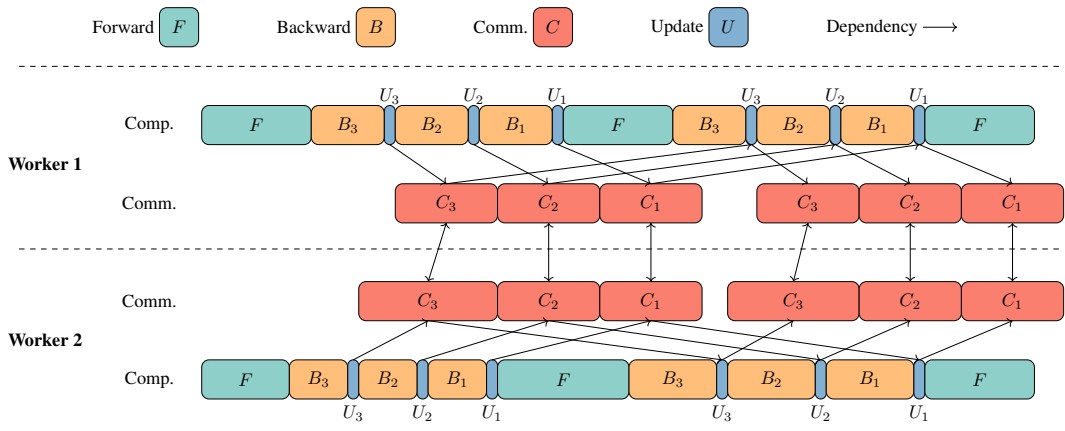

Figure 13: Timelines of two workers in decentralized training. In the example, worker 1 has uniform computation time across two iterations while worker 2 has a faster iteration and then a slower one. In the first iteration, $C_3$ of worker 2 is prolonged because it needs to synchronize with worker 1 to start the communication.

of All-Reduce training, we have

$$
\begin{aligned}
T_F^{(i,t)} &= T_U^{(i,t-1)} + p^{(i,t)} \frac{b}{N} \\
T_{B_k}^{(i,t)} &= \begin{cases} p^{(i,t)} \frac{2}{N} + T_F^{(i,t)} & \text{if } k = b \\ p^{(i,t)} \frac{2}{N} + T_{B_{k+1}}^{(i,t)} & \text{if } k \in \{1, \ldots, b-1\} \end{cases} \\
T_{C_k}^{(i,t)} &= \begin{cases} \gamma + \max_{j \in [N]} \{T_{B_k}^{(j,t)}\} & \text{if } k = b \\ \gamma + \max_{j \in [N]} \{T_{B_k}^{(j,t)}, T_{C_{k+1}}^{(j,t)}\} & \text{if } k \in \{1, \ldots, b-1\} \end{cases} \\
T_U^{(i,t)} &= T_{C_1}^{(i,t)} + \theta b
\end{aligned}
\tag{6}
$$

where all out-of-bound finish times like $T_U^{(i,0)}$ are defined to be zero.

To state similar dependency relations for decentralized training, we denote the set of neighbors of worker $i$ in iteration $t$ as $\mathcal{N}_i^{(t)}$. Then, we have

$$
\begin{aligned}
T_F^{(i,t)} &= T_{U_1}^{(i,t-1)} + p^{(i,t)} \frac{1}{N} \\
T_{B_k}^{(i,t)} &= \begin{cases} p^{(i,t)} \frac{2}{N} + T_F^{(i,t)} & \text{if } k = b \\ p^{(i,t)} \frac{2}{N} + T_{U_{k+1}}^{(i,t)} & \text{if } k \in \{1, \ldots, b-1\} \end{cases} \\
T_{U_k}^{(i,t)} &= \max\{T_{B_k}^{(i,t)}, T_{C_k}^{(i,t-1)}\} + \theta \\
T_{C_k}^{(i,t)} &= \begin{cases} \omega \gamma + \max_{j \in \mathcal{N}_i^{(t)}} \{T_{U_k}^{(j,t)}, T_{C_1}^{(j,t-1)}\} & \text{if } k = b \\ \omega \gamma + \max_{j \in \mathcal{N}_i^{(t)}} \{T_{U_k}^{(j,t)}, T_{C_{k+1}}^{(j,t)}\} & \text{if } k \in \{1, \ldots, b-1\} \end{cases}
\end{aligned}
\tag{7}
$$

where all out-of-bound finish times like $T_{U_1}^{(i,0)}$ and $T_{C_k}^{(i,0)}$ are defined as zeros.

When $\sigma^2 > 0$, we need to simulate the model and perform Monte Carlo simulations to draw conclusions. When the workloads are deterministic, on the other hand, the model can be used to derive closed-form expressions to guide the design; see Section 3 in the main document.

### A.5.1 SEQUENTIAL TIMELINE ON THE MAIN THREAD

Algorithm 3 describes the main thread of one worker and relates the different steps to the sequential timeline in Section A.5 of the appendix. In Algorithm 3 below, $F^{(i,t)}$ stands for the forward pass of worker $i$ at iteration $t$, $B_k^{(i,t)}$ stands for the backward pass on bucket $k$ of worker $i$ at iteration $t$,

$U_k^{(i,t)}$ stands for the optimizer update on bucket $k$ of worker $i$ at iteration $t$, and $C_k^{(i,t)}$ stands for the decentralized communication of bucket $k$ of worker $i$ at iteration $t$.

---

**Algorithm 3** Decentralized Training on worker $i$

---

1: **for** $t \leftarrow 1, 2, \cdots, T$ **do**
2:    Perform forward pass: $F^{(i,t)}$                                   $\triangleright$ Time taken: $p^{(i,t)} \frac{b}{N}$
3:    **for** $k \leftarrow b, b-1, \cdots, 1$ **do**
4:       Perform backward for $k$-th bucket: $B_k^{(i,t)}$                 $\triangleright$ Time taken: $p^{(i,t)} \frac{2}{N}$
5:       Wait for the comm. op. $(C_k^{(i,t-1)})$ in the last iteration.    $\triangleright$ Time taken: $T_{U_k}^{(i,t)} - T_{B_k}^{(i,t)}$
6:       Update the parameters in $k$-th bucket: $U_k^{(i,t)}$             $\triangleright$ Time taken: $\theta$
7:       Launch comm. op. for $k$-th bucket asynchronously: $C_k^{(i,t)}$  $\triangleright$ Time taken: $\sim 0$
   (non-blocking)
8:    **end for**
9: **end for**

---

### A.5.2 INFLUENCE OF COMMUNICATION TOPOLOGY ON RUNTIME

To demonstrate the influence of the communication topology on the runtime, we conduct simulations under various topologies but keep $\omega = 1$, which can exclude the impact of communication topology on reduced communication cost. The simulation result is in Figure 14. Compared with the speedup by adapting decentralized training and reduced $\omega$, the simulation result shows that the changes in the event dependencies caused by the communication topology do not have a significant influence. The difference becomes even more insignificant when the communication cost is the bottleneck ($N\gamma > 3.3$).

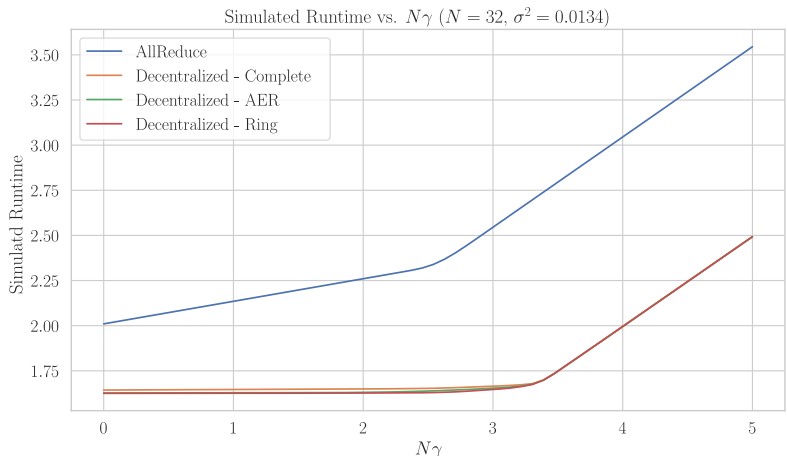

Figure 14: Simulated runtime under various communication topologies. The parameters are $b = 4$, $\theta = 0.02$, and the number of workers in one node is 4.

A.6  TASKS AND DETAILED EXPERIMENT SETUPS

All of our experiment implementation is based on PyTorch (Li et al., 2020). As for the implementation of All-Reduce training baselines, we use *DistributedDataParallel* (DDP) module. In all our experiments, we use full-precision training and activate tensor cores for devices that support the feature.

**Neural machine translation.** Following (Vaswani et al., 2017), we trained the base transformer ($\sim$65M parameters) on the WMT14 English-German dataset (Bojar et al., 2014) using the global batch size which is roughly 25K source and target tokens, and the models were trained for $\sim$130,000 steps (or 24 epochs) in total. We trained the big transformer ($\sim$213M parameters) on the WMT14 English-French dataset using the same batch size, and the models were trained for $\sim$300,000 steps (or 5 epochs) in total. The All-Reduce training baselines use Adam optimizer (Kingma & Ba, 2014). In each run, the last checkpoint is used to be evaluated by BLEU score (Papineni et al., 2002) on the corresponding test sets.

**Image classification.** As for the image classification task, we trained ResNet-50 (He et al., 2016) on ImageNet-1K ($\sim$1.28M images and 1K classes) (Deng et al., 2009). Similar to the translation task, All-Reduce training baselines use Adam optimizer. In all experiments, the models were trained for 90 epochs with 1024 as the global batch size. In each run, the last checkpoint is used to be evaluated by top-1 and top-5 accuracies.

Note that, for all experiments, we fix the global batch sizes of tasks. Unlike the experiments in (Lian et al., 2017) and (Assran et al., 2019) which keep the local batch sizes fixed, we argue that using fixed global batch sizes yields a more fair comparison with All-Reduce training because (1) increasing the global batch size as scaling up the number of workers may impose negative effects on the generalization performance of All-Reduce training (Keskar et al., 2017), (2) increasing the global batch size but keeping the total number of epochs fixed leads to speedup in runtime because of the fewer number of updates, and (3) it's important to have same number of iterations to achieve comparable performance under fixed epoch budget.

A.6.1  NEURAL MACHINE TRANSLATION

As for the model architecture, we use the transformer implementation provided by PyTorch. We adopt the scripts open-sourced by Fairseq[5] to prepare and clean the raw data.

**Transformer(base) on WMT14 En-De**  Following the experiments in (Vaswani et al., 2017), the sentence pairs are encoded using the byte-pair encoder tokenizer, which has a shared vocabulary with size 37056. The global batch size is fixed to around 25k tokens for source tokens and target tokens, respectively. The sentence pairs with similar lengths are batched together in the training. All experiments use $0.1$ as the dropout rate and $0.1$ as the label smoothing. The learning rate schedule is the inversed square root with 1 epoch of linear warmup. In the experiments of All-Reduce baselines, the learning rate is 0.0007, and the betas for the Adam optimizer are (0.9, 0.98), which are taken from Vaswani et al. (2017). In the experiments of decentralized Adam, the betas for the Adam optimizer are set to (0.974, 0.999) where the choice of $\beta_1$ is inspired by AccumAdam when setting $s = 4$ (defined in Algorithm 4) and the choice of $\beta_2$ is from Devlin et al. (2018). The choice of betas empirically shows much better performance than (0.9, 0.98), so we use the setting for all experiments of decentralized Adam. The base learning rates are 0.0020, 0.0028, and 0.0038 for 8, 16, and 32 workers, respectively. In the experiments of AccumAdam, the betas for the AccumAdam optimizer are (0.9, 0.999), and the number of accumulation iterations $s$ is $N/4$. The base learning rate is 0.0008 for 8 workers and 0.0012 for 16 and 32 workers.

**Transformer(big) on WMT14 En-Fr**  Following the experiments in Vaswani et al. (2017), the sentence pairs are encoded using the word-piece encoder tokenizer, which has a shared vocabulary with size 32000. The global batch size is fixed to around 25k tokens for source tokens and target tokens, respectively. The sentence pairs with similar lengths are batched together in the training. All experiments use $0.1$ as the dropout rate and $0.1$ as the label smoothing. The learning rate schedule is the inversed square root with 1 epoch of linear warmup. In the experiments of All-Reduce baselines,

---

[5]https://github.com/facebookresearch/fairseq/tree/main/examples/translation

the learning rate is 0.0007, and the betas for the Adam optimizer are (0.9, 0.98) (Vaswani et al., 2017). In the experiments of decentralized Adam, the betas for the Adam optimizer are set to (0.974, 0.999). The base learning rate is 0.0012 for 8 workers. In the experiments of AccumAdam, the betas for the AccumAdam optimizer are (0.9, 0.999), and the number of accumulation iterations $s$ is $N/4$. The base learning rate is 0.0008 for 8 workers.

### A.6.2 IMAGE CLASSIFICATION

**ResNet50 on ImageNet**  For the experiment setup of the image classification task, we follow the training recipe used to train the weights by torchvision[6]. To avoid the bottleneck caused by the data loading and preprocessing on CPU, we use DALI[7] for the pipeline of preprocessing. In the preprocessing and the data augmentation, the images are: (1) randomly cropped (aspect ratio is from 3/4 to 4/3, random area is from 0.08 to 1.0) (2) resized to 224×224 (3) randomly flipped horizontally with 0.5 probability (4) normalized by channel means and variances. In all experiments, the global batch size is fixed to 1024, the learning rate schedule is cosine annealing with 5 epochs of linear warmup, the weight decay is $3 \times 10^{-5}$ on non-batch-normalization parameters, and the label smoothing is 0.1. In the experiments of All-Reduce baselines, the optimizer is Adam, the base learning rate is 0.001, and the betas are (0.9, 0.999). In the experiments of decentralized Adam, the base learning rate is 0.0024, 0.0048, and 0.0070 for 8, 16, and 32 workers, respectively. The betas are (0.974, 0.999). In the experiments of AccumAdam, the base learning rate is 0.002 for all numbers of workers, the betas are (0.9, 0.999), and $s$ is set to $N/4$.

### A.6.3 GPT-2 PRETRAINING

**GPT-2 (small) on OpenWebText**  For the experiment setup of the LLM pretraining, we follow the training recipe from nanoGPT[8]. The batching strategy used in the repository is sequence packing, and the global batch size is around 0.5 million tokens. The hyperparameters of the baseline are the same as in the repository. The baseline uses AdamW optimizer. For the AccumAdam experiments with 16 A100 GPUs, we only change $\beta_2$ and $s$ to 0.999 and 8, respectively, and keep the other hyperparameters the same as the baseline.

### A.7 PYTORCH EXTENSION FOR DECENTRALIZED TRAINING

The implementation of the PyTorch extension facilitates decentralized training (with the same scheme as in Eq. 2). The design is inspired by the `DistributedDataParallel(DDP)` module in PyTorch (Li et al., 2020).

The main idea of the implementation is as follows. The implementation is a wrapper of the PyTorch models (`torch.nn.Module`). By using the hook mechanism in PyTorch, we register the backward hook for all weights before the first forward and backward passes. Then, in the first backward pass, the wrapper collects the computation order of weights, groups the weights into buckets by their order, removes the hooks, and registers new hooks on the last weight in the buckets using `register_post_accumulate_grad_hook` (available from PyTorch 2.1). The `post_accumulate_grad_hook` indicates the event that the corresponding gradient of one weight is computed and accumulated, which means the weight will no longer be used in this iteration and it's ready for update. In the hook, the wrapper checks if the decentralized communication for the bucket in the last iteration is finished, updates the weights in the bucket, and launches the decentralized communication for the next iteration. Another module is implemented for scheduling the neighbors of workers in each iteration, and it makes the extension flexible for various topologies.

Subject to the complexity of implementation and limited time, the implemented wrapper only supports one GPU per worker, and it does not support automatic mixed precision and sparse gradients like in `DDP`, but the improvements are planned for wider applications in the future.

The implementation is based on the Python API provided by PyTorch. We believe that further practical speedups could be achieved by modifying the low-level implementation of the communication to

---

[6]https://pytorch.org/blog/how-to-train-state-of-the-art-models-using-torchvision-latest-primitives/
[7]https://github.com/NVIDIA/DALI
[8]https://github.com/karpathy/nanoGPT

optimize it for gossip communication. However, for a fair comparison with `DDP` in PyTorch, our implementation is at the appropriate level of optimization.

The code bases for the PyTorch extension and the experiments are open-source at https://github.com/WangZesen/Decent-DP and https://github.com/WangZesen/Decentralized-Training-Exp, respectively.

### A.7.1 COMPARISON WITH IMPLEMENTATION IN STOCHASTIC GRADIENT PUSH

In the implementation of stochastic gradient push (SGP) (Assran et al., 2019), to utilize the fast connections (NVLinks between GPUs in one node), it groups the GPUs in one node. In each iteration, the workers in one node will aggregate the gradients and update the model shared in the node combining with the received communication messages. Therefore, the communication topology discussed in Assran et al. (2019) is on the node level. One of the advantages is that the design of topology does not involve the significant heterogeneity caused by intra-node and inter-node connections. However, heterogeneity in connections still exists because the characteristics of inter-node connections may differ depending on the cluster topology and congestion caused by concurrent communications.

Compared with our design, it's different in that workers always exchange the model parameters instead of gradients, and our communication topology is defined on the worker level. As for the topology, we could utilize the connections in the same way as SGP's implementation as the worker-level topology is a superset of node-level topology. Moreover, we argue that our implementation is more advantageous in the resilience to the variance because aggregating the gradients will cause the straggler problem within the node.

We demonstrate the difference by making one change in the dependency relationship of the update task. In SGP's implementation, the update on one bucket does not only depend on the corresponding backward and the communication from the last iteration of the work itself but also its local neighbors' backward pass and communication. Therefore, compared with the runtime model of decentralized training (Eq. 7), the finish time of $U_k$ is changed to

$$T_{U_k}^{(i,t)} = \max_{j \in \mathcal{L}(i)} \{T_{B_k}^{(j,t)}, T_{C_k}^{(j,t-1)}\} + \theta \tag{8}$$

where $\mathcal{L}(i)$ is the workers that are in the same node as worker $i$.

Note that in this simplified model, we ignore the time taken by the head worker to broadcast gossip updates to other workers in the node. In most cases, the operation does not incur significant cost to the per-iteration runtime, but only when the intra-node connection is comparable with the inter-node connection (the T4 GPU setup in § 5.2, for example).

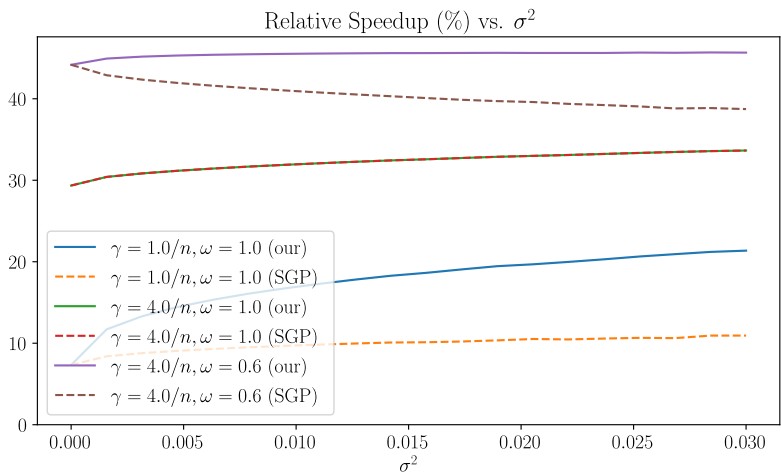

Figure 15: Comparison with Stochastic Gradient Push

The results of the simulation by the runtime models are shown in Figure 15. It shows that, with increasing variance in computation time, our implementation has better speedup (hence better resilience to computation time variance) than SGP when it's computation-bound. In the communication-bound case, since the communication time dominates the runtime, both implementations have the same speedup.

Since the dependency libraries of the open-source code of SGP[9] have been deprecated due to the fast updates of dependencies, we can not reproduce the runtime results on our hardware. Instead, we re-implemented 1-OSGP (the best asynchronous version of SGP) in our numerical experiments. The relative comparisons between 1-OSGP and All-Reduce training are as similar as in Assran et al. (2019). The comparisons of runtime performance in § 5.2 justify the advantages of our implementation.

---

[9]https://github.com/facebookresearch/stochastic_gradient_push

## A.8    ADDITIONAL EXPERIMENTS

The section presents the experiment results that are not shown in the main text due to space limitation.

### A.8.1    GENERALIZATION PERFORMANCE

Table 4 shows the generalization performance of the training transformer (big) on WMT14 English-to-French, and it shows that both DAdam and AccumAdam outperform All-Reduce training baseline, and AccumAdam outperforms DAdam in most of the metrics.

Table 4: **Neural Machine Translation on WMT14 (Bojar et al., 2014)**: The results show the generalization performance of training transformer (base) on English-to-German and transformer (big) on English-to-French. The last checkpoint of each method is evaluated by BLEU score (Papineni et al., 2002) and METEOR (Banerjee & Lavie, 2005) the corresponding newtest2014 testset. The number of workers is 8 and the number of workers per node is 4 for all experiments. The error bands with $\pm 2\sigma$ based on 3 runs are reported. Some entries are empty because of limited resources.

| Method | Topology | En-De / Transformer (base) | En-Fr / Transformer (big) |
|---|---|---|---|
| AllReduce | Complete | $27.72 \pm 0.37$ / $0.5476 \pm 0.0036$ | $45.69 \pm 0.30$ / $0.6676 \pm 0.0017$ |
| | Ring | $27.59 \pm 0.61$ / $0.5452 \pm 0.0054$ | - |
| DAdam | AER | $27.83 \pm 0.11$ / $0.5472 \pm 0.0018$ | - |
| | Complete | $27.84 \pm 0.59$ / $0.5467 \pm 0.0062$ | $45.90 \pm 0.32$ / $0.6693 \pm 0.0023$ |
| | Ring | $27.61 \pm 0.20$ / $0.5447 \pm 0.0014$ | - |
| AccumAdam | AER | $27.78 \pm 0.41$ / $0.5471 \pm 0.0020$ | - |
| | Complete | $27.82 \pm 0.13$ / $0.5473 \pm 0.0015$ | $46.20 \pm 0.22$ / $0.6718 \pm 0.0009$ |

### A.8.2    SPEEDUP

In Tables 5, we present the per-iteration runtime of both All-Reduce training and decentralized training (both with Adam optimizer) and the relative speedup of decentralized training over All-Reduce training on the image classification task. Since we conducted the experiments with a fixed iteration budget, the relative speedup of per-iteration runtime also reflects the relative speedup of total runtime.

Combined with the results of generalization performance, the experiments demonstrate the practicality of decentralized training.

Table 5: Results of speedup of training ResNet-50 on ImageNet dataset for the image classification task.

| Node Setup / Inter-Connection | Workers | Methods | Comm. Topology | Per-iteration Runtime (ms) | Speedup |
|---|---|---|---|---|---|
| 4×A40 Node / 25Gbps Ether. | 8 | AllReduce | complete | 344.44 | - |
| | | Decentralized | complete | 333.07 | 3.30% |
| | 16 | AllReduce | complete | 176.72 | - |
| | | Decentralized | complete | 163.68 | 7.38% |
| 8×T4 Node / 100Gbps Infini. | 16 | AllReduce | complete | 645.59 | - |
| | | Decentralized | complete | 617.75 | 4.31% |
| | 32 | AllReduce | complete | 331.51 | - |
| | | Decentralized | complete | 313.56 | 5.41% |

**Scalability Experiments with FFCV**    Additional experiments were performed by integrating our implementation of decentralized training with FFCV Leclerc et al. (2023). The code is adapted from the official repository of FFCV for fast training on ImageNet[10]. The following are some important details about the environment used for these tests:

---

[10]https://github.com/libffcv/ffcv-imagenet/tree/main

- Key software: PyTorch 2.5.1, CUDA 12.1, ffcv 1.0.2.

- Hardware configuration: 16 CPU cores[11] and 64 GB RAM per GPU. For nodes with 4×A100[12] each, the nodes are interconnected with 100Gbps InfiniBand. For nodes with 4×A40[13] each, the nodes are interconnected with 25Gbps Ethernet.

- Preprocessing: All images were converted to `bento` format by FFCV and compressed with 384 pixels as the maximal size. The size of the processed training set is ∼40GB (same as the statistics from ffcv[14]), which can be fully loaded into memory for best data loading performance.

- Data loader: 12 workers for each GPU, which is the default setting.

- Network architecture: ResNet-50.

The input images are resized to the size of 160×160 pixels (default input size at the beginning of the training). We measured the per-iteration and per-epoch runtime for 1, 2, and 4 nodes with the same global batch size of 2048 using two types of nodes, respectively.

| | Iteration Time (ms) / Epoch Time (s) / Relative Speedup over AllReduce | | |
| Number of Nodes | 1 | 2 | 4 |
| --- | --- | --- | --- |
| AllReduce | 123.61 / 77.26 / 0.00% | 70.23 / 43.89 / 0.00% | 41.88 / 26.17 / 0.00% |
| DT – Complete | 123.00 / 76.88 / 0.49% | 67.98 / 42.49 / 3.20% | 40.57 / 25.36 / 3.13% |
| DT – AER | N/A | 67.07 / 41.92 / 4.49% | 38.97 / 24.36 / 6.94% |

Table 6: The table shows the runtime performance on nodes with 4×A100 each. The nodes are interconnected with 100Gbps InfiniBand. The statistics are generated by the mean of runtime of 3 epochs. The architecture of the neural network is ResNet-50, the global batch size is fixed to 2048, and the input image size is 160×160 pixels. The `AER` topology needs at least two nodes.

| | Iteration Time (ms) / Epoch Time (s) / Relative Speedup over AllReduce | | |
| # of Nodes | 1 | 2 | 4 |
| --- | --- | --- | --- |
| AllReduce | 274.73 / 171.70 / 0.00% | 151.06 / 94.41 / 0.00% | 121.36 / 75.85 / 0.00% |
| DT – Complete | 273.97 / 171.23 / 0.27% | 149.48 / 93.42 / 1.05% | 120.19 / 75.12 / 0.96% |
| DT – AER | N/A | 145.14 / 90.71 / 3.91% | 85.76 / 53.60 / 29.33% |

Table 7: The table shows the runtime performance on nodes with 4×A40 each. The nodes are interconnected with 25Gbps Ethernet. The statistics are generated by the mean of runtime of 3 epochs. The architecture of the neural network is ResNet-50, the global batch size is fixed to 2048, and the input image size is 160×160 pixels. The `AER` topology needs at least two nodes.

The results show that increasing the number of nodes can still enable significant speedups: for `AllReduce` with A100 nodes, the per-epoch runtime reduces from 76.876s with 1 node to 25.355s with 4 nodes. This demonstrates that the advantages of scaling up beyond 1 node remain even when we incorporate FFCV. Moreover, the decentralized training outperforms AllReduce training in all experiments even with `complete` topology. With Alternating Exponential Ring (`AER`), the decentralized training outperforms AllReduce training by a clear margin in runtime (up to 29.33% on four 4×A40 nodes).

---

[11]Intel(R) Xeon(R) Gold 6338 CPU @ 2GHz
[12]NVIDIA Tesla A100 HGX GPU with 40GB vRAM
[13]NVIDIA Tesla A40
[14]https://docs.ffcv.io/benchmarks.html

### A.8.3 TRAINING CURVES

**Training Curves for GPT-2 Pre-training** This section provides additional details about the experiments of pre-training GPT-2 (small) (Radford et al., 2019) on OpenWebText dataset (Gokaslan et al., 2019).

The curves in Fig. 16 and Fig. 17 demonstrate the cross-entropy loss for next-word prediction where the x-axes are the number of iterations and training time, respectively. The figures illustrate that decentralized training attains comparable generalization performance to AllReduce, but at a speedup of ∼6.31%.

The training details including datasets, source code, number of iterations, and training time were not revealed in the original paper (Radford et al., 2019). The code for the experiments was adapted from the open-source implementation of GPT-2[15] which, to the best of our knowledge, is the most comprehensive pre-training repository available. The hyperparameters of the AllReduce training baseline use the default settings in the repository (the batch size is ∼0.5 million tokens). In the repository, the training was conducted for 400k steps, which we extended to 600k to better validate the performance of decentralized training.

The experiments were conducted on 4 nodes with 4×A100 each. Since the workloads are balanced (sequence packing is used as the batching strategy) and the network is relatively fast (nodes are interconnected by 100Gbps InfiniBand), using sparser topologies will not significantly improve runtime for this case. Therefore, we used the `complete` topology in the experiment of decentralized training.

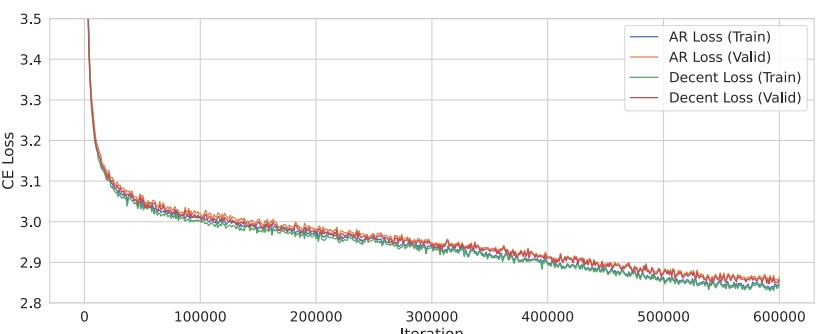

Figure 16: The curves of train and validation losses where the x-axis is the number of iterations. `AR`: AllReduce training. `Decent`: Decentralized training with AccumAdam.

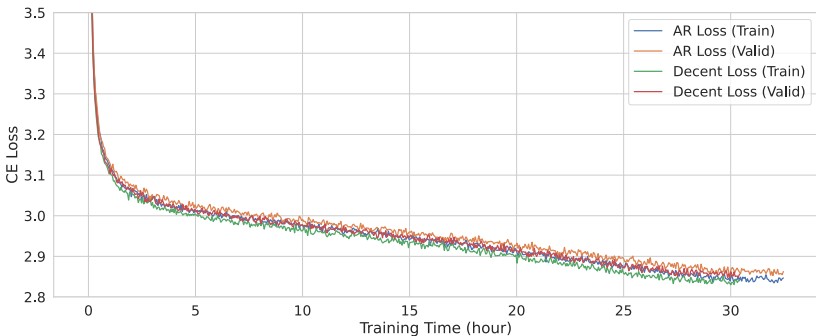

Figure 17: The curves of train and validation losses where the x-axis is training time. `AR`: AllReduce training. `Decent`: Decentralized training with AccumAdam.

---

[15]https://github.com/karpathy/nanoGPT

## B CONVERGENCE ANALYSIS

### B.1 PRELIMINARIES

#### B.1.1 NOTATION

Our convergence analysis uses the following, relatively standard, notation.

- $\| \cdot \|_2$ ($\| \cdot \|_\infty$): the $\ell_2$- ($\ell_\infty$-) norm.
- $N$: the number of workers.
- $D$: the number of dimensions of model parameters.
- $x_i^{(t)}, m_i^{(t)}, v_i^{(t)} \in \mathbb{R}^D$: the local model and the local momentums of worker $i$ at iteration $t$.
- $x_{i,d}^{(t)}, m_{i,d}^{(t)}, v_{i,d}^{(t)} \in \mathbb{R}$: the $d$-th entry of the local model and the local momentums of worker $i$ at iteration $t$.
- $F$, $F_i$: the global objective function and the local objective function of worker $i$.
- $g_i^{(t)} := \nabla f_i(x_i^{(t-1)}) := \nabla \ell(x_i^{(t-1)}; \xi_i^{(t)}) \in \mathbb{R}^D$: the stochastic gradient evaluated by the local model $x_i^{(t-1)}$ and the local sample $\xi_i^{(t)}$ on the loss function $\ell$.
- $\left[ g_i^{(t)} \right]^2$: the entry-wise square of vector $g_i^{(t)}$
- $\nabla_d f_i$: the $d$-th entry of $\nabla f_i$.
- $W = [w_{ij}] \in \mathbb{R}^{N \times N}$: the mixing matrix where $w_{ij}$ is the $ij$-th entry.
- $W^\infty \in \mathbb{R}^{N \times N}$: a matrix with all entries equal to $1/N$.
- $\mathbf{1}_N \in \mathbb{R}^{N \times N}$: an $N \times N$ matrix with all entries equal 1.
- $\mathcal{N}_i$: the set of neighbours of worker $i$ ($w_{ij} = w_{ji} > 0$).
- $[N] := \{1, 2, \ldots, N\}$
- $\mathbb{E}_{t-1}[\cdot]$: the conditional expectation given all stochastic gradients before the $t$-th iteration. By assumption, it holds that

$$\mathbb{E}_{t-1}\left[ g_i^{(t)} \right] = \nabla F_i(x_i^{(t-1)}).$$

### B.2 COMPARION WITH RELATED DECENTRALIZED METHODS

**Ability of overlap** To achieve the overlap of communication and computation, we use a combine-then-adapt (CTA) type of decentralized gradient descent (DGD) in Algorithm 1. In contrast, another type, adapt-then-combine (ATC) DGD in the form of $x_i^{(t)} \leftarrow \sum_{j \in \mathcal{N}_i^{(t)}} w_{ij}^{(t)}(x_j^{(t-1)} - \alpha^{(t)} d_j^{(t)})$ cannot straightforwardly achieve this overlap, even though it has been shown faster convergence in terms of communication rounds Cattivelli & Sayed (2009). Specifically, for worker $i$ to compute $x_i^{(t)}$, assume that a neighbor $j \in \mathcal{N}_i^{(t)}$ takes a very long time to calculate its direction $d_j^{(t)}$ and then take a long time to send $x_j^{(t-1)} - \alpha^{(t)} d_j^{(t)}$ to worker $i$. Worker $i$ must then wait for receiving $x_j^{(t-1)} - \alpha^{(t)} d_j^{(t)}$ from the straggler $j$ before it can compute $x_i^{(t)}$. In contrast, with the CTA type algorithm, worker $i$ only needs the previous iterate $x_j^{(t-1)}$ from the straggler $j$ which is already on the way, not the $d_j^{(t)}$ which is being calculated with $x_j^{(t-1)}$. This allows for the overlap of computation of $d_j^{(t)}$ and the communication of $x_j^{(t-1)}$. In other words, the communication task of the straggler worker $j$, $x_j^{(t-1)}$, is hidden within the computation of $d_j^{(t)}$.

**Comparison with state-of-art decentralized Adam-based methods** The authors in Nazari et al. (2022) proposed DADAM for online optimization over a decentralized network, with the algorithm based on the principle of CTA. Following CTA, the authors in Chen et al. (2023) introduced a unifying decentralized adaptive gradient framework, which involves communication not only of the

local models but also of the second-order momentums $v_i$. While this enhances the convergence accuracy of the algorithm, it also increases the communication cost. The authors in Ying et al. (2021b) proposed DAG-Adam, which incorporates consensus constraints as a penalty in the objective function. Based on the gradient of this penalized augmented objective function, the DAG-Adam algorithm is proposed, improving accuracy in non-iid data scenarios for decentralized Adam. Although DAG-Adam theoretically explains that using different $v_i$ across workers $i$ can lead to inexact convergence when data are non-iid, our algorithm is applied under the iid assumption, which holds for our HPC environments. Additionally, the paper Lin et al. (2021) introduces QG-ADAM which uses momentum technique to handle non-iid data. Since QG-ADAM uses the ATC approach, each worker must wait to receive all neighboring workers' local messages before updating its local model, thus limiting the ability to effectively overlap computation and communication.

### B.2.1 USEFUL INEQUALITIES

$$\forall a, b \in \mathbb{R}^d \text{ and } \forall \beta > 0, \quad \langle a, b \rangle \leq \frac{\beta}{2} \|a\|_2^2 + \frac{1}{2\beta} \|b\|_2^2. \tag{9}$$

$$\forall x_i \in \mathbb{R}^d \text{ and } n \in \mathbb{N}^+, \quad \left\| \sum_{i=1}^n x_i \right\|_2^2 \leq n \sum_{i=1}^n \|x_i\|_2^2. \tag{10}$$

$$\forall A \in \mathbb{R}^{n \times n} \text{ and } x \in \mathbb{R}^n, \quad \|Ax\|_2^2 \leq \lambda_{\max}^2 \|x\|_2^2, \tag{11}$$

where $\lambda_{\max}$ is largest singular value of $A$.

We first establish the theorem for $\beta_1 = 0$.

**Theorem B.1** (Without Heavy-ball Momentum). *Under Assumptions 1–3, for Algorithm 1, we have*

$$\frac{1}{T} \sum_{t=1}^T \left\| \nabla_d F(\bar{x}^{(t-1)}) \right\|_2^2 \leq \frac{4R}{\alpha T} \left[ F\left(\bar{x}^{(0)}\right) - F_* \right] + E\left[ \frac{1}{T} \ln \left( 1 + \frac{R^2}{\epsilon(1-\beta_2)} \right) - \ln(\beta_2) \right]$$

*where $\bar{x}^{(t)} = \frac{1}{N} \sum_{i=1}^N x_i^{(t)}$ and*

$$E := \frac{8DR^2}{\sqrt{1-\beta_2}} + \frac{2\alpha DLR}{(1-\beta_2)} + \frac{8\alpha^2(1+\lambda^2)L^2 DR}{(1-\lambda^2)^2(1-\beta_2)^{3/2}\sqrt{\epsilon}}.$$

The proof can be found in Appendix B.3. Compared with (Défossez et al., 2020, Theorem 2) for the single-machine Adam, Theorem B.1 shows that our decentralized Adam achieves the same results as Défossez et al. (2020) (ignoring scaled constants), except that there is an additional third term in $E$, which is related to the second largest absolute value eigenvalue of the mixing matrix $\lambda$. Since a $\lambda$ close to 1 indicates poor connectivity of the topology, it increases the third term in $E$. Notably, this term is proportional to the square of the step size, $\alpha$. By selecting a smaller $\alpha$, we can effectively mitigate this term.

### B.3 PROOF OF THOREM B.1: DECENTRALIZED ADAM WITHOUT HEAVY-BALL MOMENTUM

### B.3.1 ALGORITHM UPDATE UNDER ANALYSIS

We set $\beta_1 = 0$, which means that there is no heavy-ball momentum. We have the algorithm update under analysis is

$$g_i^{(t)} = \nabla \ell(x_i^{(t-1)}; \xi_i^{(t)}), \quad v_{i,d}^{(t)} = \beta_2 v_{i,d}^{(t-1)} + \left[ g_{i,d}^{(t)} \right]^2,$$

$$\alpha_t = \alpha \sqrt{\frac{1-\beta_2^t}{1-\beta_2}}, \quad x_{i,d}^{(t)} = -\alpha_t \frac{g_{i,d}^{(t)}}{\sqrt{\epsilon + v_{i,d}^{(t)}}} + \sum_{j \in \mathcal{N}_i} w_{ij} x_{j,d}^{(t-1)}. \tag{12}$$

Taking the average across workers and using the Assumption 3, we get that the (virtual) update of the globally averaged model is

$$\bar{x}_d^{(t)} = -\frac{\alpha_t}{N} \sum_{i=1}^N \frac{g_{i,d}^{(t)}}{\sqrt{\epsilon + v_{i,d}^{(t)}}} + \bar{x}_d^{(t-1)}. \tag{13}$$

We denote $\tilde{v}_i^{(t)} := \beta_2 v_i^{(t-1)} + \mathbb{E}_{t-1}\left[\left(g_{i,d}^{(t)}\right)^2\right]$ which differs from $v_{i,d}^{(t)}$ by replacing the last term of $v_{i,d}^{(t)}$ with its conditional expectation. To simplify the notation, we denote

$$\bar{G}_d^{(t)} := \nabla_d F\left(\bar{x}^{(t-1)}\right) = \frac{1}{N}\sum_{i=1}^{N} \nabla_d F_i\left(\bar{x}^{(t-1)}\right) \quad \text{and} \quad G_{i,d}^{(t)} := \nabla_d F_i(x_i^{(t-1)}). \tag{14}$$

### B.3.2 TECHNICAL LEMMAS

**Lemma 1** (adaptive update approximately follow a descent direction)  Under Assumption 2, for Algorithm 1, we have

$$\kappa \geq \frac{1}{4}\sum_{i=1}^{N}\frac{\left[\nabla_d F\left(\bar{x}^{(t-1)}\right)\right]^2}{\sqrt{\epsilon + \tilde{v}_{i,d}^{(t)}}} - \sum_{i=1}^{N}\frac{\left[\nabla_d F_i\left(x_i^{(t-1)}\right) - \nabla_d F\left(\bar{x}^{(t-1)}\right)\right]^2}{\sqrt{\epsilon + \tilde{v}_{i,d}^{(t)}}} - 2R\sum_{i=1}^{N}\mathbb{E}_{t-1}\left[\frac{\left(\nabla_d f_i(x_i^{(t-1)})\right)^2}{\epsilon + v_{i,d}^{(t)}}\right] \tag{15}$$

where

$$\kappa := \mathbb{E}_{t-1}\left[\nabla_d F\left(\bar{x}^{(t-1)}\right)\left(\sum_{i=1}^{N}\frac{\nabla_d f_i\left(x_i^{(t-1)}\right)}{\sqrt{\epsilon + v_{i,d}^{(t)}}}\right)\right].$$

*Proof.* According to the definition of $\kappa$, we have

$$\kappa = \bar{G}_d^{(t)}\mathbb{E}_{t-1}\left[\sum_{i=1}^{N}\frac{g_{i,d}^{(t)}}{\sqrt{\epsilon + v_{i,d}^{(t)}}}\right]$$

$$= \bar{G}_d^{(t)}\mathbb{E}_{t-1}\left[\sum_{i=1}^{N}g_{i,d}^{(t)}\left(\frac{1}{\sqrt{\epsilon + v_{i,d}^{(t)}}} \pm \frac{1}{\sqrt{\epsilon + \tilde{v}_{i,d}^{(t)}}}\right)\right]$$

$$= \bar{G}_d^{(t)}\mathbb{E}_{t-1}\left[\sum_{i=1}^{N}\frac{g_{i,d}^{(t)}}{\sqrt{\epsilon + \tilde{v}_{i,d}^{(t)}}}\right] + \mathbb{E}_{t-1}\left[\bar{G}_d^{(t)}\sum_{i=1}^{N}\left(\frac{g_{i,d}^{(t)}}{\sqrt{\epsilon + v_{i,d}^{(t)}}} - \frac{g_{i,d}^{(t)}}{\sqrt{\epsilon + \tilde{v}_{i,d}^{(t)}}}\right)\right] \tag{16}$$

$$= \underbrace{\bar{G}_d^{(t)}\sum_{i=1}^{N}\frac{G_{i,d}^{(t)}}{\sqrt{\epsilon + \tilde{v}_{i,d}^{(t)}}}}_{A} + \underbrace{\mathbb{E}_{t-1}\left[\bar{G}_d^{(t)}\sum_{i=1}^{N}\left(\frac{g_{i,d}^{(t)}}{\sqrt{\epsilon + v_{i,d}^{(t)}}} - \frac{g_{i,d}^{(t)}}{\sqrt{\epsilon + \tilde{v}_{i,d}^{(t)}}}\right)\right]}_{B}.$$

Firstly, for the term $A$, we have

$$A = \bar{G}_d^{(t)}\left(\sum_{i=1}^{N}\frac{G_{i,d}^{(t)} \pm \bar{G}_d^{(t)}}{\sqrt{\epsilon + \tilde{v}_{i,d}^{(t)}}}\right) = \bar{G}_d^{(t)}\left(\sum_{i=1}^{N}\frac{G_{i,d}^{(t)} - \bar{G}_d^{(t)}}{\sqrt{\epsilon + \tilde{v}_{i,d}^{(t)}}}\right) + \sum_{i=1}^{N}\frac{\left[\bar{G}_d^{(t)}\right]^2}{\sqrt{\epsilon + \tilde{v}_{i,d}^{(t)}}}. \tag{17}$$

By applying the Young's inequality (9) for $N$ times on the first term on the right hand of equation 17 and letting

$$\gamma = \frac{1}{2\sqrt{\epsilon + \tilde{v}_{i,d}^{(t)}}}, \quad a = \bar{G}_d^{(t)}, \quad b = \frac{G_{i,d}^{(t)} - \bar{G}_d^{(t)}}{\sqrt{\epsilon + \tilde{v}_{i,d}^{(t)}}},$$

we have

$$A \geq -\frac{1}{4}\sum_{i=1}^{N}\frac{\left[\bar{G}_d^{(t)}\right]^2}{\sqrt{\epsilon + \tilde{v}_{i,d}^{(t)}}} - \sum_{i=1}^{N}\frac{\left[G_{i,d}^{(t)} - \bar{G}_d^{(t)}\right]^2}{\sqrt{\epsilon + \tilde{v}_{i,d}^{(t)}}} + \sum_{i=1}^{N}\frac{\left[\bar{G}_d^{(t)}\right]^2}{\sqrt{\epsilon + \tilde{v}_{i,d}^{(t)}}}$$

$$= \frac{3}{4}\sum_{i=1}^{N}\frac{\left[\bar{G}_d^{(t)}\right]^2}{\sqrt{\epsilon + \tilde{v}_{i,d}^{(t)}}} - \sum_{i=1}^{N}\frac{\left[G_{i,d}^{(t)} - \bar{G}_d^{(t)}\right]^2}{\sqrt{\epsilon + \tilde{v}_{i,d}^{(t)}}}. \tag{18}$$

For the term $B$ in (16), we have

$$
\begin{aligned}
B &= \bar{G}_d^{(t)} \sum_{i=1}^N \left( \frac{g_{i,d}^{(t)}}{\sqrt{\epsilon + v_{i,d}^{(t)}}} - \frac{g_{i,d}^{(t)}}{\sqrt{\epsilon + \tilde{v}_{i,d}^{(t)}}} \right) \\
&= \bar{G}_d^{(t)} \sum_{i=1}^N \left[ \frac{g_{i,d}^{(t)} \left( \tilde{v}_{i,d}^{(t)} - v_{i,d}^{(t)} \right)}{\sqrt{\epsilon + v_{i,d}^{(t)}} \sqrt{\epsilon + \tilde{v}_{i,d}^{(t)}} \left( \sqrt{\epsilon + v_{i,d}^{(t)}} + \sqrt{\epsilon + \tilde{v}_{i,d}^{(t)}} \right)} \right] \\
&= \bar{G}_d^{(t)} \sum_{i=1}^N \left[ \frac{g_{i,d}^{(t)} \left[ \mathbb{E}_{t-1} \left[ \left( g_{i,d}^{(t)} \right)^2 \right] - \left( g_{i,d}^{(t)} \right)^2 \right]}{\sqrt{\epsilon + v_{i,d}^{(t)}} \sqrt{\epsilon + \tilde{v}_{i,d}^{(t)}} \left( \sqrt{\epsilon + v_{i,d}^{(t)}} + \sqrt{\epsilon + \tilde{v}_{i,d}^{(t)}} \right)} \right].
\end{aligned}
\tag{19}
$$

Given the fact that $\sqrt{\epsilon + v_{i,d}^{(t)}} > 0$ and $\sqrt{\epsilon + \tilde{v}_{i,d}^{(t)}} > 0$, we have

$$
\sqrt{\epsilon + v_{i,d}^{(t)}} + \sqrt{\epsilon + \tilde{v}_{i,d}^{(t)}} > \sqrt{\epsilon + v_{i,d}^{(t)}} \quad \text{and} \quad \sqrt{\epsilon + v_{i,d}^{(t)}} + \sqrt{\epsilon + \tilde{v}_{i,d}^{(t)}} > \sqrt{\epsilon + \tilde{v}_{i,d}^{(t)}}.
\tag{20}
$$

By taking the absolute value of $B$, we have

$$
|B| \leq \underbrace{\sum_{i=1}^N \frac{\left| \bar{G}_d^{(t)} g_{i,d}^{(t)} \right| \mathbb{E}_{t-1} \left[ \left( g_{i,d}^{(t)} \right)^2 \right]}{\sqrt{\epsilon + v_{i,d}^{(t)}} \left( \epsilon + \tilde{v}_{i,d}^{(t)} \right)}}_{C} + \underbrace{\sum_{i=1}^N \frac{\left| \bar{G}_d^{(t)} g_{i,d}^{(t)} \right| \left( g_{i,d}^{(t)} \right)^2}{\sqrt{\epsilon + \tilde{v}_{i,d}^{(t)}} \left( \epsilon + v_{i,d}^{(t)} \right)}}_{H}.
\tag{21}
$$

To bound $C$, by applying the Young's inequality equation 9 for $N$ times and let

$$
\gamma = \frac{\sqrt{\epsilon + \tilde{v}_{i,d}^{(t)}}}{2}, \quad a = \frac{\left| \bar{G}_d^{(t)} \right|}{\sqrt{\epsilon + \tilde{v}_{i,d}^{(t)}}}, \quad b = \frac{\left| g_{i,d}^{(t)} \right| \mathbb{E}_{t-1} \left[ \left( g_{i,d}^{(t)} \right)^2 \right]}{\sqrt{\epsilon + v_{i,d}^{(t)}} \sqrt{\epsilon + \tilde{v}_{i,d}^{(t)}}},
$$

we have

$$
C \leq \frac{1}{4} \sum_{i=1}^N \frac{\left[ \bar{G}_d^{(t)} \right]^2}{\sqrt{\epsilon + \tilde{v}_{i,d}^{(t)}}} + \sum_{i=1}^N \frac{\left( g_{i,d}^{(t)} \right)^2 \mathbb{E}_{t-1} \left[ \left( g_{i,d}^{(t)} \right)^2 \right]^2}{\left( \epsilon + v_{i,d}^{(t)} \right) \left( \epsilon + \tilde{v}_{i,d}^{(t)} \right)^{3/2}}.
\tag{22}
$$

Given the fact that $\sqrt{\epsilon + \tilde{v}_{i,d}^{(t)}} \geq \sqrt{\mathbb{E}_{t-1} \left[ \left( g_{i,d}^{(t)} \right)^2 \right]}$, using Assumption 2, and taking the conditional expectation on $C$, we have

$$
\begin{aligned}
\mathbb{E}_{t-1}[C] &\leq \frac{1}{4} \sum_{i=1}^N \frac{\left[ \bar{G}_d^{(t)} \right]^2}{\sqrt{\epsilon + \tilde{v}_{i,d}^{(t)}}} + \sum_{i=1}^N \mathbb{E}_{t-1} \left[ \frac{\left( g_{i,d}^{(t)} \right)^2}{\epsilon + v_{i,d}^{(t)}} \right] \frac{\mathbb{E}_{t-1} \left[ \left( g_{i,d}^{(t)} \right)^2 \right]^2}{\left( \epsilon + \tilde{v}_{i,d}^{(t)} \right)^{3/2}} \\
&\leq \frac{1}{4} \sum_{i=1}^N \frac{\left[ \bar{G}_d^{(t)} \right]^2}{\sqrt{\epsilon + \tilde{v}_{i,d}^{(t)}}} + \sum_{i=1}^N \mathbb{E}_{t-1} \left[ \frac{\left( g_{i,d}^{(t)} \right)^2}{\epsilon + v_{i,d}^{(t)}} \right] \sqrt{\mathbb{E}_{t-1} \left[ \left( g_{i,d}^{(t)} \right)^2 \right]} \\
&\leq \frac{1}{4} \sum_{i=1}^N \frac{\left[ \bar{G}_d^{(t)} \right]^2}{\sqrt{\epsilon + \tilde{v}_{i,d}^{(t)}}} + R \sum_{i=1}^N \mathbb{E}_{t-1} \left[ \frac{\left( g_{i,d}^{(t)} \right)^2}{\epsilon + v_{i,d}^{(t)}} \right].
\end{aligned}
\tag{23}
$$

Now we focus on $H$ in (21). By applying the Young's inequality equation 9 for $N$ times and let

$$\gamma = \frac{\sqrt{\epsilon + \tilde{v}_{i,d}^{(t)}}}{2\mathbb{E}_{t-1}\left[\left(g_{i,d}^{(t)}\right)^2\right]}, a = \frac{\left|\bar{G}_d^{(t)} g_{i,d}^{(t)}\right|}{\sqrt{\epsilon + \tilde{v}_{i,d}^{(t)}}}, b = \frac{\left(g_{i,d}^{(t)}\right)^2}{\epsilon + v_{i,d}^{(t)}},\tag{24}$$

we have

$$H \leq \frac{1}{4}\sum_{i=1}^{N}\frac{\left[\bar{G}_d^{(t)}\right]^2}{\sqrt{\epsilon + \tilde{v}_{i,d}^{(t)}}}\frac{\left(g_{i,d}^{(t)}\right)^2}{\mathbb{E}_{t-1}\left[\left(g_{i,d}^{(t)}\right)^2\right]} + \sum_{i=1}^{N}\frac{\mathbb{E}_{t-1}\left[\left(g_{i,d}^{(t)}\right)^2\right]}{\sqrt{\epsilon + \tilde{v}_{i,d}^{(t)}}}\frac{\left(g_{i,d}^{(t)}\right)^4}{\left(\epsilon + v_{i,d}^{(t)}\right)^2}.\tag{25}$$

Given the fact that $\epsilon + v_{i,d}^{(t)} \geq \left(g_{i,d}^{(t)}\right)^2$ and $\sqrt{\epsilon + \tilde{v}_{i,d}^{(t)}} \geq \sqrt{\mathbb{E}_{t-1}\left[\left(g_{i,d}^{(t)}\right)^2\right]^2}$, using Assumption 2, and taking the conditional expectation, we have

$$\begin{aligned}
E_{t-1}[H] &\leq \frac{1}{4}\sum_{i=1}^{N}\frac{\left[\bar{G}_d^{(t)}\right]^2}{\sqrt{\epsilon + \tilde{v}_{i,d}^{(t)}}}\frac{\mathbb{E}_{t-1}\left[\left(g_{i,d}^{(t)}\right)^2\right]}{\mathbb{E}_{t-1}\left[\left(g_{i,d}^{(t)}\right)^2\right]} + \sum_{i=1}^{N}\frac{\mathbb{E}_{t-1}\left[\left(g_{i,d}^{(t)}\right)^2\right]}{\sqrt{\epsilon + \tilde{v}_{i,d}^{(t)}}}\mathbb{E}_{t-1}\left[\frac{\left(g_{i,d}^{(t)}\right)^4}{\left(\epsilon + v_{i,d}^{(t)}\right)^2}\right]\\
&\leq \frac{1}{4}\sum_{i=1}^{N}\frac{\left[\bar{G}_d^{(t)}\right]^2}{\sqrt{\epsilon + \tilde{v}_{i,d}^{(t)}}} + R\sum_{i=1}^{N}\mathbb{E}_{t-1}\left[\frac{\left(g_{i,d}^{(t)}\right)^2}{\epsilon + v_{i,d}^{(t)}}\right].
\end{aligned}\tag{26}$$

By plugging (23) and (26) back to (21), we have

$$\begin{aligned}
\mathbb{E}_{t-1}[B] &\geq -\mathbb{E}_{t-1}[|B|]\\
&\geq -\mathbb{E}_{t-1}[|C|] - \mathbb{E}_{t-1}[|H|]\\
&\geq -\frac{1}{2}\sum_{i=1}^{N}\frac{\left[\bar{G}_d^{(t)}\right]^2}{\sqrt{\epsilon + \tilde{v}_{i,d}^{(t)}}} - 2R\sum_{i=1}^{N}\mathbb{E}_{t-1}\left[\frac{\left(g_{i,d}^{(t)}\right)^2}{\epsilon + v_{i,d}^{(t)}}\right].
\end{aligned}\tag{27}$$

By plugging (18) and (27) back to (16), we have

$$\begin{aligned}
\kappa &= A + \mathbb{E}_{t-1}[B]\\
&\geq \frac{1}{4}\sum_{i=1}^{N}\frac{\left[\bar{G}_d^{(t)}\right]^2}{\sqrt{\epsilon + \tilde{v}_{i,d}^{(t)}}} - \sum_{i=1}^{N}\frac{\left[G_{i,d}^{(t)} - \bar{G}_d^{(t)}\right]^2}{\sqrt{\epsilon + \tilde{v}_{i,d}^{(t)}}} - 2R\sum_{i=1}^{N}\mathbb{E}_{t-1}\left[\frac{\left(g_{i,d}^{(t)}\right)^2}{\epsilon + v_{i,d}^{(t)}}\right]\\
&= \frac{1}{4}\sum_{i=1}^{N}\frac{\left[\nabla_d F\left(\bar{x}^{(t-1)}\right)\right]^2}{\sqrt{\epsilon + \tilde{v}_{i,d}^{(t)}}} - \sum_{i=1}^{N}\frac{\left[\nabla_d F_i\left(x_i^{(t-1)}\right) - \nabla_d F\left(\bar{x}^{(t-1)}\right)\right]^2}{\sqrt{\epsilon + \tilde{v}_{i,d}^{(t)}}}\\
&\quad - 2R\sum_{i=1}^{N}\mathbb{E}_{t-1}\left[\frac{\left(\nabla_d f_i(x_i^{(t-1)})\right)^2}{\epsilon + v_{i,d}^{(t)}}\right],
\end{aligned}\tag{28}$$

which completes the proof of Lemma 1. $\qquad\square$

**Lemma 2** (sum of ratios with the denominator being the sum of past numerators) Given $0 < \beta_2 < 1$ and a non-negative sequence $(a_t)_{t\in\mathbb{N}^+}$, define $b_t = \sum_{j=1}^{t}\beta_2^{t-j}a_j$ for all $t \in \mathbb{N}^+$, then we have

$$\sum_{j=1}^{T}\frac{a_j}{\epsilon + b_j} \leq \ln\left(1 + \frac{b_T}{\epsilon}\right) - T\ln(\beta_2).\tag{29}$$

*Proof.* Refer to Lemma 5.2 in Défossez et al. (2020). □

**Lemma 3** (averaged consensus error is bounded if sampling from i.i.d. data distribution)    Under Assumptions 1–3, for Algorithm 1, we have

$$\frac{1}{T}\sum_{t=1}^{T}\sum_{i=1}^{N}\sum_{d=1}^{D}\left[\nabla_d F_i(x_i^{(t-1)}) - \nabla_d F(\bar{x}_i^{(t-1)})\right]^2$$

$$\leq \frac{2(1+\lambda^2)\alpha^2 N L^2 D}{(1-\lambda^2)^2(1-\beta_2)}\left[\frac{1}{T}\ln\left(1+\frac{R^2}{\epsilon(1-\beta_2)}\right) - \ln(\beta_2)\right]. \tag{30}$$

*Proof.* Denote the concatenation of the $d$-th dimension of all local models at $t$-th iteration as

$$\mathbf{x}_d^{(t)} := \begin{bmatrix} x_{1,d}^{(t)} & x_{2,d}^{(t)} & \cdots & x_{N,d}^{(t)} \end{bmatrix}^\top \in \mathbb{R}^N.$$

Denote the concatenation of $N$ $d$-th dimension of the (virtual) averaged model as

$$\bar{\mathbf{x}}_d^{(t)} := \begin{bmatrix} \bar{x}_d^{(t)} & \bar{x}_d^{(t)} & \cdots & \bar{x}_d^{(t)} \end{bmatrix}^\top \in \mathbb{R}^N.$$

Denote the concatenation of the $d$-th dimension of updates of all local models at $t$-th iteration as

$$\mathbf{e}_d^{(t)} = -\alpha_t \left[\frac{\nabla_d f_1\left(x_1^{(t-1)}\right)}{\sqrt{\epsilon + v_{1,d}^{(t)}}}, \cdots, \frac{\nabla_d f_N\left(x_N^{(t-1)}\right)}{\sqrt{\epsilon + v_{N,d}^{(t)}}}\right] \in \mathbb{R}^N.$$

Denote the concatenation of $N$ $d$-th dimension of updates of the (virtual) averaged model at $t$-th iteration as

$$\bar{\mathbf{e}}_d^{(t)} = -\frac{\alpha_t}{N}\left[\sum_{i=1}^{N}\frac{\nabla_d f_i\left(x_i^{(t-1)}\right)}{\sqrt{\epsilon + v_{i,d}^{(t)}}}, \cdots, \sum_{i=1}^{N}\frac{\nabla_d f_i\left(x_i^{(t-1)}\right)}{\sqrt{\epsilon + v_{i,d}^{(t)}}}\right] \in \mathbb{R}^N.$$

From the update function (12), it can be inferred that

$$\bar{\mathbf{x}}_d^{(t)} = \bar{\mathbf{x}}_d^{(t-1)} + \bar{\mathbf{e}}_d^{(t)} = \frac{1}{N}\mathbf{1}_N \mathbf{x}_d^{(t-1)} + \frac{1}{N}\mathbf{1}_N \mathbf{e}_d^{(t)} = W^\infty \mathbf{x}_d^{(t-1)} + W^\infty \mathbf{e}_d^{(t)} \tag{31}$$

The update of $\mathbf{x}_d^{(t)}$ can be rewritten as

$$\mathbf{x}_d^{(t)} = W\mathbf{x}_d^{(t-1)} + \mathbf{e}_d^{(t)} \tag{32}$$

Then we have

$$\left\|\mathbf{x}_d^{(t)} - \bar{\mathbf{x}}_d^{(t)}\right\|_2^2 = \left\|W\mathbf{x}_d^{(t-1)} + \mathbf{e}_d^{(t)} - \bar{\mathbf{x}}_d^{(t-1)} - \bar{\mathbf{e}}_d^{(t)}\right\|_2^2$$

$$= \left\|\left(W - \frac{1}{N}\mathbf{1}_N\right)\mathbf{x}_d^{(t-1)} + \left(I - \frac{1}{N}\mathbf{1}_N\right)\mathbf{e}_d^{(t)}\right\|_2^2$$

$$= \left\|\left(W - \frac{1}{N}\mathbf{1}_N\right)\mathbf{x}_d^{(t-1)} - \left(W - \frac{1}{N}\mathbf{1}_N\right)\bar{\mathbf{x}}_d^{(t)} + \left(I - \frac{1}{N}\mathbf{1}_N\right)\mathbf{e}_d^{(t)}\right\|_2^2$$

$$= \left\|\left(W - \frac{1}{N}\mathbf{1}_N\right)\left(\mathbf{x}_d^{(t-1)} - \bar{\mathbf{x}}_d^{(t-1)}\right) + \left(I - \frac{1}{N}\mathbf{1}_N\right)\mathbf{e}_d^{(t)}\right\|_2^2 \tag{33}$$

$$= \left\|\left(W - \frac{1}{N}\mathbf{1}_N\right)\left(\mathbf{x}_d^{(t-1)} - \bar{\mathbf{x}}_d^{(t-1)}\right)\right\|_2^2 + \left\|\left(I - \frac{1}{N}\mathbf{1}_N\right)\mathbf{e}_d^{(t)}\right\|_2^2$$

$$+ 2\left[\left(W - \frac{1}{N}\mathbf{1}_N\right)\left(\mathbf{x}_d^{(t-1)} - \bar{\mathbf{x}}_d^{(t-1)}\right)\right]^\top\left[\left(I - \frac{1}{N}\mathbf{1}_N\right)\mathbf{e}_d^{(t)}\right].$$

Recall the definition of $\lambda$ in Assumption 3. By applying the Young's inequality equation 9 and letting

$$\gamma = \frac{1 - \lambda^2}{2\lambda^2} > 0, \quad a = \left( W - \frac{1}{N} \mathbf{1}_N \right) \left( \mathbf{x}_d^{(t-1)} - \bar{\mathbf{x}}_d^{(t-1)} \right), \quad b = \left( I - \frac{1}{N} \mathbf{1}_N \right) \mathbf{e}_d^{(t)},$$

we have

$$\left\| \mathbf{x}_d^{(t)} - \bar{\mathbf{x}}_d^{(t)} \right\|_2^2 \leq \frac{1 + \lambda^2}{2\lambda^2} \left\| \left( W - \frac{1}{N} \mathbf{1}_N \right) \left( \mathbf{x}_d^{(t-1)} - \bar{\mathbf{x}}_d^{(t-1)} \right) \right\|_2^2 + \frac{1 + \lambda^2}{1 - \lambda^2} \left\| \left( I - \frac{1}{N} \mathbf{1}_N \right) \mathbf{e}_d^{(t)} \right\|_2^2. \tag{34}$$

By using (11), we have

$$\left\| \mathbf{x}_d^{(t)} - \bar{\mathbf{x}}_d^{(t)} \right\|_2^2 \leq \frac{1 + \lambda^2}{2} \left\| \mathbf{x}_d^{(t-1)} - \bar{\mathbf{x}}_d^{(t-1)} \right\|_2^2 + \frac{1 + \lambda^2}{1 - \lambda^2} \left\| \mathbf{e}_d^{(t)} \right\|_2^2. \tag{35}$$

Given that all local models start from the same point $x^{(0)}$, by summing up for all $t \in [T]$ and using telescoping, we have

$$\frac{1 - \lambda^2}{2} \sum_{t=1}^{T} \left\| \mathbf{x}_d^{(t-1)} - \bar{\mathbf{x}}_d^{(t-1)} \right\|_2^2 \leq \frac{1 + \lambda^2}{1 - \lambda^2} \sum_{t=1}^{T} \left\| \mathbf{e}_d^{(t)} \right\|_2^2. \tag{36}$$

Given that $\alpha_t \leq \frac{\alpha}{\sqrt{1 - \beta_2}}$ and using Lemma 2, we have

$$\frac{1 - \lambda^2}{2} \sum_{t=1}^{T} \left\| \mathbf{x}_d^{(t-1)} - \bar{\mathbf{x}}_d^{(t-1)} \right\|_2^2 \leq \frac{1 + \lambda^2}{1 - \lambda^2} \sum_{i=1}^{N} \sum_{t=1}^{T} \alpha_t^2 \frac{\left( g_{i,d}^{(t)} \right)^2}{\epsilon + v_{i,d}^{(t)}}$$

$$\leq \frac{(1 + \lambda^2)\alpha^2}{(1 - \lambda^2)(1 - \beta_2)} \sum_{i=1}^{N} \sum_{t=1}^{T} \frac{\left( g_{i,d}^{(t)} \right)^2}{\epsilon + v_{i,d}^{(t)}} \tag{37}$$

$$\leq \frac{(1 + \lambda^2)\alpha^2}{(1 - \lambda^2)(1 - \beta_2)} \sum_{i=1}^{N} \left[ \ln \left( 1 + \frac{v_{i,d}^{(t)}}{\epsilon} \right) - T \ln(\beta_2) \right].$$

By using Assumption 2 on the above inequality, we have

$$\frac{1}{T} \sum_{t=1}^{T} \left\| \mathbf{x}_d^{(t-1)} - \bar{\mathbf{x}}_d^{(t-1)} \right\|_2^2 \leq \frac{2(1 + \lambda^2)\alpha^2 N}{(1 - \lambda^2)^2(1 - \beta_2)} \left[ \frac{1}{T} \ln \left( 1 + \frac{R^2}{\epsilon(1 - \beta_2)} \right) - \ln(\beta_2) \right]. \tag{38}$$

We now focus on the left side of (30). By using iid assumption and Assumption 1, we have

$$\frac{1}{T} \sum_{t=1}^{T} \sum_{i=1}^{N} \sum_{d=1}^{D} \left[ \nabla_d F_i(x_i^{(t-1)}) - \nabla_d F(\bar{x}_i^{(t-1)}) \right]^2 = \frac{1}{T} \sum_{t=1}^{T} \sum_{i=1}^{N} \sum_{d=1}^{D} \left[ \nabla_d F(x_i^{(t-1)}) - \nabla_d F(\bar{x}_i^{(t-1)}) \right]^2$$

$$= \frac{1}{T} \sum_{t=1}^{T} \sum_{i=1}^{N} \left\| \nabla F(x_i^{(t-1)}) - \nabla F(\bar{x}_i^{(t-1)}) \right\|_2^2$$

$$\leq \frac{L^2}{T} \sum_{t=1}^{T} \sum_{i=1}^{N} \left\| x_i^{(t-1)} - \bar{x}^{(t-1)} \right\|_2^2$$

$$= \frac{L^2}{T} \sum_{t=1}^{T} \sum_{d=1}^{D} \left\| \mathbf{x}_d^{(t-1)} - \bar{\mathbf{x}}_d^{(t-1)} \right\|_2^2 \tag{39}$$

Plugging (38) to (39), we have

$$\frac{1}{T} \sum_{t=1}^{T} \sum_{i=1}^{N} \sum_{d=1}^{D} \left[ \nabla_d F_i(x_i^{(t-1)}) - \nabla_d F(\bar{x}_i^{(t-1)}) \right]^2 \leq \frac{2(1 + \lambda^2)\alpha^2 N L^2 D}{(1 - \lambda^2)^2(1 - \beta_2)} \left[ \frac{1}{T} \ln \left( 1 + \frac{R^2}{\epsilon(1 - \beta_2)} \right) - \ln(\beta_2) \right], \tag{40}$$

which completes the proof of Lemma 3. $\square$

### B.3.3 MAIN PROOF OF THEOREM B.1

From Assumption 1, we have

$$F\left(\bar{x}^{(t)}\right) \leq F\left(\bar{x}^{(t-1)}\right) + \nabla F\left(\bar{x}^{(t-1)}\right)^{\top}\left(\bar{x}^{(t)} - \bar{x}^{(t-1)}\right) + \frac{L}{2}\left\|\bar{x}^{(t)} - \bar{x}^{(t-1)}\right\|_2^2. \tag{41}$$

By substituting equation 13 into equation 41 and taking the conditional expectation on both sides, we have

$$\begin{aligned}
\mathbb{E}_{t-1}\left[F\left(\bar{x}^{(t)}\right)\right] &\leq F\left(\bar{x}^{(t-1)}\right) - \frac{\alpha_t}{N}\sum_{d=1}^{D}\mathbb{E}_{t-1}\left[\nabla_d F\left(\bar{x}^{(t-1)}\right)^{\top}\sum_{i=1}^{N}\frac{\nabla_d f_i\left(x_i^{(t-1)}\right)}{\sqrt{\epsilon + v_{i,d}^{(t)}}}\right] \\
&\quad + \frac{L\alpha_t^2}{2}\sum_{d=1}^{D}\mathbb{E}_{t-1}\left[\left\|\frac{1}{N}\sum_{i=1}^{N}\frac{\nabla_d f_i(x_i^{(t-1)})}{\sqrt{\epsilon + v_{i,d}^{(t)}}}\right\|_2^2\right].
\end{aligned} \tag{42}$$

By applying (10) to (42) and letting

$$a_i = \frac{\nabla_d f_i(x_i^{(t-1)})}{N\sqrt{\epsilon + v_{i,d}^{(t)}}},$$

we have

$$\begin{aligned}
\mathbb{E}_{t-1}\left[F\left(\bar{x}^{(t)}\right)\right] &\leq F\left(\bar{x}^{(t-1)}\right) - \frac{\alpha_t}{N}\sum_{d=1}^{D}\mathbb{E}_{t-1}\left[\nabla_d F\left(\bar{x}^{(t-1)}\right)^{\top}\sum_{i=1}^{N}\frac{\nabla_d f_i\left(x_i^{(t-1)}\right)}{\sqrt{\epsilon + v_{i,d}^{(t)}}}\right] \\
&\quad + \frac{L\alpha_t^2}{2N}\sum_{d=1}^{D}\mathbb{E}_{t-1}\left[\sum_{i=1}^{N}\frac{\left[\nabla_d f_i(x_i^{(t-1)})\right]^2}{\epsilon + v_{i,d}^{(t)}}\right].
\end{aligned} \tag{43}$$

By applying Lemma 1 to (43), we have

$$\begin{aligned}
\mathbb{E}_{t-1}\left[F\left(\bar{x}^{(t)}\right)\right] &\leq F\left(\bar{x}^{(t-1)}\right) \underbrace{- \frac{\alpha_t}{4N}\sum_{d=1}^{D}\sum_{i=1}^{N}\frac{\left[\nabla_d F\left(\bar{x}^{(t-1)}\right)\right]^2}{\sqrt{\epsilon + \tilde{v}_{i,d}^{(t)}}}}_{A} \\
&\quad + \frac{\alpha_t}{N}\sum_{d=1}^{D}\sum_{i=1}^{N}\frac{\left[\nabla_d F_i\left(x_i^{(t-1)}\right) - \nabla_d F\left(\bar{x}^{(t-1)}\right)\right]^2}{\sqrt{\epsilon + \tilde{v}_{i,d}^{(t)}}} \\
&\quad + \left(\frac{2\alpha_t R}{N} + \frac{L\alpha_t^2}{2N}\right)\sum_{d=1}^{D}\mathbb{E}_{t-1}\left[\sum_{i=1}^{N}\frac{\left[\nabla_d f_i(x_i^{(t-1)})\right]^2}{\epsilon + v_{i,d}^{(t)}}\right].
\end{aligned} \tag{44}$$

Now we focus on the term $A$ in (44). From Assumption 2, we have $\sqrt{\epsilon + \tilde{v}_{i,d}^{(t)}} \leq R\sqrt{\frac{1-\beta_2^t}{1-\beta_2}}$, and recall that $\alpha_t = \alpha\sqrt{\frac{1-\beta_2^t}{1-\beta_2}}$. Then we have

$$A \leq -\frac{\alpha}{4R}\sum_{d=1}^{D}\left(\nabla_d F(\bar{x}^{(t-1)})\right)^2 = -\frac{\alpha}{4R}\left\|\nabla_d F(\bar{x}^{(t-1)})\right\|_2^2. \tag{45}$$

Given the fact that $\alpha_t \leq \frac{\alpha}{\sqrt{1-\beta_2}}$, by plugging (45) into (44), we have

$$
\begin{aligned}
\mathbb{E}_{t-1}\left[F\left(\bar{x}^{(t)}\right)\right] \leq {} & F\left(\bar{x}^{(t-1)}\right) - \frac{\alpha}{4R}\left\|\nabla_d F(\bar{x}^{(t-1)})\right\|_2^2 \\
& + \frac{\alpha}{N\sqrt{1-\beta_2}}\sum_{d=1}^{D}\sum_{i=1}^{N}\frac{\left[\nabla_d F_i\left(x_i^{(t-1)}\right) - \nabla_d F\left(\bar{x}^{(t-1)}\right)\right]^2}{\sqrt{\epsilon + \tilde{v}_{i,d}^{(t)}}} \\
& + \left(\frac{2\alpha R}{N\sqrt{1-\beta_2}} + \frac{L\alpha^2}{2N(1-\beta_2)}\right)\sum_{d=1}^{D}\mathbb{E}_{t-1}\left[\sum_{i=1}^{N}\frac{\left[\nabla_d f_i(x_i^{(t-1)})\right]^2}{\epsilon + v_{i,d}^{(t)}}\right].
\end{aligned}
\tag{46}
$$

By taking the total expectation and summing up (46) over $t \in [T]$, we have

$$
\begin{aligned}
\mathbb{E}\left[F\left(\bar{x}^{(T)}\right)\right] \leq {} & F\left(\bar{x}^{(0)}\right) - \frac{\alpha}{4R}\sum_{t=1}^{T}\left\|\nabla_d F(\bar{x}^{(t-1)})\right\|_2^2 \\
& + \underbrace{\frac{\alpha}{N\sqrt{1-\beta_2}}\sum_{t=1}^{T}\sum_{d=1}^{D}\sum_{i=1}^{N}\frac{\left[\nabla_d F_i\left(x_i^{(t-1)}\right) - \nabla_d F\left(\bar{x}^{(t-1)}\right)\right]^2}{\sqrt{\epsilon + \tilde{v}_{i,d}^{(t)}}}}_{B} \\
& + \underbrace{\left(\frac{2\alpha R}{N\sqrt{1-\beta_2}} + \frac{L\alpha^2}{2N(1-\beta_2)}\right)\sum_{t=1}^{T}\sum_{d=1}^{D}\mathbb{E}_{t-1}\left[\sum_{i=1}^{N}\frac{\left[\nabla_d f_i(x_i^{(t-1)})\right]^2}{\epsilon + v_{i,d}^{(t)}}\right]}_{C}.
\end{aligned}
\tag{47}
$$

By applying Lemma 2 for each $d \in [D]$ on $C$, we have

$$
\begin{aligned}
C & \leq \left(\frac{2\alpha R}{N\sqrt{1-\beta_2}} + \frac{L\alpha^2}{2N(1-\beta_2)}\right)\sum_{d=1}^{D}\sum_{i=1}^{N}\left[\ln\left(1 + \frac{v_{i,d}^{(t)}}{\epsilon}\right) - T\ln(\beta_2)\right] \\
& \leq \left(\frac{2\alpha R}{N\sqrt{1-\beta_2}} + \frac{L\alpha^2}{2N(1-\beta_2)}\right)\sum_{d=1}^{D}\sum_{i=1}^{N}\left[\ln\left(1 + \frac{R^2}{\epsilon(1-\beta_2)}\right) - T\ln(\beta_2)\right] \\
& = \left(\frac{2\alpha DR}{\sqrt{1-\beta_2}} + \frac{DL\alpha^2}{2(1-\beta_2)}\right)\left[\ln\left(1 + \frac{R^2}{\epsilon(1-\beta_2)}\right) - T\ln(\beta_2)\right].
\end{aligned}
\tag{48}
$$

Given that $\sqrt{\epsilon + \tilde{v}_{i,d}^{(t)}} > \sqrt{\epsilon}$, by applying Lemma 3 on $B$ in (47), we have

$$
B \leq \frac{2(1+\lambda^2)\alpha^3 L^2 D}{(1-\lambda^2)^2(1-\beta_2)^{3/2}\sqrt{\epsilon}}\left[\ln\left(1 + \frac{R^2}{\epsilon(1-\beta_2)}\right) - T\ln(\beta_2)\right].
\tag{49}
$$

By plugging (49) and (48) into (47), we have

$$
\frac{1}{T}\sum_{t=1}^{T}\left\|\nabla_d F(\bar{x}^{(t-1)})\right\|_2^2 \leq \frac{4R}{\alpha T}\left[F\left(\bar{x}^{(0)}\right) - F_*\right] + \frac{1}{T}\zeta\left[\ln\left(1 + \frac{R^2}{\epsilon(1-\beta_2)}\right) - T\ln(\beta_2)\right],
\tag{50}
$$

where

$$
E := \frac{8DR^2}{\sqrt{1-\beta_2}} + \frac{2DLR\alpha}{(1-\beta_2)} + \frac{8(1+\lambda^2)\alpha^2 L^2 DR}{(1-\lambda^2)^2(1-\beta_2)^{3/2}\sqrt{\epsilon}}.
\tag{51}
$$

which completes the proof of Theorem B.1.

## B.4 PROOF OF THEOREM 4.1: DECENTRALIZED ADAM WITH HEAVY-BALL MOMENTUM

### B.4.1 ALGORITHM UPDATE UNDER ANALYSIS

With heavy-ball momentum, we set $0 < \beta_1 < \beta_2 < 1$. We have the algorithm update under analysis is

$$
g_i^{(t)} = \nabla\ell(x_i^{(t-1)}; \xi_i^{(t)}), \quad m_{i,d}^{(t)} = \beta_1 m_{i,d}^{(t-1)} + g_{i,d}^{(t)}, \quad v_{i,d}^{(t)} = \beta_2 v_{i,d}^{(t-1)} + \left[g_{i,d}^{(t)}\right]^2,
$$

$$
\alpha_t = \alpha(1-\beta_1)\sqrt{\frac{1-\beta_2^t}{1-\beta_2}}, \quad x_{i,d}^{(t)} = -\alpha_t \frac{m_{i,d}^{(t)}}{\sqrt{\epsilon + v_{i,d}^{(t)}}} + \sum_{j\in\mathcal{N}_i} w_{ij} x_{j,d}^{(t-1)} \tag{52}
$$

It should be noted that there is a modification on $\alpha_t$, which should originally be $\alpha_t = \alpha\frac{1-\beta_1}{1-\beta_1^t}\sqrt{\frac{1-\beta_2^t}{1-\beta_2}}$. However, $1 - \beta_2^t$ could potentially make the learning rate non-monotonic. Since replacing it by 1 has little practical influence, to simplify the proof, we replace it by 1 as in Défossez et al. (2020).

The (virtual) update of the globally averaged weights is

$$
\bar{x}_d^{(t)} = -\frac{\alpha_t}{N} \sum_{i=1}^{N} \frac{m_{i,d}^{(t)}}{\sqrt{\epsilon + v_{i,d}^{(t)}}} + \bar{x}_d^{(t-1)}. \tag{53}
$$

We denote $\tilde{v}_i^{(t)} \in \mathbb{R}^D$ as $\tilde{v}_i^{(t)} = \beta_2 v_i^{(t-1)} + \mathbb{E}_{t-1}\left[\left(g_{i,d}^{(t)}\right)^2\right]$ which differs from $v_{i,d}^{(t)}$ by replacing the last term with its conditional expectation. In addition, we denote

$$
\bar{G}_d^{(t)} := \nabla_d F\left(\bar{x}^{(t-1)}\right) = \frac{1}{N}\sum_{i=1}^{N}\nabla_d F_i\left(\bar{x}^{(t-1)}\right) \quad \text{and} \quad G_{i,d}^{(t)} := \nabla_d F_i(x_i^{(t-1)}). \tag{54}
$$

We denote the $d$-th dimension of the local update of the $i$-th local model at $t$-th iteration without the heavy-ball momentum as

$$
U_{i,d}^{(t)} := \frac{g_{i,d}^{(t)}}{\sqrt{\epsilon + v_{i,d}^{(t)}}}.
$$

We denote the $d$-th dimension of the local update of the $i$-th local model at $t$-th iteration with the heavy-ball momentum as

$$
u_{i,d}^{(t)} := \frac{m_{i,d}^{(t)}}{\sqrt{\epsilon + v_{i,d}^{(t)}}} = \frac{1}{\sqrt{\epsilon + v_{i,d}^{(t)}}} \sum_{k=0}^{t-1} \beta_2^k g_{i,d}^{(t-k)}.
$$

By replacing the last $k$ terms in $v_{i,d}^{(t)}$ with their expectation, we denote $\tilde{v}_{i,d}^{(t,k)}$ as

$$
\tilde{v}_{i,d}^{(t,k)} := \beta_2^k v_{i,d}^{(t-k)} + \mathbb{E}_{t-k-1}\left[\sum_{j=t-k+1}^{t} \beta_2^{t-j}\left(g_{i,d}^{(j)}\right)^2\right]. \tag{55}
$$

### B.4.2 TECHNICAL LEMMAS

**Lemma 4** (adaptive update with momentum approximately follow a descent direction)

$$
\kappa \geq \frac{1}{4}\sum_{d=1}^{D}\sum_{k=0}^{t-1}\beta_1^k\sum_{i=1}^{N}\mathbb{E}\left[\frac{\left(\bar{G}_d^{(t-k)}\right)^2}{\sqrt{\epsilon + \tilde{v}_{i,d}^{(t,k)}}}\right] - \sum_{d=1}^{D}\sum_{k=0}^{t-1}\beta_1^k\sum_{i=1}^{N}\mathbb{E}\left[\frac{\left(G_{i,d}^{(t-k)} - \bar{G}_d^{(t-k)}\right)^2}{\sqrt{\epsilon + \tilde{v}_{i,d}^{(t,k)}}}\right]
$$

$$
- \frac{3R}{\sqrt{1-\beta_1}}\sum_{k=0}^{t-1}\left(\frac{\beta_1}{\beta_2}\right)^k \sqrt{k+1}\sum_{i=1}^{N}\mathbb{E}\left[\left\|U_i^{(t-k)}\right\|_2^2\right] - \frac{\alpha_t^2 L^2\sqrt{1-\beta_1}}{4R}\sum_{i=1}^{N}\sum_{l=1}^{t-1}\mathbb{E}\left[\left\|u_i^{(t-l)}\right\|_2^2\right]\sum_{k=l}^{t-1}\beta_1^k\sqrt{k},
\tag{56}
$$

where

$$\kappa = \mathbb{E}\left[\sum_{d=1}^{D} \bar{G}_d^{(t)}\left(\sum_{i=1}^{N} \frac{m_{i,d}^{(t)}}{\sqrt{\epsilon + v_{i,d}^{(t)}}}\right)\right]. \tag{57}$$

*Proof.* According to the definition of $\kappa$, we have

$$\begin{aligned}
\kappa &= \sum_{d=1}^{D}\sum_{k=0}^{t-1}\beta_1^k \bar{G}_d^{(t)}\sum_{i=1}^{N}\frac{g_{i,d}^{(t-k)}}{\sqrt{\epsilon + v_{i,d}^{(t)}}} \\
&= \underbrace{\sum_{d=1}^{D}\sum_{k=0}^{t-1}\beta_1^k \bar{G}_d^{(t-k)}\sum_{i=1}^{N}\frac{g_{i,d}^{(t-k)}}{\sqrt{\epsilon + v_{i,d}^{(t)}}}}_{A} + \underbrace{\sum_{d=1}^{D}\sum_{k=0}^{t-1}\sum_{i=1}^{N}\beta_1^k\left(\bar{G}_d^{(t)} - \bar{G}_d^{(t-k)}\right)\frac{g_{i,d}^{(t-k)}}{\sqrt{\epsilon + v_{i,d}^{(t)}}}}_{B}.
\end{aligned} \tag{58}$$

Now focus on $B$. By applying the Young's inequality equation 9 and letting

$$\gamma = \frac{\sqrt{1-\beta_1}}{2R\sqrt{k+1}}, a = \left|\bar{G}_d^{(t)} - \bar{G}_d^{(t-k)}\right|, b = \frac{\left|g_{i,d}^{(t-k)}\right|}{\sqrt{\epsilon + v_{i,d}^{(t)}}},$$

we have

$$|B| \le \sum_{d=1}^{D}\sum_{k=0}^{t-1}\sum_{i=1}^{N}\beta_1^k\left[\frac{\sqrt{1-\beta_1}}{4R\sqrt{k+1}}\left(\bar{G}_d^{(t)} - \bar{G}_d^{(t-k)}\right)^2 + \frac{R\sqrt{k+1}}{\sqrt{1-\beta_1}}\frac{\left(g_{i,d}^{(t-k)}\right)^2}{\epsilon + v_{i,d}^{(t)}}\right]. \tag{59}$$

Given the fact that $\epsilon + v_{i,d}^{(t)} \ge \epsilon + \beta_2^k v_{i,d}^{(t-k)} \ge \beta_2^k(\epsilon + v_{i,d}^{(t-k)})$, we have

$$\sum_{d=1}^{D}\frac{\left(g_{i,d}^{(t-k)}\right)^2}{\epsilon + v_{i,d}^{(t)}} \le \frac{1}{\beta_2^k}\left\|U_i^{(t-k)}\right\|_2^2. \tag{60}$$

By using Assumption 1 and (10), we have

$$\begin{aligned}
\sum_{d=1}^{D}\left(\bar{G}_d^{(t)} - \bar{G}_d^{(t-k)}\right)^2 &= \left\|\bar{G}^{(t)} - \bar{G}^{(t-k)}\right\|_2^2 \\
&= \left\|\nabla F\left(\bar{x}^{(t)}\right) - \nabla F\left(\bar{x}^{(t-k)}\right)\right\|_2^2 \\
&\le L^2\left\|\frac{1}{N}\sum_{i=1}^{N}\sum_{l=1}^{k}\alpha_{t-l}u_i^{(t-l)}\right\|_2^2 \\
&\le \frac{\alpha_t^2 L^2 k}{N}\sum_{i=1}^{N}\sum_{l=1}^{k}\left\|u_i^{(t-l)}\right\|_2^2.
\end{aligned} \tag{61}$$

By plugging (61) and (60) back to (59), and given that $k/\sqrt{k+1} \le \sqrt{k}$ for $k \ge 0$, we have

$$\begin{aligned}
|B| &\le \left(\sum_{k=0}^{t-1}\frac{\alpha_t^2 L^2 \beta_1^k\sqrt{1-\beta_1}\sqrt{k}}{4R}\sum_{l=1}^{k}\sum_{i=1}^{N}\left\|u_i^{(t-l)}\right\|_2^2\right) + \left[\sum_{k=0}^{t-1}\frac{R\sqrt{k+1}}{\sqrt{1-\beta_1}}\left(\frac{\beta_1}{\beta_2}\right)^k\sum_{i=1}^{N}\left\|U_i^{(t-k)}\right\|_2^2\right] \\
&= \frac{\alpha_t^2 L^2\sqrt{1-\beta_1}}{4R}\sum_{k=0}^{t-1}\beta_1^k\sqrt{k}\sum_{l=1}^{k}\sum_{i=1}^{N}\left\|u_i^{(t-l)}\right\|_2^2 + \frac{R}{\sqrt{1-\beta_1}}\sum_{k=0}^{t-1}\sqrt{k+1}\left(\frac{\beta_1}{\beta_2}\right)^k\sum_{i=1}^{N}\left\|U_i^{(t-k)}\right\|_2^2 \\
&= \frac{\alpha_t^2 L^2\sqrt{1-\beta_1}}{4R}\sum_{i=1}^{N}\sum_{l=1}^{t-1}\left\|u_i^{(t-l)}\right\|_2^2\sum_{k=l}^{t-1}\beta_1^k\sqrt{k} + \frac{R}{\sqrt{1-\beta_1}}\sum_{i=1}^{N}\sum_{k=0}^{t-1}\sqrt{k+1}\left(\frac{\beta_1}{\beta_2}\right)^k\left\|U_i^{(t-k)}\right\|_2^2.
\end{aligned} \tag{62}$$

Now focus on $A$ in (58). Temporarily introduce the simplified notation as

$$\delta^2 = \sum_{j=t-k}^{t} \beta_2^{t-j} \left( g_{i,d}^{(j)} \right)^2, \quad r^2 = \mathbb{E}_{t-k-1} \left[ \delta^2 \right]$$

$$G := G_{i,d}^{(t-k)}, \quad \bar{G} = \bar{G}_d^{(t-k)}, \quad g := g_{i,d}^{(t-k)}, \quad v := v_{i,d}^{(t)}, \quad \tilde{v} = \tilde{v}_{i,d}^{(t,k)}.$$

Then we have $\tilde{v} - v = r^2 - \delta^2$.

Now focus on $A$ in (58), with the new notation, we have

$$\mathbb{E}\left[ \bar{G}_d^{(t-k)} \frac{g_{i,d}^{(t-k)}}{\sqrt{\epsilon + v_{i,d}^{(t)}}} \right] = \mathbb{E}\left[ \bar{G}g \left( \frac{1}{\sqrt{\epsilon + v}} \pm \frac{1}{\sqrt{\epsilon + \tilde{v}}} \right) \right]$$

$$= \mathbb{E}\left[ \mathbb{E}_{t-k-1} \left[ \frac{\bar{G}g}{\sqrt{\epsilon + \tilde{v}}} \right] + \bar{G}g \frac{r^2 - \delta^2}{\sqrt{\epsilon + v}\sqrt{\epsilon + \tilde{v}}(\sqrt{\epsilon + v} + \sqrt{\epsilon + \tilde{v}})} \right] \quad (63)$$

$$= \mathbb{E}\left[ \underbrace{\frac{\bar{G}(G \pm \bar{G})}{\sqrt{\epsilon + \tilde{v}}}}_{C} + \underbrace{\bar{G}g \frac{r^2 - \delta^2}{\sqrt{\epsilon + v}\sqrt{\epsilon + \tilde{v}}(\sqrt{\epsilon + v} + \sqrt{\epsilon + \tilde{v}})}}_{H} \right].$$

Now focus on $C$ in (63). By applying the Young's inequality equation 9 and letting

$$\gamma = \frac{1}{2\sqrt{\epsilon + \tilde{v}}}, \quad a = \bar{G}, \quad b = \frac{G - \bar{G}}{\sqrt{\epsilon + \tilde{v}}},$$

then we have

$$\mathbb{E}[C] \geq \frac{3}{4}\mathbb{E}\left[ \frac{\bar{G}^2}{\sqrt{\epsilon + \tilde{v}}} \right] - \mathbb{E}\left[ \frac{(G - \bar{G})^2}{\sqrt{\epsilon + \tilde{v}}} \right]. \quad (64)$$

Now focus on $H$ in (63). Given the fact that $\sqrt{\epsilon + v} > 0$ and $\sqrt{\epsilon + \tilde{v}} > 0$, we have

$$\sqrt{\epsilon + v} + \sqrt{\epsilon + \tilde{v}} > \sqrt{\epsilon + v} \quad \text{and} \quad \sqrt{\epsilon + v} + \sqrt{\epsilon + \tilde{v}} > \sqrt{\epsilon + \tilde{v}}, \quad (65)$$

and given that $|r^2 - \delta^2| \leq r^2 + \delta^2$, we have

$$|H| \leq |\bar{G}g| \frac{r^2}{\sqrt{\epsilon + v}(\epsilon + \tilde{v})} + |\bar{G}g| \frac{\delta^2}{\sqrt{\epsilon + \tilde{v}}(\epsilon + v)}. \quad (66)$$

By applying the Young's inequality equation 9 to (66) and letting

$$\gamma = \frac{\sqrt{1 - \beta_2}\sqrt{\epsilon + \tilde{v}}}{2}, \quad a = \frac{|\bar{G}|}{\sqrt{\epsilon + \tilde{v}}}, \quad b = \frac{|g|r^2}{\sqrt{\epsilon + \tilde{v}}\sqrt{\epsilon + v}},$$

we have

$$|\bar{G}g| \frac{r^2}{\sqrt{\epsilon + v}(\epsilon + \tilde{v})} \leq \frac{\sqrt{1 - \beta_2}\bar{G}^2}{4\sqrt{\epsilon + \tilde{v}}} + \frac{g^2 r^4}{\sqrt{1 - \beta_2}(\epsilon + \tilde{v})^{3/2}(\epsilon + v)}$$

$$\leq \frac{\bar{G}^2}{4\sqrt{\epsilon + \tilde{v}}} + \frac{g^2 r^4}{\sqrt{1 - \beta_2}(\epsilon + \tilde{v})^{3/2}(\epsilon + v)}. \quad (67)$$

Given that $\epsilon + \tilde{v} \geq r^2$ and taking the conditional expectation, we have

$$\mathbb{E}_{t-k-1}\left[ |\bar{G}g| \frac{r^2}{\sqrt{\epsilon + v}(\epsilon + \tilde{v})} \right] \leq \frac{\bar{G}^2}{4\sqrt{\epsilon + \tilde{v}}} + \frac{r^2}{\sqrt{1 - \beta_2}\sqrt{\epsilon + \tilde{v}}}\mathbb{E}_{t-k-1}\left[ \frac{g^2}{\epsilon + v} \right]. \quad (68)$$

Now focus on the second term in (66). By applying the Young's inequality equation 9 and letting

$$\gamma = \frac{\sqrt{1 - \beta_1}\sqrt{\epsilon + \tilde{v}}}{2r^2}, \quad a = \frac{|\bar{G}\delta|}{\sqrt{\epsilon + \tilde{v}}}, \quad b = \frac{|g\delta|}{\epsilon + v},$$

we have

$$
\begin{aligned}
|\bar{G}g|\frac{\delta^2}{\sqrt{\epsilon+\tilde{v}}(\epsilon+v)} &\leq \frac{\bar{G}^2\delta^2\sqrt{1-\beta_2}}{4r^2\sqrt{\epsilon+\tilde{v}}} + \frac{g^2\delta^2 r^2}{\sqrt{1-\beta_2}\sqrt{\epsilon+\tilde{v}}(\epsilon+v)^2} \\
&\leq \frac{\bar{G}^2\delta^2}{4r^2\sqrt{\epsilon+\tilde{v}}} + \frac{g^2\delta^2 r^2}{\sqrt{1-\beta_2}\sqrt{\epsilon+\tilde{v}}(\epsilon+v)^2}.
\end{aligned}
\tag{69}
$$

Given that $\epsilon+v \geq \delta^2$ and $\mathbb{E}_{t-k-1}\left[\frac{\delta^2}{r^2}\right]=1$, after taking the conditional expectation, we have

$$
\mathbb{E}_{t-k-1}\left[|\bar{G}g|\frac{\delta^2}{\sqrt{\epsilon+\tilde{v}}(\epsilon+v)}\right] \leq \frac{\bar{G}^2}{4\sqrt{\epsilon+\tilde{v}}} + \frac{r^2}{\sqrt{1-\beta_2}\sqrt{\epsilon+\tilde{v}}}\mathbb{E}_{t-k-1}\left[\frac{g^2}{\epsilon+v}\right].
\tag{70}
$$

By plugging (70) and (68) back to (66), we have

$$
\mathbb{E}_{t-k-1}\left[|H|\right] \leq \frac{\bar{G}^2}{2\sqrt{\epsilon+\tilde{v}}} + \frac{2r^2}{\sqrt{1-\beta_2}\sqrt{\epsilon+\tilde{v}}}\mathbb{E}_{t-k-1}\left[\frac{g^2}{\epsilon+v}\right].
\tag{71}
$$

Given that $r \leq \sqrt{\epsilon+\tilde{v}}$, and $r \leq \sqrt{k+1}R$ by Assumption 2, we have

$$
\mathbb{E}_{t-k-1}\left[|H|\right] \leq \frac{\bar{G}^2}{2\sqrt{\epsilon+\tilde{v}}} + \frac{2R\sqrt{k+1}}{\sqrt{1-\beta_2}}\mathbb{E}_{t-k-1}\left[\frac{g^2}{\epsilon+v}\right].
\tag{72}
$$

By recovering the temporarily introduced notation, we have

$$
\mathbb{E}_{t-k-1}\left[|H|\right] \leq \frac{\left(\bar{G}_d^{(t-k)}\right)^2}{2\sqrt{\epsilon+\tilde{v}_{i,d}^{(t,k)}}} + \frac{2R\sqrt{k+1}}{\sqrt{1-\beta_2}}\mathbb{E}_{t-k-1}\left[\frac{\left(g_{i,d}^{(t-k)}\right)^2}{\epsilon+v_{i,d}^{(t)}}\right].
\tag{73}
$$

Given that $\epsilon+v_{i,d}^{(t)} \geq \epsilon+\beta_2^k v_{i,d}^{(t-k)} \geq \beta_2^k(\epsilon+v_{i,d}^{(t-k)})$, and by taking the complete expectation, we have

$$
\mathbb{E}[|H|] \leq \frac{1}{2}\mathbb{E}\left[\frac{\left(\bar{G}_d^{(t-k)}\right)^2}{\sqrt{\epsilon+\tilde{v}_{i,d}^{(t,k)}}}\right] + \frac{2R\sqrt{k+1}}{\beta_2^k\sqrt{1-\beta_2}}\mathbb{E}\left[\frac{\left(g_{i,d}^{(t-k)}\right)^2}{\epsilon+v_{i,d}^{(t-k)}}\right].
\tag{74}
$$

By plugging (74) and (64) back to (63), we have

$$
\begin{aligned}
\mathbb{E}\left[\bar{G}_d^{(t-k)}\frac{g_{i,d}^{(t-k)}}{\sqrt{\epsilon+v_{i,d}^{(t)}}}\right] \geq{} &\frac{1}{4}\mathbb{E}\left[\frac{\left(\bar{G}_d^{(t-k)}\right)^2}{\sqrt{\epsilon+\tilde{v}_{i,d}^{(t,k)}}}\right] - \frac{2R\sqrt{k+1}}{\beta_2^k\sqrt{1-\beta_2}}\mathbb{E}\left[\frac{\left(g_{i,d}^{(t-k)}\right)^2}{\epsilon+v_{i,d}^{(t-k)}}\right] \\
&- \mathbb{E}\left[\frac{\left(G_{i,d}^{(t-k)}-\bar{G}_d^{(t-k)}\right)^2}{\sqrt{\epsilon+\tilde{v}_{i,d}^{(t,k)}}}\right].
\end{aligned}
\tag{75}
$$

By plugging (75) back to $A$ in (58), we have

$$
\begin{aligned}
\mathbb{E}[A] \geq & \frac{1}{4} \sum_{d=1}^{D} \sum_{k=0}^{t-1} \beta_1^k \sum_{i=1}^{N} \mathbb{E}\left[\frac{\left(\bar{G}_d^{(t-k)}\right)^2}{\sqrt{\epsilon + \tilde{v}_{i,d}^{(t,k)}}}\right] - \frac{2R}{\sqrt{1-\beta_2}} \sum_{d=1}^{D} \sum_{k=0}^{t-1} \beta_1^k \sum_{i=1}^{N} \frac{\sqrt{k+1}}{\beta_2^k} \mathbb{E}\left[\frac{\left(g_{i,d}^{(t-k)}\right)^2}{\epsilon + v_{i,d}^{(t-k)}}\right] \\
& - \sum_{d=1}^{D} \sum_{k=0}^{t-1} \beta_1^k \sum_{i=1}^{N} \mathbb{E}\left[\frac{\left(G_{i,d}^{(t-k)} - \bar{G}_d^{(t-k)}\right)^2}{\sqrt{\epsilon + \tilde{v}_{i,d}^{(t,k)}}}\right] \\
= & \frac{1}{4} \sum_{d=1}^{D} \sum_{k=0}^{t-1} \beta_1^k \sum_{i=1}^{N} \mathbb{E}\left[\frac{\left(\bar{G}_d^{(t-k)}\right)^2}{\sqrt{\epsilon + \tilde{v}_{i,d}^{(t,k)}}}\right] - \frac{2R}{\sqrt{1-\beta_2}} \sum_{k=0}^{t-1} \left(\frac{\beta_1}{\beta_2}\right)^k \sqrt{k+1} \sum_{i=1}^{N} \mathbb{E}\left[\left\|U_i^{(t-k)}\right\|_2^2\right] \\
& - \sum_{d=1}^{D} \sum_{k=0}^{t-1} \beta_1^k \sum_{i=1}^{N} \mathbb{E}\left[\frac{\left(G_{i,d}^{(t-k)} - \bar{G}_d^{(t-k)}\right)^2}{\sqrt{\epsilon + \tilde{v}_{i,d}^{(t,k)}}}\right].
\end{aligned}
\tag{76}
$$

By plugging (76) and (62) back to (58), we have

$$
\begin{aligned}
\kappa \geq & \frac{1}{4} \sum_{d=1}^{D} \sum_{k=0}^{t-1} \beta_1^k \sum_{i=1}^{N} \mathbb{E}\left[\frac{\left(\bar{G}_d^{(t-k)}\right)^2}{\sqrt{\epsilon + \tilde{v}_{i,d}^{(t,k)}}}\right] - \sum_{d=1}^{D} \sum_{k=0}^{t-1} \beta_1^k \sum_{i=1}^{N} \mathbb{E}\left[\frac{\left(G_{i,d}^{(t-k)} - \bar{G}_d^{(t-k)}\right)^2}{\sqrt{\epsilon + \tilde{v}_{i,d}^{(t,k)}}}\right] \\
& - \frac{3R}{\sqrt{1-\beta_2}} \sum_{k=0}^{t-1} \left(\frac{\beta_1}{\beta_2}\right)^k \sqrt{k+1} \sum_{i=1}^{N} \mathbb{E}\left[\left\|U_i^{(t-k)}\right\|_2^2\right] - \frac{\alpha_t^2 L^2 \sqrt{1-\beta_1}}{4R} \sum_{i=1}^{N} \sum_{l=1}^{t-1} \mathbb{E}\left[\left\|u_i^{(t-l)}\right\|_2^2\right] \sum_{k=l}^{t-1} \beta_1^k \sqrt{k},
\end{aligned}
\tag{77}
$$

which completes the proof of Lemma 4. $\qquad\square$

**Lemma 5** (sum of ratios of the square of a decayed sum and a decayed sum of square)  Given $0 < \beta_1 < \beta_2 \leq 1$ and a sequence of real number $(a_n)_{n\in\mathbb{N}+}$. We define $b_n = \sum_{j=1}^{n} \beta_2^{n-j} a_j^2$ and $c_n = \sum_{j=1}^{n} \beta_1^{n-j} a_j$. Then we have

$$
\sum_{j=1}^{n} \frac{c_j^2}{\epsilon + b_j} \leq \frac{1}{(1-\beta_1)(1-\beta_1/\beta_2)} \left[\ln\left(1 + \frac{b_n}{\epsilon}\right) - n\ln(\beta_2)\right].
\tag{78}
$$

*Proof.* Refer to A.2 in Défossez et al. (2020). $\qquad\square$

**Lemma 6** (sum of a geometric term times a square root)  Given $0 < a < 1$ and $Q \in \mathbb{N}$, we have

$$
\sum_{q=0}^{Q-1} a^q \sqrt{q+1} \leq \frac{1}{1-a}\left(1 + \frac{\sqrt{\pi}}{2\sqrt{-\ln(a)}}\right) \leq \frac{2}{(1-a)^{3/2}}.
\tag{79}
$$

*Proof.* Refer to A.3 in Défossez et al. (2020). $\qquad\square$

**Lemma 7** (sum of a geometric term times a square root)  Given $0 < a < 1$ and $Q \in \mathbb{N}$, we have

$$
\sum_{q=0}^{Q-1} a^q \sqrt{q}(q+1) \leq \frac{4a}{(1-a)^{5/2}}.
\tag{80}
$$

*Proof.* Refer to A.4 in Défossez et al. (2020). $\qquad\square$

**Lemma 8** (with heavy-ball momentum, averaged consensus error is bounded if sampling from i.i.d. data distribution)  Under Assumptions 1–3, if $0 < \beta_1 < \beta_2 < 1$, we have

$$\frac{1}{T}\sum_{t=1}^{T}\sum_{i=1}^{N}\sum_{d=1}^{D}\left[G_{i,d}^{(t)} - \bar{G}_d^{(t)}\right]^2 \leq \frac{2(1+\lambda^2)(1-\beta_1)\alpha^2 NL^2 D}{(1-\lambda^2)^2(1-\beta_2)(1-\beta_1/\beta_2)}\left[\frac{1}{T}\ln\left(1+\frac{R^2}{(1-\beta_2)\epsilon}\right) - \ln(\beta_2)\right]. \tag{81}$$

*Proof.* Proof of Lemma 8 is similar to Lemma 3.

Since the heavy-ball momentum is introduced, the definitions of $\mathbf{e}_d^{(t)}$ and $\bar{\mathbf{e}}_d^{(t)}$ are changed to

$$\mathbf{e}_d^{(t)} = -\alpha_t\left[\frac{m_{1,d}^{(t)}}{\sqrt{\epsilon + v_{1,d}^{(t)}}}, \cdots, \frac{m_{N,d}^{(t)}}{\sqrt{\epsilon + v_{N,d}^{(t)}}}\right] \in \mathbb{R}^N,$$

$$\bar{\mathbf{e}}_d^{(t)} = -\frac{\alpha_t}{N}\left[\sum_{i=1}^{N}\frac{m_{i,d}^{(t)}}{\sqrt{\epsilon + v_{i,d}^{(t)}}}, \cdots, \sum_{i=1}^{N}\frac{m_{i,d}^{(t)}}{\sqrt{\epsilon + v_{i,d}^{(t)}}}\right] \in \mathbb{R}^N.$$

The proof is same as Lemma 3 until (36). Now we have

$$\sum_{t=1}^{T}\left\|\mathbf{x}_d^{(t-1)} - \bar{\mathbf{x}}_d^{(t-1)}\right\|_2^2 \leq \frac{2(1+\lambda^2)}{(1-\lambda^2)^2}\sum_{t=1}^{T}\left\|\mathbf{e}_d^{(t)}\right\|_2^2. \tag{82}$$

By applying Lemma 5, and given the fact that $\alpha_T \leq \alpha\frac{1-\beta_1}{\sqrt{1-\beta_2}}$, we have

$$\sum_{t=1}^{T}\left\|\mathbf{e}_d^{(t)}\right\|_2^2 = \sum_{i=1}^{N}\sum_{t=1}^{T}\alpha_t^2\frac{\left(m_{i,d}^{(t)}\right)^2}{\epsilon + v_{i,d}^{(t)}}$$

$$\leq \frac{(1-\beta_1)^2\alpha^2}{1-\beta_2}\sum_{i=1}^{N}\frac{1}{(1-\beta_1)(1-\beta_1/\beta_2)}\left[\ln\left(1+\frac{v_{i,d}^{(T)}}{\epsilon}\right) - T\ln(\beta_2)\right]. \tag{83}$$

By using Assumption 2, it is given that $v_{i,d}^{(T)} \leq R^2/(1-\beta_2)$. Then we have

$$\sum_{t=1}^{T}\left\|\mathbf{e}_d^{(t)}\right\|_2^2 \leq \frac{(1-\beta_1)\alpha^2 N}{(1-\beta_2)(1-\beta_1/\beta_2)}\left[\ln\left(1+\frac{R^2}{(1-\beta_2)\epsilon}\right) - T\ln(\beta_2)\right]. \tag{84}$$

By plugging (84) back to (82) and (39), we have

$$\frac{1}{T}\sum_{t=1}^{T}\sum_{i=1}^{N}\sum_{d=1}^{D}\left[G_{i,d}^{(t)} - \bar{G}_d^{(t)}\right]^2 \leq \frac{2(1+\lambda^2)(1-\beta_1)\alpha^2 NL^2 D}{(1-\lambda^2)^2(1-\beta_2)(1-\beta_1/\beta_2)}\left[\frac{1}{T}\ln\left(1+\frac{R^2}{(1-\beta_2)\epsilon}\right) - \ln(\beta_2)\right], \tag{85}$$

which completes the proof of Lemma 8. $\qquad\square$

### B.4.3  MAIN PROOF OF THEOREM 4.1

Firstly introduce the notation $\Omega_t$ as

$$\Omega_t = \sqrt{\sum_{j=0}^{t-1}\beta_2^j} \quad\Rightarrow\quad \alpha_t = (1-\beta_1)\Omega_t\alpha.$$

By using Assumption 1, we have

$$F\left(\bar{x}^{(t)}\right) \leq F\left(\bar{x}^{(t-1)}\right) + \nabla F\left(\bar{x}^{(t-1)}\right)^\top\left(\bar{x}^{(t)} - \bar{x}^{(t-1)}\right) + \frac{L}{2}\left\|\bar{x}^{(t)} - \bar{x}^{(t-1)}\right\|_2^2. \tag{86}$$

By using (10) and taking the complete expectation, we have

$$
\mathbb{E}\left[F\left(\bar{x}^{(t)}\right)\right] \leq \mathbb{E}\left[F\left(\bar{x}^{(t-1)}\right)\right] - \frac{\alpha_t}{N}\mathbb{E}\left[\sum_{d=1}^{D}\nabla_d F\left(\bar{x}^{(t-1)}\right)\sum_{i=1}^{N}\frac{m_{i,d}^{(t)}}{\sqrt{\epsilon + v_{i,d}^{(t)}}}\right] + \frac{L\alpha_t^2}{2N^2}\mathbb{E}\left[\left\|\sum_{i=1}^{N}u_i^{(t)}\right\|_2^2\right]
$$

$$
\leq \mathbb{E}\left[F\left(\bar{x}^{(t-1)}\right)\right] - \frac{\alpha_t}{N}\mathbb{E}\left[\sum_{d=1}^{D}\nabla_d F\left(\bar{x}^{(t-1)}\right)\sum_{i=1}^{N}\frac{m_{i,d}^{(t)}}{\sqrt{\epsilon + v_{i,d}^{(t)}}}\right] + \frac{L\alpha_t^2}{2N}\sum_{i=1}^{N}\mathbb{E}\left[\left\|u_i^{(t)}\right\|_2^2\right].
$$

$$(87)$$

From Assumption 2, we have $\sqrt{\epsilon + \tilde{v}_{i,d}^{(t,k)}} \leq R\Omega_t$, and by applying Lemma 4, we have

$$
\mathbb{E}\left[F\left(\bar{x}^{(t)}\right)\right] \leq \mathbb{E}\left[F\left(\bar{x}^{(t-1)}\right)\right] - \frac{\alpha_t}{4R\Omega_t}\sum_{k=0}^{t-1}\beta_1^k\mathbb{E}\left[\left\|\bar{G}^{(t-k)}\right\|_2^2\right]
$$

$$
+ \frac{3R\alpha_t}{N\sqrt{1-\beta_1}}\sum_{k=0}^{t-1}\left(\frac{\beta_1}{\beta_2}\right)^k\sqrt{k+1}\sum_{i=1}^{N}\mathbb{E}\left[\left\|U_i^{(t-k)}\right\|_2^2\right]
$$

$$
+ \frac{\alpha_t^3 L^2\sqrt{1-\beta_1}}{4RN}\sum_{i=1}^{N}\sum_{l=1}^{t-1}\mathbb{E}\left[\left\|u_i^{(t-l)}\right\|_2^2\right]\sum_{k=l}^{t-1}\beta_1^k\sqrt{k} \tag{88}
$$

$$
+ \frac{\alpha_t}{N}\sum_{d=1}^{D}\sum_{k=0}^{t-1}\beta_1^k\sum_{i=1}^{N}\mathbb{E}\left[\frac{\left(G_{i,d}^{(t-k)} - \bar{G}_d^{(t-k)}\right)^2}{\sqrt{\epsilon + \tilde{v}_{i,d}^{(t,k)}}}\right] + \frac{L\alpha_t^2}{2N}\sum_{i=1}^{N}\mathbb{E}\left[\left\|u_i^{(t)}\right\|_2^2\right].
$$

By summing over all iterations $t \in [T]$, moving terms, and given that $\alpha_t$ is set to be non-decreasing, we have

$$
\underbrace{\frac{1}{4R}\sum_{t=1}^{T}\frac{\alpha_t}{\Omega_t}\sum_{k=0}^{t-1}\beta_1^k\mathbb{E}\left[\left\|\bar{G}_d^{(t-k)}\right\|_2^2\right]}_{A} \leq \mathbb{E}\left[F\left(\bar{x}^{(0)}\right)\right] - F_* + \underbrace{\frac{L\alpha_T^2}{2N}\sum_{i=1}^{N}\sum_{t=1}^{T}\mathbb{E}\left[\left\|u_i^{(t)}\right\|_2^2\right]}_{B}
$$

$$
+ \underbrace{\frac{\alpha_T^3 L^2\sqrt{1-\beta_1}}{4RN}\sum_{i=1}^{N}\sum_{t=1}^{T}\sum_{l=1}^{t-1}\mathbb{E}\left[\left\|u_i^{(t-l)}\right\|_2^2\right]\sum_{k=l}^{t-1}\beta_1^k\sqrt{k}}_{C}
$$

$$
+ \underbrace{\frac{3R\alpha_T}{N\sqrt{1-\beta_1}}\sum_{t=1}^{T}\sum_{k=0}^{t-1}\left(\frac{\beta_1}{\beta_2}\right)^k\sqrt{k+1}\sum_{i=1}^{N}\mathbb{E}\left[\left\|U_i^{(t-k)}\right\|_2^2\right]}_{H}
$$

$$
+ \underbrace{\frac{\alpha_T}{N}\sum_{d=1}^{D}\sum_{k=0}^{t-1}\beta_1^k\sum_{i=1}^{N}\sum_{t=1}^{T}\mathbb{E}\left[\frac{\left(G_{i,d}^{(t-k)} - \bar{G}_d^{(t-k)}\right)^2}{\sqrt{\epsilon + \tilde{v}_{i,d}^{(t,k)}}}\right]}_{I}.
$$

$$(89)$$

Now focus on $B$ in (89). By using Lemma 5, and given that $v_{i,d}^{(t)} \leq R^2/(1-\beta_2)$ (from Assumption 2) and $\alpha_T \leq \alpha\frac{1-\beta_1}{\sqrt{1-\beta_2}}$, we have

$$
B \leq \frac{DL\alpha^2(1-\beta_1)}{2(1-\beta_2)(1-\beta_1/\beta_2)}\left[\ln\left(1 + \frac{R^2}{(1-\beta_2)\epsilon}\right) - T\ln(\beta_2)\right]. \tag{90}
$$

Now focus in $C$ in (89). By introducing the index $j = t - l$, we have

$$
C = \frac{\alpha_T^3 L^2\sqrt{1-\beta_1}}{4RN}\sum_{i=1}^{N}\sum_{j=1}^{T}\mathbb{E}\left[\left\|u_i^{(j)}\right\|_2^2\right]\sum_{k=0}^{T-1}\beta_1^k\sqrt{k}(k+1). \tag{91}
$$

By applying Lemma 7 and then Lemma 5 (similar to (90)), we have

$$
\begin{aligned}
C &\leq \frac{\beta_1 \alpha_T^3 L^2}{(1-\beta_1)^2 RN} \sum_{i=1}^{N} \sum_{j=1}^{T} \mathbb{E}\left[\left\|u_i^{(j)}\right\|_2^2\right] \\
&\leq \frac{\beta_1 \alpha_T^3 L^2 D}{(1-\beta_1)^3(1-\beta_1/\beta_2)R}\left[\ln\left(1+\frac{R^2}{(1-\beta_2)\epsilon}\right) - T\ln(\beta_2)\right] \\
&\leq \frac{\beta_1 \alpha^3 L^2 D}{(1-\beta_1/\beta_2)(1-\beta_2)^{3/2} R}\left[\ln\left(1+\frac{R^2}{(1-\beta_2)\epsilon}\right) - T\ln(\beta_2)\right].
\end{aligned}
\tag{92}
$$

Now focus on $H$ in (89). By introducing the index $j = t - k$ (similar to (91)) and applying Lemma 6 and then Lemma 2, we have

$$
\begin{aligned}
H &= \frac{3R\alpha_T}{N\sqrt{1-\beta_1}} \sum_{i=1}^{N} \sum_{j=1}^{T} \mathbb{E}\left[\left\|U_i^{(j)}\right\|_2^2\right] \sum_{t=j}^{T}\left(\frac{\beta_1}{\beta_2}\right)^{t-j}\sqrt{1+t-j} \\
&\leq \frac{6R\alpha\sqrt{1-\beta_1}}{(1-\beta_1/\beta_2)^{3/2}\sqrt{1-\beta_2}}\left[\ln\left(1+\frac{R^2}{(1-\beta_2)\epsilon}\right) - T\ln(\beta_2)\right].
\end{aligned}
\tag{93}
$$

Now focus on $I$ in (89). Given that $\sqrt{\epsilon + \tilde{v}_{i,d}^{(t,k)}} \geq \sqrt{\epsilon}$, and by introducing index $j = t - k$, we have

$$
\begin{aligned}
I &\leq \frac{\alpha_T}{N\sqrt{\epsilon}} \sum_{i=1}^{N} \sum_{d=1}^{D} \sum_{j=1}^{T} \mathbb{E}\left[\left(G_{i,d}^{(j)} - \bar{G}_d^{(j)}\right)^2\right] \sum_{k=0}^{T-j} \beta_1^k \\
&\leq \frac{\alpha}{N\sqrt{\epsilon}\sqrt{1-\beta_2}} \sum_{i=1}^{N} \sum_{d=1}^{D} \sum_{j=1}^{T} \mathbb{E}\left[\left(G_{i,d}^{(j)} - \bar{G}_d^{(j)}\right)^2\right].
\end{aligned}
\tag{94}
$$

By applying Lemma 8, we have

$$
I \leq \frac{2(1+\lambda^2)\sqrt{1-\beta_1}\alpha^3 L^2 D}{(1-\lambda^2)^2(1-\beta_2)(1-\beta_1/\beta_2)\sqrt{\epsilon}}\left[\ln\left(1+\frac{R^2}{(1-\beta_2)\epsilon}\right) - T\ln(\beta_2)\right].
\tag{95}
$$

Now focus on $A$ in (89). By introducing the index $j = t - k$, we have

$$
\begin{aligned}
A &= \frac{1}{4R} \sum_{t=1}^{T} \frac{\alpha_t}{\Omega_t} \sum_{k=0}^{t-1} \beta_1^k \mathbb{E}\left[\left\|\bar{G}_d^{(t-k)}\right\|_2^2\right] \\
&= \frac{\alpha(1-\beta_1)}{4R} \sum_{j=1}^{T} \mathbb{E}\left[\left\|\bar{G}_d^{(j)}\right\|_2^2\right] \sum_{t=j}^{T} \beta_1^{t-j} \\
&= \frac{\alpha}{4R} \sum_{j=1}^{T}(1-\beta_1^{T-j}) \mathbb{E}\left[\left\|\bar{G}_d^{(j)}\right\|_2^2\right] \\
&= \frac{\alpha}{4R} \sum_{j=1}^{T}(1-\beta_1^{T-j}) \mathbb{E}\left[\left\|\nabla F(\bar{x}^{(j)})\right\|_2^2\right].
\end{aligned}
\tag{96}
$$

For any $T \in \mathbb{N}^+$, we define $\tau$ a random index with value $\{0, \cdots, T-1\}$ with distribution

$$
\forall j \in \mathbb{N}, j < T, \quad \mathbb{P}[\tau = j] \propto 1 - \beta_1^{T-j}.
\tag{97}
$$

Then we have

$$
\sum_{j=0}^{T-1}(1-\beta_1^{T-j}) = T - \beta_1 \frac{1-\beta_1^T}{1-\beta_1} \geq T - \frac{\beta_1}{1-\beta_1}.
\tag{98}
$$

By introducing $\tilde{T} = T - \frac{\beta_1}{1-\beta_1}$, we have

$$A \geq \frac{\alpha\tilde{T}}{4R}\mathbb{E}\left[\left\|\nabla F(\bar{x}^{(\tau)})\right\|_2^2\right]. \tag{99}$$

By putting together (99), (95), (93), (92), and (90) into (89), we have

$$\mathbb{E}\left[\left\|\nabla F(\bar{x}^{(\tau)})\right\|_2^2\right] \leq \frac{4R}{\alpha\tilde{T}}\left(F(\bar{x}^{(0)}) - F_*\right) + E\left[\frac{1}{\tilde{T}}\ln\left(1 + \frac{R}{\epsilon(1-\beta_2)}\right) - \frac{T}{\tilde{T}}\ln(\beta_2)\right], \tag{100}$$

where

$$E := \frac{2DLR(1-\beta_1)\alpha}{(1-\beta_2)(1-\beta_1/\beta_2)} + \frac{4L^2D\beta_1\alpha^2}{(1-\beta_1/\beta_2)(1-\beta_2)^{3/2}} + \frac{24DR^2\sqrt{1-\beta_1}}{(1-\beta_1/\beta_2)^{3/2}\sqrt{1-\beta_2}}$$
$$+ \frac{8(1+\lambda^2)RL^2D\sqrt{1-\beta_1}\alpha^2}{(1-\lambda^2)^2(1-\beta_2)(1-\beta_1/\beta_2)\sqrt{\epsilon}},$$

which completes the proof of Theorem 4.1.

## C    DECENTRALIZED ACCUMULATED ADAM

Algorithm 4 describes the decentralized accumulated Adam. As is displayed in Section 5, the decentralized accumulated Adam demonstrates competitive practical performances for several training tasks compared to alternative methods.

---

**Algorithm 4** Decentralized Accumulated Adam on worker $i$

---

1:  $s \leftarrow$ # of iterations in one accumulation loop (for example, 4); $T$ such that $(T \mod s) = 0$
2:  $\hat{m}_i^{(0)}, \hat{v}_i^{(0)} \leftarrow 0; x_i^{(0)} \leftarrow x^0; b_i \leftarrow 0$
3:  **for** $t = 1, 2, \ldots, T$ **do**
4:      $\hat{t} \leftarrow \lceil t/s \rceil$
5:      $g_i^{(t)} \leftarrow \nabla\ell(x_i^{(t-1)}; \xi_i^{(t)})$
6:      $m_i^{(t)} \leftarrow \beta_1\hat{m}_i^{(\hat{t}-1)} + (1-\beta_1)g_i^{(t)}$
7:      $v_i^{(t)} \leftarrow \beta_2\hat{v}_i^{(\hat{t}-1)} + (1-\beta_2)\left[g_i^{(t)}\right]^2$
8:      $x_i^{(t)} \leftarrow -\alpha\frac{m_i^{(t)}/(1-\beta_1^{\hat{t}})}{\sqrt{v_i^{(t)}/(1-\beta_2^{\hat{t}})}+\epsilon} + \sum_{j\in\mathcal{N}_i^{(t)}} w_{ij}^{(t)}x_j^{(t-1)}$
9:      $b_i \leftarrow b_i + \frac{1}{s}g_i^{(t)}$
10:     **if** $t \mod s == 0$ **then**
11:         $\hat{m}_i^{(\hat{t})} \leftarrow \beta_1\hat{m}_i^{(\hat{t}-1)} + (1-\beta_1)b_i$
12:         $\hat{v}_i^{(\hat{t})} \leftarrow \beta_1\hat{v}_i^{(\hat{t}-1)} + (1-\beta_1)[b_i]^2$
13:         $b_i \leftarrow 0$
14:     **end if**
15: **end for**

---

To understand Algorithm 4, let us introduce $\hat{t}$ to divide the sequence of iteration index $t$ into groups of every $s$ iterations:

$$t \in \underbrace{1, \ldots, s}, \quad \underbrace{s+1, \ldots, 2s}, \quad \ldots, \quad \underbrace{\left(\frac{T}{s}-1\right)s+1, \ldots, \frac{T}{s}s}$$
$$\hat{t} \in \qquad 1, \qquad\qquad 2, \qquad \ldots, \qquad\qquad\qquad \frac{T}{s}$$

For a given iteration index $t$, it falls in the $\hat{t} = \lceil t/s \rceil$-th group, so that $m_i^{(t)}$ in Algorithm 4 can be equivalently expressed as

$$m_i^{(t)} = (1-\beta_1)\left(\beta_1^0 g_i^{(t)} + \beta_1^1 G_i^{(\hat{t}-1)} + \cdots + \beta_1^{\hat{t}-1} G_i^{(1)}\right), \tag{101}$$

where $G_i^{(\hat{t})} := \frac{1}{s}\left(g_i^{((\hat{t}-1)s+1)} + \cdots + g_i^{(\hat{t}s)}\right)$. The update of $v_i^{(t)}$ is similar.

Algorithm 4 uses accumulated stochastic gradients to construct the first-order momentum $m_i^{(t)}$ and second-momentum $v_i^{(t)}$, which reduces the variance of the stochastic gradients while does not reduce the number of model updates, i.e., within $T$ stochastic gradient computations at worker $i$, the model $x_i^{(t)}$ is updated for $T$ instead of $T/s$ times. To see these, assume the empirically optimal batch size in a single-machine environment is $B$, and we call a batch of size $B$ a global mini-batch. The variance of the global mini-batch gradients for the global gradient $\nabla F$ is $\Sigma^2$. Under a decentralized setup, these $B$ batches are evenly distributed to $N$ workers, with each worker receiving $B/N$ samples, which we call a local mini-batch. Then, the variance of each worker's local mini-batch gradient for the global gradient $\nabla F$ increases to $N\Sigma^2$. This increased variance exacerbates the consensus errors among workers. To compensate for the increased variance, a straightforward idea is to have each worker accumulate $s$ (proportional to $N$) local mini-batch random gradients before performing an update, which improves the variance to $\frac{N}{s}\Sigma^2$. However, this accumulation leads to a reduction in the number of model update iterations, i.e., within $T$ stochastic gradient computations, the number of model updates is $T/s$ instead of $T$. With a novel accumulation technique, this issue is avoided in Algorithm 4. As described in Algorithm 4, when a worker computes a local mini-batch $g_i^{(t)}$, it assigns it a weight of 1, adds it to the exponential averaging of all historical $G^{(1)}$ to $G^{(\hat{t}-1)}$ to generate $m_i^{(t)}$ and $v_i^{(t)}$, and updates the local model once. In this way, the number of updates is equal to $T$ instead of $T/s$.

Algorithm 4 has close connections with existing well-known algorithms. On one hand, it performs exponential moving averaging over each group of $s$ stochastic gradients, which yields a higher weight of the oldest information compared to the vanilla Adam ($\beta_1^{\hat{t}-1}$ versus $\beta_1^{t-1}$; see equation 101). This technique is similar to SlowMo Wang et al. (2019) that has been shown to be effective for DNN training. On the other hand, Algorithm 4 updates the model upon receiving each $g_i^t$, which is similar to FedAvg Konečnỳ et al. (2016). It is worth noting that FedAvg does not communicate every iteration and may yield a lower communication cost compared to Algorithm 4. Whether this technique can further accelerate Algorithm 4 is a worthwhile research direction. We leave it as well as the convergence analysis of Algorithm 4 as future work.

Similar to the changes in AdamW Loshchilov (2017) compared with Adam, AccumAdam can also be similarly adapted to AccumAdamW.

## D   DISCUSSION AND LIMITATION

Our analysis of the runtime model in Section 3 and convergence guarantees in § 4 show how the communication topology affects both the per-iteration runtime and the convergence. A sparser topology shortens the per-iteration runtime but may result in a topology with slower mixing and worse training results. Consistent with Kong et al. (2021), we have found that for a limited number of workers (32 or less), we can obtain significant runtime improvements by sparsifying the communication without affecting the quality of the final model. For a network with a large number of workers, the sparse topology plays a more important role and we may need to study the runtime and optimization aspects separately. Although decentralized training may face challenges with a large number of workers ($> 64$), data parallelism is not the only approach for scaling up distributed training. For large language models, tensor parallelism and model parallelism are also crucial techniques, and the superiority of decentralized training in interleaving communication and computation can be a great complement to overall efficiency. In Shoeybi et al. (2019), for example, they used 64-way data parallel and 8-way model parallel to scale up the training to 512 GPUs. Additionally, in common machine learning workloads, too-small local batch size could lead to low utilization for a single GPU, and a 64-way data parallel is already sufficient for most tasks.

As for memory consumption, since the decentralized training has asynchronous communications, it is inevitable to allocate additional memory for the communication buffers to hide the time taken by communications. Similar implementations can be found in Assran et al. (2019); Wang et al. (2019). Even though the decentralized training allocates more memory, it is a trade-off between memory and per-iteration runtime if compared with All-Reduce training. Moreover, it should also be noted

that more memory is spent on the backward pass and the optimizer states (like Adam Kingma & Ba (2014)). The additional memory usage can be alleviated by orthogonal techniques like compression.

In our experiments, we used a time-varying mixing matrix, but for ease of convergence analysis, we used a fixed one. Techniques from some works for time-varying topologies Nedic et al. (2017); Saadatniaki et al. (2020); Metelev et al. (2023) can be applied here. Our accumulated Adam performs well in practice, but it lacks theoretical analysis. It could be finished by using the analysis techniques of the Adam method Zaheer et al. (2018) and decentralized algorithms Yuan et al. (2016); Zeng & Yin (2018).

