# OpenReview forum: "From Promise to Practice: Realizing High-performance Decentralized Training"
_ICLR.cc/2025/Conference — ICLR 2025 Poster_

### Official Review · Reviewer_LAtq · 2024-10-31

**Soundness:** 3
**Presentation:** 3
**Contribution:** 2
**Rating:** 6
**Confidence:** 3

**Summary:**

The paper explores decentralized training of deep neural networks (DNNs) as an alternative to synchronous data-parallel methods like All-Reduce, which are known for their scalability issues. The authors identify three key factors that can lead to speedups in decentralized training: overlapping communication and computation, respecting heterogeneous communication costs, and reducing sensitivity to varying computation times. They propose a runtime model to optimize decentralized training configurations and design a decentralized variant of the Adam optimizer, called DAdam, which supports overlapping communications and computations. Additionally, they introduce an accumulation technique to mitigate the high variance caused by small local batch sizes.

**Strengths:**

1. The authors provide a runtime model that quantifies key environmental parameters and estimates potential speedups, offering valuable insights into the conditions under which decentralized training is advantageous.
2. There is theoretical analysis to support the proposed algorithm.
3. The experiment consists of 64 GPUs, which is a suitable scale.
4. There is detailed hyper-parameter settings for reproducibility.

**Weaknesses:**

1. Some important related works in MLsys [1,2] are missed, in which the dynamic gossip communication topologies are also discussed, and the communication compression is utilized.
2. The novelty of the DAdam is limited. There is no improvements on the Adam itself. The proposed method can be seen as a system optimization instead of the optimizer itself. However, the systematic optimization including the overlapping between communication and computation is largely used in many ML systems [3,4].
3. While the paper presents extensive experiments, it lacks detailed ablation studies to isolate the impact of individual components of the proposed approach.
4. The tested topologies are limited.
5. For training GPT-2 on OpenWebText, it is unclear whether the final convergence of DAdam can match the All-Reduce Adam when training with more iterations. The convergence curves of training on OpenWebText are not provided.
6. There lacks enough details of how the decentralized training is implemented built upon PyTorch.


[1] GossipFL: A Decentralized Federated Learning Framework With Sparsified and Adaptive Communication. In TPDS 2022.
[2] Communication-efficient decentralized learning with sparsification and adaptive peer selection. In ICDCS 2020.
[3] Overlap Communication with Dependent Computation via Decomposition in Large Deep Learning Models. In ASPLOS 2022.
[4] Centauri: Enabling Efficient Scheduling for Communication-Computation Overlap in Large Model Training via Communication Partitioning. In ASPLOS 2024.

**Questions:**

Please refer to the weaknesses.

It would be better to provide training convergence curves of using DAdam and the All-Reduce Adam on OpenWebText.

Also, it is important to illustrate how the decentralized training is implemented, as the Pytorch and DeepSpeed have conducted many system optimizations on the All-Reduce, there should be some convincing illustrations to show that the implemented decentralized training can outperform the All-Reduce. Specifically, how the communication is launched, using torchrun or MPIRUN or something else? how the synchronization between different nodes is implemented, is there a distributed.barrier()? how the overlapping between communication and computation is implemented? How do you manage the communication topology? The source code with a document may be very helpful.

If these questions are addressed, I'd like to increase my score.

---

> ### Author Response · Authors · 2024-11-19
> **Response to Reviewer LAtq (Part 1/2)**
>
> We appreciate the time and effort that you invested in reviewing our paper and giving constructive feedback that will allow us to improve our paper further. Note that our reply is separated in two, to meet the character limit in OpenReview. Please also note that we have updated the supplementary material to include training curves for the GPT-2 experiments and provided more details on the implementation of the extension. We refer to this updated supplementary in our response below.
>
> **W1 (related work)**: Thank you for providing us with these references. These are really interesting papers that also focus on practical decentralized training. We would like to emphasize that the application scenario that they worked on and the problems that they tackled are different from ours. In [1,2], the workers are distributed in different cities, and the bandwidth of the interconnections is significantly limited. The compression and sparsification techniques demonstrate more impact in the low-bandwidth scenarios. In our case, as stated in the introduction and in the experimental setups, we focus on self-built servers or shared clusters which may not have high-end interconnections as those dedicated clusters built by big tech companies. Still, their bandwidths are at least 25Gbps, which is much higher than those described in [1,2]. As for problems considered, our paper includes decentralized variants of Adam and its analysis to achieve good performance also on Transformer-based models. Moreover, [1,2] investigated communication scheduling in asymmetric communication environments, which is related to our findings in Section 2.1.
>
> In Section 1.1, we discuss decentralized training, several works focusing on practical speedup, and some orthogonal techniques including compression. We believe that [1,2] will be good additions to both "Decentralized training" and "Orthogonal scalability techniques".
>
> **W2 (overlapping of comm. and comp.)**: We agree that the idea of using adaptive momentum methods is not novel, and the technical innovation in the convergence proof is indeed limited. However, we do believe that the aggregation mechanism is novel, and it is an essential component in attaining our level of performance in both speedup and generalization. In both image classification and machine translation, the proposed variant of DAdam outperforms DAdam by a margin in generalization performance while maintaining similar speedup performance.
>
> As for the overlapping between communication and computation, we are aware that the overlapping by the gradient bucketing technique is widely used, and we also mentioned it at the beginning of Section 2.1. We would like to emphasize that one of our contributions is to further extend this technique the decentralized training. In decentralized training, the model parameters, instead of gradients, are transmitted among workers, and the local updates depend on the information of its neighbors from the last iteration. The removal of the dependency of local updates on the information from the current iteration further enhances this technique, since it allows for overlapping between the forward pass and the communication.
>
> As for [3,4], the authors focus on achieving higher computation resource utilization in the context of pipeline parallelism and tensor parallelism, while we focus on data parallelism. The distributed strategies distinguish our paper from [3, 4], but it is certainly possible to combine all of them. In the pre-training of LLMs, for example, it is common to see multi-dimensional parallelism in use (like [DeepSpeed](https://github.com/microsoft/DeepSpeed) and [Colossal AI](https://colossalai.org/)). It will be interesting to see how the optimization techniques in different dimensions can be combined and contribute to speedup at an even larger scale. However, we defer such studies to future work.
>
> [1] GossipFL: A Decentralized Federated Learning Framework With Sparsified and Adaptive Communication. In TPDS 2022.
>
> [2] Communication-efficient decentralized learning with sparsification and adaptive peer selection. In ICDCS 2020.
>
> [3] Overlap Communication with Dependent Computation via Decomposition in Large Deep Learning Models. In ASPLOS 2022.
>
> [4] Centauri: Enabling Efficient Scheduling for Communication-Computation Overlap in Large Model Training via Communication Partitioning. In ASPLOS 2024.

---

> ### Author Response · Authors · 2024-11-19
> **Response to Reviewer LAtq (Part 2/2)**
>
> **W3 (ablation study)**: Thanks for the valuable comment. We agree that the ablation study will make the presentation clearer to the readers. From the perspective of the necessity of the second-order momentum in the optimizer, we conducted additional experiments with decentralized SGD that we have included in the supplementary material (see Section "Transformer with SGD Optimizer"). We would like to emphasize that we already have the comparisons with the data parallal implementation in PyTorch, stochastic gradiant push, various communication topology, and decentralized Adam with accumulation trick. We believe that we can create a section that performs a cleaner ablation study by combining and re-organizing these results.
>
> **W4 (topology)**: Please refer to Table 1 for the communication time of various topologies. The choices of topologies are subject to the structure of the cluster. We are aware of other topologies like the torus graph and the expander graph, but we believe the chosen topologies are suitable to demonstrate advantages in either runtime speedup or convergence rates. In the numerical experiments, we chose `complete` and `alternating-exp-ring` because other topologies may perform worse in both convergence rate and practical runtime. In Section A.8.1 in the appendix, we have results with the one-peer ring topology for the generalization performance on the machine translation task.
>
> **W5 & Q1 (convergence behavior on GPT-2)**: Since the training details of GPT-2 are not public, we adapted the code from an open-source implementation from [nanoGPT](https://github.com/karpathy/nanoGPT). In the results of nanoGPT, the GPT-2 (small) was trained for 400k steps, which we further extended to 600k steps and verified comparable generalization performance of decentralized training in our numerical experiments. In the newly uploaded supplementary material, we provide the training curves of training and validation losses.
>
> **W6 & Q2 (implementation detail)**: In the section "Implementation Details of Decentralized Training Extension", we have provided examples of how to launch the training and how to use the extension. We have also uploaded the source code of the PyTorch extension in a separate `code` folder. Please refer to the supplementary material for details.

---

> > ### Comment · Reviewer_LAtq · 2024-11-25
> > **Thanks for responses**
> >
> > 1. I agree that this work focuses on the relativelty high-bandwidth scenarios. However, those works in the low-bandwidth scenarios and this work maybe both general in both low- and high-bandwidth scenarios. Nevertheless, the new description nicely distinguishes this work from the previous one.
> > 2. I find that I cannot open the provided supplementary fold due to the without permission. Could you check it? For both reviewing both the convergence curve and the code implementation.

---

> > > ### Author Response · Authors · 2024-11-25
> > > **Response to Reviewer LAtq**
> > >
> > > 1. Thank you for your positive comments about the revisions!
> > > 2. In the first rebuttal revision, we generated the supplementary in a Linux system, which could be the reason. We did check the files are accessible under the Windows system. We now have uploaded another revision. Please check if the permission problem is fixed. Sorry for the inconvenience.

---

> > > > ### Comment · Reviewer_LAtq · 2024-11-25
> > > > **Thanks for responses**
> > > >
> > > > The provided convergence curve helps to illustrate the convergence behavior of the decentralized training. I have checked the codes. Could you provide the other codes including parts of training GPT-2 with OpenWebText and the complete scripts to allow convenient reproducing?

---

> > > > > ### Author Response · Authors · 2024-11-25
> > > > > **Response to Reviewer LAtq**
> > > > >
> > > > > Thank you for acknowledging the clarity of our presentation and for taking the time to review the core part of our code. We truly appreciate your engagement and valuable feedback!
> > > > >
> > > > > While we recognize the importance of providing the complete project code, we respectfully feel that it would be challenging to anonymize and test the full codebase within the constraints of the discussion period. Specifically, our project uses the core code—already shared as a Python package—as a dependency. To ensure anonymity, we would need to remove this dependency to avoid revealing our identities. However, this process risks introducing bugs, and the updated code would require testing. Considering the computational cost of this testing—approximately 30 hours on 16 A100 GPUs for one run—it is unlikely we could complete the release of an anonymized and fully tested codebase within the required timeframe.
> > > > >
> > > > > To address these constraints, we are committed to releasing the complete project code upon acceptance, ensuring transparency while adhering to review guidelines. We believe that the core code, which we have already shared, effectively demonstrates the feasibility and practicality of our approach. The detailed explanations in the supplementary materials further support reproducibility and offer a clear understanding of our method.
> > > > >
> > > > > We hope these steps reflect our commitment to transparency and rigor. In light of the efforts we have already made to share the most critical elements of our work, we kindly request you to reconsider the score.

---

> > > > > > ### Comment · Reviewer_LAtq · 2024-11-26
> > > > > > **Thanks for the responses**
> > > > > >
> > > > > > Thanks. Based on the commitment of open source, I'd like to increase my score, but decrease the confidence.

---

> > > > > > > ### Author Response · Authors · 2024-11-29
> > > > > > > **Reponse to Reviewer LAtq**
> > > > > > >
> > > > > > > Thank you for raising the score and recognizing our efforts and commitment to improving the paper! With the extended discussion period, we have successfully reorganized, anonymized, and thoroughly tested the codebase for training GPT-2 using both AllReduce and decentralized methods. We are pleased to share the link to the anonymous repository on Anonymous GitHub: [Decentralized GPT-2 Codebase](https://anonymous.4open.science/r/Decentralized-GPT2-362A). We hope this codebase effectively demonstrates the practicality of our approach and addresses your concerns, and we welcome any additional comments or suggestions to further enhance the quality of our work.

---

### Official Review · Reviewer_pREt · 2024-11-03

**Soundness:** 3
**Presentation:** 3
**Contribution:** 3
**Rating:** 8
**Confidence:** 3

**Summary:**

This paper investigates the gap between decentralized training of DNN and data parallel (DP) methods.
In particularly, although decentralized training is theoretically superior to DP, certain network topologies make the use of this technique challenging.
The present propose to quantify such gap through the use of a "runtime model".
Moreover, the author introduce a decentralized version of Adam particularly suited to the training of transformer based models.
Extensive numerical experiments illustrate the speedup investigation.

**Strengths:**

- This paper proposes to study an important, impactful problem (practical gap between DP and decentralized training)
- the accuracy of the model (ie how it captures real time speed discrepancies) is well illustrated
- the numerical experiments are extensive and thorough. There is a clear focus on transformer based architectures, making it relevant to challenges posed by modern LLMs.
- the version of decentralized Adam optimizer is thoroughly compared to other decentralized adam based methods
- the papers lists 3 factors that could explain the aforementioned discrepancy  1) overlapping communication and computation 2) heterogeneous communications costs 3) sensitivity to varying computation times. For each factor, an extensive investigation is proposed supported by numerical experiments.

**Weaknesses:**

- the explanation/details of the runtime model in Appendix A.5 could be stated more clearly
(the reviewers would like to respectfully suggest an explanation similar in spirit to the way Algorithm 1 is stated, although this is relatively minor)
- there is no clear link to the reviewer between the decentralized adam optimizer and the runtime model (please correct me if missed anything).

**Questions:**

The reviewer is curious about the intuition behind the runtime model: were the assumptions behind the model proposed previously by some other work or is this coming solely from the author?

---

> ### Author Response · Authors · 2024-11-19
> **Response to Reviewer pREt**
>
> We thank the reviewer for acknowledging the strong performance of our work and the quality of the presentation. Below, we answer the questions and weaknesses that you have raised in your review. Please note that we have updated the supplementary material to include a pseudo code listing that clarifies the idea of the runtime model. We refer to this new supplementary in our response below.
>
> **W1 (explanation of the runtime model)**: Thank you for pointing out parts of our paper that you find unclear. We agree that the dependency relationships are hard to visualize. To address your concern, we have annotated the pseudocode for the main worker thread with references to the sequential timeline in Appendix A.5. Please refer to the algorithm in the last part of the section "Implementation Details of Decentralized Training Extension" in the new supplementary material for the pseudocode listing. We hope that this, together with Section A.5 in the appendix, will explain the mechanism in a clearer way.
>
> **W2 (connection between runtime model and DAdam optimizer)**: In our paper, we first explored key factors that enable speedups of decentralized training and proposed the runtime model to estimate the influences of these factors on the runtime. One important insight is that the algorithm should follow the combine-then-adapt (CTA) scheme of decentralized gradient descent (see Section 2.1), which is why the decentralized Adam variant in our paper is designed to belong to this family of algorithms. Moreover, speedup and generalization performance are two essential aspects of the practical decentralized training pipeline, and the runtime model and the decentralized Adam optimizer represent two complimentary contributions toward this goal.
>
> **Q1 (inspiration of the runtime model)**: To assess the efficiency of our decentralized training extension and ensure that other factors like data loading do not impede the speedup brought by decentralized training, we used the [PyTorch profiler](https://pytorch.org/tutorials/recipes/recipes/profiler_recipe.html) to record computation and communication times in our initial experiments. Through experiments conducted on different hardware setups, we found a strong interplay between the organization of the training algorithm, the communication topology, and the hardware configuration. These observations inspired us to quantify the influence and to design a runtime model to gain insight into tradeoffs and guide the overall design process.

---

### Official Review · Reviewer_oHbd · 2024-11-03

**Soundness:** 2
**Presentation:** 3
**Contribution:** 2
**Rating:** 6
**Confidence:** 4

**Summary:**

The paper investigates decentralized training of DNNs. This used to be a fairly hot topic a few years ago (e.g. Lian et al 2017 and follow-up work), but has to some extent cooled down recently as models and workloads have changed, and model-parallelism has become more standard.
The paper focuses on the classic data-parallel setting; its contributions are as follows:

- An analytical model to understand what are the conditions under which decentralized training can bring gains.

- A decentralized version of Adam, which decouples computation from communication and provides convergence guarantees under fairly standard assumptions (for Adam)

- An implementation of the algorithm and its technical evaluation. This is done fairly thoroughly, on systems with up to 64 GPUs, showing the method’s potential.

**Strengths:**

Strengths:

1. new perspectives on an “old” problem by today’s standards (decentralized training), from the point of view of modeling and adaptive optimization

2. analytical results are a plus

3. the experiments are fairly thorough

**Weaknesses:**

Weaknesses:

1. unfortunately the paper seems to be completely missed some important related work in the area, which makes it very hard to position the paper properly in terms of its contribution

2. more broadly, the experimental results are in a system parameter range that has been rendered somewhat obsolete by current-day systems, which are able to e.g. train ImageNet in minutes on a single node https://github.com/libffcv/ffcv

**Questions:**

Detailed comments and questions:

Q1. The paper seems to ignore work on the decentralized setting since late 2020 to early 2021. Here are some of the many references that are missed:

- There is a lot of nice work by Anastasia Koloskova and co-authors on analyzing Gossip variants of SGD, many of which are missed:

1. Koloskova, Anastasia, et al. "A unified theory of decentralized sgd with changing topology and local updates." International Conference on Machine Learning. PMLR, 2020.

2. Koloskova, Anastasiia, Tao Lin, and Sebastian U. Stich. "An improved analysis of gradient tracking for decentralized machine learning." Advances in Neural Information Processing Systems 34 (2021): 11422-11435.

See also:

Zhang, Jiaqi, and Keyou You. "Fully asynchronous distributed optimization with linear convergence in directed networks." arXiv preprint arXiv:1901.08215 (2019).

"SQuARM-SGD: Communication-Efficient Momentum SGD for Decentralized Optimization" by Navjot Singh et al. (2020)

In addition:

- Nadiradze, Giorgi, et al. "Asynchronous decentralized SGD with quantized and local updates." Advances in Neural Information Processing Systems 34 (2021): 6829-6842.

This paper seems to be trying to do something similar to what is done here–decoupling communication from computation–but in some sense goes further since it also supports quantization and completely non-blocking reads. Even the experimental setup is very similar to the one presented in this paper, so I am surprised to see that there is no mention of this work whatsoever.

- Similar work:
 Li, Shigang, et al. "Breaking (global) barriers in parallel stochastic optimization with wait-avoiding group averaging." IEEE Transactions on Parallel and Distributed Systems 32.7 (2020): 1725-1739.

I would therefore recommend that the authors go over related work more thoroughly, and provide more complete positioning of their work relative to prior art. I don’t think the paper can be accepted in the absence of this.

Q2: Are the loss improvements quoted in L488 (2.841 vs 2.846) actually statistically significant?

Q3: Is your setup really faster than just setting up large-batch SGD using e.g. FFCV on a single 4-GPU system?

---

> ### Author Response · Authors · 2024-11-19
> **Response to Reviewer oHbd (Part 1/2)**
>
> We appreciate the time and effort that you have invested in providing us with constructive feedback. It has been invaluable in strengthening our paper.
>
> In what follows, we address all the questions that you raised in your initial review. Note that our reply is separated in two, to meet the character limit in OpenReview. Please, also note that we have uploaded a supplementary material in the rebuttal revision that describes the new experiments with FFCV and provides additional implementation details. We refer to it in our response below.
>
> **Q1 (related work)**: Thank you for bringing these papers to our attention. While our work is related to Papers 1–6, it differs in several critical aspects. Notably, apart from Paper 4, which employs first-order momentum, the other papers do not utilize momentum. In contrast, our proposed optimizer is an Adam-based adaptive method that incorporates both first- and second-order momentum, necessitating a distinct algorithmic design and analysis. Moreover, our paper focuses on the design and implementation space, particularly the runtime model, which distinguishes it from these other works. We have already included comparisons with related research in Sections 1.1 and Appendix B.2 of the submitted version, where some of the papers are similar to those you suggested. We will integrate the additional papers that you recommended into our revision. Below, we outline how they can be incorporated.
>
> Our algorithm builds upon decentralized gradient descent (DGD) and Adam. Consequently, we have compared it against DGD and state-of-the-art decentralized Adam-based algorithms. Please refer to Section 1.1 and Appendix B.2 for further details. The algorithm by Chen et al. (2023), for instance, adopts a tracking technique similar to gradient tracking (GT) (see line 11 of Algorithm 2). As noted in our introduction of Chen et al. (2023), this tracking technique requires an additional round of communication, whereas our algorithm requires only a single round of communication in each iteration. Papers 1-4 in your review fall into the categories of DGD, GT, and extensions of GT to directed graphs, all of which can be integrated into the comparison in Section 1.1 and Appendix B.2. Specifically, here is an overview of the works you referenced as papers 1-4: Paper 1 proposes a unified theory of decentralized SGD with time-varying topology and local updates. Paper 2 improves the convergence rate of gradient tracking and the relationship with network topology parameters by leveraging new analytical techniques. Paper 3 extends the push-pull algorithm to asynchronous scenarios and establishes a linear convergence rate under the assumption of strongly convex problems. Paper 4 presents a decentralized first-order momentum SGD which employs quantization to reduce communication costs.
>
> In Section 1.1 of our submission, we discuss decentralized training and reference practical studies (e.g., Lian et al., 2017; Assran et al., 2019) reporting speedups from decentralized approaches. We also briefly review orthogonal scalability techniques like compression. Papers 5 and 6, which you mentioned, can complement this discussion. Specifically, Paper 6 introduces wait-avoiding group averaging SGD, which uses group AllReduce to limit communication operations within non-overlapping process groups while requiring periodic global synchronization for convergence. Paper 5 extends this with SwarmSGD, a variant of SGD that integrates non-blocking communication, quantization, and local steps, demonstrating convergence through randomized, pairwise communication among nodes.
>
> We have highlighted the contributions and focus of this paper in the general response. Combined with the comparisons to related work provided here, we hope to have clarified the position of the paper.
>
> [1] Koloskova, Anastasia, et al. "A unified theory of decentralized sgd with changing topology and local updates." International Conference on Machine Learning. PMLR, 2020.
>
> [2] Koloskova, Anastasiia, Tao Lin, and Sebastian U. Stich. "An improved analysis of gradient tracking for decentralized machine learning." Advances in Neural Information Processing Systems 34 (2021): 11422-11435.
>
> [3] Zhang, Jiaqi, and Keyou You. "Fully asynchronous distributed optimization with linear convergence in directed networks." arXiv preprint arXiv:1901.08215 (2019).
>
> [4] Singh, Navjot, et al. "SQuARM-SGD: Communication-efficient momentum SGD for decentralized optimization." IEEE Journal on Selected Areas in Information Theory 2.3 (2021): 954-969.
>
> [5] Nadiradze, Giorgi, et al. "Asynchronous decentralized SGD with quantized and local updates." Advances in Neural Information Processing Systems 34 (2021): 6829-6842.
>
> [6] Li, Shigang, et al. "Breaking (global) barriers in parallel stochastic optimization with wait-avoiding group averaging." IEEE Transactions on Parallel and Distributed Systems 32.7 (2020): 1725-1739.

---

> ### Author Response · Authors · 2024-11-19
> **Response to Reviewer oHbd (Part 2/2)**
>
> **Q2 (statistical difference in losses)**: This is a great question. Indeed, the difference in loss is **not** statistically significant, which supports our conclusion that decentralized training can achieve comparable generalization performance to AllReduce training with the same iteration budget.
>
> **Q3 (FFCV)**: Thank you for sharing this framework which also focuses on accelerating the DNN training systems. We would like to emphasize that our contribution in this paper is orthogonal to the idea of FFCV [7]. The FFCV framework accelerates training by removing bottlenecks in data loading and data augmentation. To be specific, the framework achieves its goal by loading the whole dataset into the memory, avoiding the much slower disk IO, and compiling image augmentation and preprocessing operations into high-performance kernels, which alleviates the bottleneck caused by the CPUs. The paper does not contribute to improving the throughput of GPUs, reducing communication overhead, or handling imbalanced workloads, and the distributed strategy of its implementation in [fast training on ImageNet](https://github.com/libffcv/ffcv-imagenet/tree/main) is based on `DistributedDataParallel` of PyTorch which is exactly the AllReduce training baseline that we used in our numerical experiments. Our paper explores the potential of decentralized training in achieving better utilization of computational resources (see Sections 2 and 3), which makes it a better alternative than AllReduce training. Nevertheless, we have integrated our decentralized training extension with the FFCV framework and performed runtime performance comparisons that we report in the recently revised supplementary material. The numerical experiments demonstrate that our decentralized training framework brings better scalability to the FFCV framework. Please see `supplementary.pdf` in the zip file for details.
>
> [7] Leclerc, Guillaume, et al. "FFCV: Accelerating training by removing data bottlenecks." Proceedings of the IEEE/CVF Conference on Computer Vision and Pattern Recognition. 2023.

---

> > ### Comment · Reviewer_oHbd · 2024-11-25
> > **Thanks**
> >
> > Thank you for the response; I have decided to upgrade my score to 6, to reflect the clarifications by the authors. \
> > I am not going higher since I am not fully convinced by the practical potential of the method, and since it is highly unusual for a submission to miss such a large amount of related work.

---

> > > ### Author Response · Authors · 2024-11-29
> > > **Response to Reviewer oHbd**
> > >
> > > We sincerely thank the reviewer for the valuable suggestions. Our paper covers a broad scope, including system design, training algorithm development, and evaluation, and we aimed to capture key works across these areas through the nearly four pages of references already included. Nevertheless, we appreciate the reviewer’s recommendation to further refine our selection of references. We will carefully incorporate the suggested references into Sections 1.1 and Appendix B.2, where related work closely aligned with these papers is discussed, and will clearly highlight the key differences and connections between our work and Papers 1–6 in the final manuscript.  As for the practicality of the method, to reply to the reviewer LAtq, we have uploaded the anonymized repository of training GPT-2 to Anonymous GitHub ([link](https://anonymous.4open.science/r/Decentralized-GPT2-362A/)), and we hope that your concern can be addressed by the implementation.
> > >
> > > Thank you again for your thoughtful feedback, which has helped us enhance the quality of our paper.

---

### Author Response · Authors · 2024-11-19
**General Response**

We would like to thank all the reviewers for their kind and constructive reviews of our paper.

Generally, the reviewers affirmed our contributions on
1. the runtime model that quantifies key factors in enabling speedup of decentralized training,
2. thorough numerical experiments with a focus on Transformer-based models, and
3. design and analysis of decentralized optimizers with second-order adaptive momentum.

The reviewers also raised questions about
1. related works in decentralized training and the benefits over other efficient training frameworks,
2. a few minor issues that were unclear in the initial submission, and
3. implementation details of the decentralized training extension.

In the responses, we addressed the questions by:
1. clarifying how the related work suggested by the reviewers relates to our paper, conducting new experiments to highlight differences, and describing how the reviewers' suggestions will be incorporated in the revised version of our paper,
2. providing additional explanations and experiments that clarify our work, and
3. uploading the source code of our implementation with examples and detailed explanations.

In general, our paper investigates the design space of decentralized training systems that facilitate practical runtime speedups over AllReduce training and maintain a comparable generalization performance, especially on Transformer-based models. Key contributions include a runtime model that exposes tradeoffs and guides the overall design process, a decentralized variant of the Adam optimizer with a novel accumulation mechanism, an overall system design implemented as a PyTorch extension, and extensive numerical experiments that validate both runtime and generalization performance.

---

### Meta-Review · Area_Chair_3P8u · 2024-12-20

**Metareview:**

The paper studies decentralized training in the classic data parallel setting. It gives an analytical model to understand under what conditions decentralized training can bring gains. It also gives a relevant decentralized version of Adam, with a novel convergence analysis. Finally empirical results are convincing on realistic  settings with up to 64 GPUs.
Some concerns were raised about missing discussions of related work, which we hope authors will resolve in the final version.
All four reviewers ultimately recommend acceptance.

**Additional Comments On Reviewer Discussion:**

The author feedback phase was productive, converging to good agreement between all parties

---

### Decision · Program_Chairs · 2025-01-22

Accept (Poster)